# Nonconvex Sparse Graph Learning under Laplacian Constrained Graphical Model

**Jiaxi Ying**[1]
jx.ying@connect.ust.hk

**José Vinícius de M. Cardoso**[1]
jvdmc@connect.ust.hk

**Daniel P. Palomar**[1,2]
palomar@ust.hk

Department of Electronic and Computer Engineering[1]
Department of Industrial Engineering and Decision Analytics[2]
The Hong Kong University of Science and Technology, Clear Water Bay, Hong Kong

## Abstract

In this paper, we consider the problem of learning a sparse graph from the Laplacian constrained Gaussian graphical model. This problem can be formulated as a penalized maximum likelihood estimation of the precision matrix under Laplacian structural constraints. Like in the classical graphical lasso problem, recent works made use of the $\ell_1$-norm with the goal of promoting sparsity in the Laplacian constrained precision matrix estimation. However, through empirical evidence, we observe that the $\ell_1$-norm is not effective in imposing a sparse solution in this problem. From a theoretical perspective, we prove that a large regularization parameter will surprisingly lead to a solution representing a complete graph, i.e., every pair of vertices is connected by an edge. To address this issue, we propose a nonconvex penalized maximum likelihood estimation method, and establish the order of the statistical error. Numerical experiments involving synthetic and real-world data sets demonstrate the effectiveness of the proposed method. An open source R package is available at https://github.com/mirca/sparseGraph.

## 1 Introduction

Gaussian graphical models (GGM) have been widely used in a number of fields such as finance, bioinformatics, and image analysis [1, 31, 38]. Graph learning under GGM can be formulated to estimate the precision matrix that captures the conditional dependency relations between random variables [10, 29]. In this paper, the goal is to learn a sparse graph under the Laplacian constrained GGM, where the precision matrix obeys Laplacian structural constraints.

The general GGM has received broad interest in statistical machine learning, where the problem can be formulated as a sparse precision matrix estimation. The papers [1, 9, 58] proposed the $\ell_1$-norm penalized maximum likelihood estimation, also known as graphical lasso, to encourage sparsity in its entries. Various extensions of graphical lasso and their theoretical properties are also studied [11, 18, 21, 35, 41, 42, 47, 55, 56]. To reduce the estimation bias, nonconvex penalties like the smooth clipped absolute deviation (SCAD) [16], minimax concave penalty (MCP) [60], and capped $\ell_1$-penalty [61] have been introduced in estimating a sparse precision matrix [2, 7, 27, 33, 46, 54, 60]. However, those methods mentioned above focus on general graphical models, and cannot be directly extended to Laplacian constrained GGM because of the multiple constraints on the precision matrices. Moreover, unlike the above GGM cases, this paper will show that the $\ell_1$-norm is not effective in

promoting sparsity in the penalized maximum likelihood estimation of the Laplacian constrained precision matrices.

In recent years, Laplacian constrained GGM has received increasing attention in signal processing and machine learning over graphs [6, 13, 23, 26, 30, 37, 48]. Under the Laplacian constrained GGM, graph learning can be formulated as Laplacian constrained precision matrix estimation. Unlike the general GGM, the precision matrix in Laplacian constrained GGM enjoys the spectral property that its eigenvalues and eigenvectors can be interpreted as spectral frequencies and Fourier basis [48], which is useful in computing graph Fourier transform in graph signal processing [37, 48], and graph convolutional networks [3, 36, 45]. The authors in [12, 14, 19, 59] formulated the graph signals as random variables under the Laplacian constrained GGM. The learned graph under Laplacian constrained GGM favours smooth graph signal representations [12], since the graph Laplacian quadratic term quantifies the smoothness of graph signals [22, 24]. However, sparse graph learning under the Laplacian constrained GGM remains to be further explored. For example, how to effectively and efficiently learn a sparse graph and how to characterize the estimation error under the Laplacian constrained GGM are to be investigated.

This paper focuses on the problem of learning a sparse graph under the Laplacian constrained GGM. The contributions of this paper are summarized as follows. First, we find an unexpected behavior of the $\ell_1$-norm in Laplacian constrained GGM. Through empirical evidence, we observe that the widely used $\ell_1$-norm is not effective in imposing a sparse solution under the Laplacian constrained GGM. From a theoretical perspective, we prove that a large regularization parameter of the $\ell_1$-norm will lead to a solution representing a complete graph, i.e., every pair of vertices is connected by an edge, instead of a sparse graph. Second, we propose a nonconvex penalized maximum likelihood estimation method by solving a sequence of weighted $\ell_1$-norm penalized sub-problems, and establish the order of the statistical error. To the best of our knowledge, this is the first work to analyze the non-asymptotic optimization performance guarantees on both optimization error and statistical error under the Laplacian constrained GGM. Finally, numerical experiments on both synthetic and real-world data sets demonstrate the effectiveness of the proposed method.

The remainder of the paper is organized as follows. Problem formulation and related work are provided in Section 2. We present the proposed method and the theoretical results in Section 3. Experimental results are provided in Section 4. We draw the conclusions in Section 5.

**Notation**   Lower case bold letters denote vectors and upper case bold letters denote matrices. Both $X_{ij}$ and $[\boldsymbol{X}]_{ij}$ denote the $(i,j)$-th entry of the matrix $\boldsymbol{X}$. $\boldsymbol{X}^\top$ denotes transpose of matrix $\boldsymbol{X}$. $[p]$ denotes the set $\{1, \ldots, p\}$. The all-zero and all-one vectors or matrices are denoted by $\boldsymbol{0}$ and $\boldsymbol{1}$, respectively. $\|\boldsymbol{x}\|$, $\|\boldsymbol{X}\|_{\mathrm{F}}$ and $\|\boldsymbol{X}\|_2$ denote Euclidean norm, Frobenius norm, and operator norm, respectively. Let $\mathrm{supp}^+(\boldsymbol{x}) = \{i \in [p]\,|\,x_i > 0\}$ for $\boldsymbol{x} \in \mathbb{R}^p$. $\lceil x \rceil$ denotes the least integer greater than or equal to $x$. Let $\|\boldsymbol{x}\|_{\max} = \max_i |x_i|$ and $\langle \boldsymbol{x}, \boldsymbol{y} \rangle = \sum_i x_i y_i$. For functions $f(n)$ and $g(n)$, we use $f(n) \lesssim g(n)$ if $f(n) \leq C g(n)$ for some constant $C \in (0, +\infty)$. $\mathcal{S}_+^p$ and $\mathcal{S}_{++}^p$ denote the sets of positive semi-definite and positive definite matrices with the size $p \times p$, respectively.

## 2   Problem Formulation and Related Work

We first formulate the problem of learning a graph under the Laplacian constrained GGM. After that, we discuss related work.

### 2.1   Laplacian Constrained Gaussian Graphical Model

We define a weighted, undirected graph $\mathcal{G} = (\mathcal{V}, \mathcal{E}, \boldsymbol{W})$, where $\mathcal{V}$ denotes the set of nodes, and the pair $(i, j) \in \mathcal{E}$ if and only if there is an edge between node $i$ and node $j$. $\boldsymbol{W} \in \mathbb{R}_+^{p \times p}$ is the weighted adjacency matrix with $W_{ij}$ denoting the graph weight between nodes $i$ and $j$. The graph Laplacian $\boldsymbol{L} \in \mathbb{R}^{p \times p}$, also known as combinatorial graph Laplacian, is defined as $\boldsymbol{L} = \boldsymbol{D} - \boldsymbol{W}$, where $\boldsymbol{D}$ is a diagonal matrix with $D_{ii} = \sum_{j=1}^p W_{ij}$. In this paper, we focus on the case of connected graphs, implying that there is only one graph component. According to spectral graph theory [8], the rank of the Laplacian matrix for a connected graph with $p$ nodes is $p - 1$. Then it is easy to check that the set of Laplacian matrices for connected graphs can be formulated as

$$\mathcal{S}_L = \{\boldsymbol{\Theta} \in \mathcal{S}_+^p \,|\, \Theta_{ij} = \Theta_{ji} \leq 0,\, \forall\, i \neq j,\, \boldsymbol{\Theta} \cdot \boldsymbol{1} = \boldsymbol{0},\, \mathrm{rank}(\boldsymbol{\Theta}) = p - 1\}, \tag{1}$$

where $\mathbf{0}$ and $\mathbf{1}$ denote the constant zero and one vectors, respectively. Next, we will define Laplacian constrained Gaussian Markov random fields, and without loss of generality we assume the random vector $\boldsymbol{x}$ has zero mean.

**Definition 2.1.** A zero-mean random vector $\boldsymbol{x} = [x_1, \ldots, x_p]^\top \in V^{p-1}$ is called a Laplacian constrained Gaussian Markov Random Fields (LGMRF) with parameters $(\mathbf{0}, \boldsymbol{\Theta})$ with $\boldsymbol{\Theta} \in \mathcal{S}_L$, if and only if its density function $q_L : V^{p-1} \to \mathbb{R}$ follows

$$q_L(\boldsymbol{x}) = (2\pi)^{-\frac{p-1}{2}} \det{}^\star(\boldsymbol{\Theta})^{\frac{1}{2}} \exp\left(-\frac{1}{2}\boldsymbol{x}^\top \boldsymbol{\Theta} \boldsymbol{x}\right), \tag{2}$$

where $\det{}^\star$ denotes the pseudo determinant defined by the product of nonzero eigenvalues [20], $V^{p-1}$ is a $(p-1)$-dimensional subspace of $\mathbb{R}^p$ defined by $V^{p-1} := \{\boldsymbol{x} \in \mathbb{R}^p | \mathbf{1}^\top \boldsymbol{x} = 0\}$.

Note that we restrict $\boldsymbol{x}$ into a subspace because the LGMRF does not have a density with respect to the $p$-dimensional Lebesgue measure. According to the disintegration theorem, we can construct a conditional probability measure defined on $V^{p-1}$ and then the density of LGMRF with this measure satisfies (2). In this sense, the LGMRF can be interpreted as a GMRF conditioned on the linear constraint $\mathbf{1}^\top \boldsymbol{x} = 0$ and thus each observation $\boldsymbol{x}^{(k)}$ of an LGMRF also satisfies $\mathbf{1}^\top \boldsymbol{x}^{(k)} = 0$. For convenience, we still denote $\boldsymbol{\Theta}$ in (2) as the precision matrix, though it formally does not exist [44].

Sparse graph learning under the Laplacian constrained Gaussian graphical model could be formulated as the penalized maximum likelihood of the precision matrix with Laplacian structural constraints,

$$\min_{\boldsymbol{\Theta} \in \mathcal{S}_L} -\log\det(\boldsymbol{\Theta} + \boldsymbol{J}) + \operatorname{tr}(\boldsymbol{\Theta}\boldsymbol{S}) + \sum_{i>j} h_\lambda(\Theta_{ij}), \tag{3}$$

where $\boldsymbol{S}$ is the sample covariance matrix, $h_\lambda$ is a *regularizer*, depending on a regularization parameter $\lambda \geq 0$, which serves to enforce sparsity. For example, $h_\lambda(\Theta_{ij}) = \lambda|\Theta_{ij}|$. $\boldsymbol{J} = \frac{1}{p}\mathbf{1}_{p\times p}$ is a constant matrix with each element equal to $\frac{1}{p}$. Note that we replace $\det{}^\star(\boldsymbol{\Theta})$ with $\det(\boldsymbol{\Theta} + \boldsymbol{J})$ in (3) as done in [14], because the matrix $\boldsymbol{J}$ is rank-1 and the nonzero eigenvalue of $\boldsymbol{J}$ is 1 with the associated eigenvector orthogonal to the row and column spaces of $\boldsymbol{\Theta} \in \mathcal{S}_L$.

## 2.2 Related Work

Recently, the authors in [14, 24, 32, 62] proposed the $\ell_1$-norm penalized maximum likelihood estimation under the Laplacian constrained GGM to learn a sparse graph. The authors in [14, 62] designed a primal-dual algorithm that introduces additional variables to handle Laplacian structural constraints, while the authors in [32] proposed a block coordinate descent method to solve the optimization problem. To learn a structured graph such as $k$-component graph, the authors in [24] proposed the $\ell_1$-norm regularized maximum likelihood with Laplacian spectral constrains. More recently, the authors in [25] proposed a framework with re-weighted $\ell_1$-norm to learn structured graphs by imposing spectral constraints on graph matrices. However, the proposed algorithms in [24, 25] have to compute the eigenvalue decomposition in each iteration which is computationally expensive in the high-dimensional regime. Note that all the methods mentioned above lack theoretical analysis on estimation error. In this paper, we will show that the $\ell_1$-norm is not effective in promoting sparsity under the Laplacian constrained GGM, and further propose a nonconvex estimation method with theoretical guarantees on estimation error.

# 3 $\ell_1$-norm Analysis and Proposed Method

In this section, we first present an unexpected behavior of the $\ell_1$-norm in learning a sparse graph under the Laplacian constrained GGM. Then, we propose a nonconvex penalized maximum likelihood estimation method. Finally, we present the theoretical results in the analysis of estimation error. The proofs of all the theorems are deferred to the supplementary material.

## 3.1 $\ell_1$-norm Regularizer

Sparsity is often explored in high-dimensional Gaussian graphical models in order to reduce the number of samples required. The effectiveness of the $\ell_1$-norm regularized maximum likelihood estimation has been widely demonstrated in a number of fields. One common rule of thumb for

graphical lasso is that the estimated graph will get sparser when a larger regularization parameter is used. However, we find an unexpected behavior of the $\ell_1$-norm in the Laplacian constrained GGM.

The $\ell_1$-norm regularized maximum likelihood estimation of the Laplacian constrained precision matrices [14, 62] can be formulated as

$$\min_{\boldsymbol{\Theta} \in \mathcal{S}_L} -\log \det(\boldsymbol{\Theta} + \boldsymbol{J}) + \operatorname{tr}(\boldsymbol{\Theta S}) + \lambda \sum_{i>j} |\Theta_{ij}|, \tag{4}$$

where $\boldsymbol{S}$ is the sample covariance matrix and $\lambda$ is the regularization parameter. It is easy to check that (4) is a convex optimization problem.

**Theorem 3.1.** *Let $\hat{\boldsymbol{\Theta}} \in \mathbb{R}^{p \times p}$ be the global minimum of (4) with $p > 3$. Define $s_1 = \max_k S_{kk}$ and $s_2 = \min_{ij} S_{ij}$. If the regularization parameter $\lambda$ in (4) satisfies $\lambda \in [(2 + 2\sqrt{2})(p + 1)(s_1 - s_2), +\infty)$, then the estimated graph weight $\hat{W}_{ij} = -\hat{\Theta}_{ij}$ obeys*

$$\hat{W}_{ij} \geq \frac{1}{(s_1 - (p+1)s_2 + \lambda)p} > 0, \quad \forall\, i \neq j.$$

Theorem 3.1 ensures that a large regularization parameter of the $\ell_1$-norm will make every graph weight strictly positive and thus the estimated graph is a complete graph, i.e., every pair of vertices is connected by an edge. This theoretical result is consistent with empirical observations depicted in Figure 1, where the number of edges in the graph estimated by the optimization (4) grows along with the increase of $\lambda$, and the estimated graph in Figure 1(d) is fully connected with all the graph weights strictly positive and small.

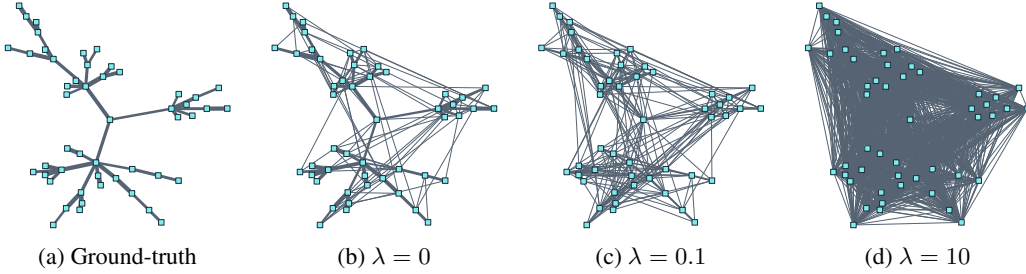

|  (a) Ground-truth  |  (b) $\lambda = 0$  |  (c) $\lambda = 0.1$  |  (d) $\lambda = 10$  |

Figure 1: Graph learning using the $\ell_1$-norm regularization with different regularization parameters. The number of nonzero edges in (b), (c) and (d) are 135, 286 and 1225, respectively. The true graph in (a) has 49 edges and the graph in (d) is fully connected. The relative errors of the learned graphs in (b), (c) and (d) are 0.14, 0.64 and 0.99, respectively.

It is well-known that a larger regularization parameter of graphical lasso will lead to a larger threshold, and the elements in the solution with their absolute values less than the threshold will be shrunk to zero. Therefore, the resultant solution of the graphical lasso will get sparser. The unexpected behavior of the $\ell_1$-norm characterized in Theorem 3.1 is due to the Laplacian constraints in the formulation (4). Because of the Laplacian constrains $\boldsymbol{\Theta} \cdot \mathbf{1}$ and $\Theta_{ij} = \Theta_{ji} \leq 0$ for any $i \neq j$, the term $\operatorname{tr}(\boldsymbol{\Theta S}) + \lambda \sum_{i>j} |\Theta_{ij}|$ in (4) can be written as $\sum_{i>j}(\lambda + S_{ii} + S_{jj} - S_{ij} - S_{ji})|\Theta_{ij}|$. To intuitively understand the behavior of the $\ell_1$-norm in (4), suppose that $\lambda$ is sufficiently large such that $\operatorname{tr}(\boldsymbol{\Theta S}) + \lambda \sum_{i>j} |\Theta_{ij}|$ can be approximated well by $\sum_{i>j} \lambda |\Theta_{ij}|$ well. Then (4) will be reduced to the optimization problem as below

$$\min_{\boldsymbol{\Theta} \in \mathcal{S}_L} -\log \det(\boldsymbol{\Theta} + \boldsymbol{J}) + \lambda \sum_{i>j} |\Theta_{ij}|. \tag{5}$$

Let $\tilde{\boldsymbol{\Theta}}$ be the optimal solution of (5). By calculation, we obtain that $\tilde{W}_{ij} = -\tilde{\Theta}_{ij} = \frac{2}{p\lambda}$ for any $i \neq j$. Notice that every graph weight $\tilde{W}_{ij}$ is strictly positive, and a large $\lambda$ will lead every weight to be very small, which are consistent with the empirical results in Figure 1.

It is worth mentioning that the optimization (4) can be approximated well by (5) only when the two constraints $\boldsymbol{\Theta} \cdot \mathbf{1}$ and $\Theta_{ij} = \Theta_{ji} \leq 0$ appear together. Therefore, the $\ell_1$-norm can still work well under the generalized Laplacian constrains [14, 39, 40], i.e., $\mathcal{S}_g = \left\{ \boldsymbol{\Theta} \in \mathcal{S}_{++}^p | \Theta_{ij} = \Theta_{ji} \leq 0,\ \forall i \neq j \right\}$. The set $\mathcal{S}_g$ is the convex cone of the symmetric $M$-matrices, and this graphical model with sign constraints is closely related to Gaussian distributions with multivariate total positivity [15, 28, 49, 50, 53].

## 3.2 Proposed Algorithm

Problem (3) is a constrained optimization problem with $\Theta \in \mathcal{S}_L$ including multiple constraints. We first simplify the Laplacian structural constraints in (1) by introducing a linear operator defined in [25] that maps a vector $\boldsymbol{x} \in \mathbb{R}^{p(p-1)/2}$ to a matrix $\mathcal{L}\boldsymbol{x} \in \mathbb{R}^{p \times p}$ as below.

**Definition 3.2.** The linear operator $\mathcal{L} : \mathbb{R}^{p(p-1)/2} \to \mathbb{R}^{p \times p}$, $\boldsymbol{x} \mapsto \mathcal{L}\boldsymbol{x}$, is defined by

$$[\mathcal{L}\boldsymbol{x}]_{ij} = \begin{cases} -x_k & i > j, \\ [\mathcal{L}\boldsymbol{x}]_{ji} & i < j, \\ -\sum_{j \neq i} [\mathcal{L}\boldsymbol{x}]_{ij} & i = j, \end{cases} \tag{6}$$

where $k = i - j + \frac{j-1}{2}(2p - j)$.

The adjoint operator $\mathcal{L}^*$ of $\mathcal{L}$ is defined so as to satisfy $\langle \mathcal{L}\boldsymbol{x}, \boldsymbol{Y} \rangle = \langle \boldsymbol{x}, \mathcal{L}^*\boldsymbol{Y} \rangle$, $\forall \boldsymbol{x} \in \mathbb{R}^{p(p-1)/2}$ and $\boldsymbol{Y} \in \mathbb{R}^{p \times p}$.

**Definition 3.3.** The adjoint operator $\mathcal{L}^* : \mathbb{R}^{p \times p} \to \mathbb{R}^{p(p-1)/2}$, $\boldsymbol{Y} \mapsto \mathcal{L}^*\boldsymbol{Y}$, is defined by

$$[\mathcal{L}^*\boldsymbol{Y}]_k = Y_{i,i} - Y_{i,j} - Y_{j,i} + Y_{j,j}, \tag{7}$$

where $i, j \in [p]$ obeying $k = i - j + \frac{j-1}{2}(2p - j)$ and $i > j$.

By introducing the linear operator $\mathcal{L}$, we can simplify the definition of $\mathcal{S}_L$ in (1) as below.

**Theorem 3.4.** *The Laplacian set $\mathcal{S}_L$ defined in* (1) *can be written as*

$$\mathcal{S}_L = \left\{ \mathcal{L}\boldsymbol{x} \mid \boldsymbol{x} \geq \boldsymbol{0}, \ (\mathcal{L}\boldsymbol{x} + \boldsymbol{J}) \in \mathcal{S}_{++}^p \right\}, \tag{8}$$

*where $\boldsymbol{J} = \frac{1}{p}\mathbf{1}_{p \times p}$ and $\boldsymbol{x} \geq \boldsymbol{0}$ means every entry of $\boldsymbol{x}$ is non-negative.*

As a result of Theorem 3.4, we introduce the linear operator $\mathcal{L}$ defined in (6) and reformulate the optimization (3) as

$$\min_{\boldsymbol{w} \geq \boldsymbol{0}} -\log \det(\mathcal{L}\boldsymbol{w} + \boldsymbol{J}) + \operatorname{tr}(\boldsymbol{S}\mathcal{L}\boldsymbol{w}) + \sum_i h_\lambda(w_i). \tag{9}$$

Notice that we remove the constraint $(\mathcal{L}\boldsymbol{x} + \boldsymbol{J}) \in \mathcal{S}_{++}^p$ in (9) compared with the constraint set in (8), because any $\boldsymbol{w}$ in the feasible set of (9) must obey $(\mathcal{L}\boldsymbol{x} + \boldsymbol{J}) \in \mathcal{S}_{++}^p$, following from the fact that $\mathcal{L}\boldsymbol{w} + \boldsymbol{J}$ must be positive semi-definite for any $\boldsymbol{w} \geq \boldsymbol{0}$.

To solve the problem (9), we follow the majorization-minimization framework [51], which consists of two steps. In the majorization step, we design a majorized function $f(\boldsymbol{w}|\hat{\boldsymbol{w}}^{(k-1)})$ that locally approximates the objective function $F(\boldsymbol{w})$ at $\hat{\boldsymbol{w}}^{(k-1)}$ satisfying

$$f(\boldsymbol{w}|\hat{\boldsymbol{w}}^{(k-1)}) \geq F(\boldsymbol{w}) \quad \text{and} \quad f(\hat{\boldsymbol{w}}^{(k-1)}|\hat{\boldsymbol{w}}^{(k-1)}) = F(\hat{\boldsymbol{w}}^{(k-1)}). \tag{10}$$

Then in the minimization step, we minimize the majorized function $f(\boldsymbol{w}|\hat{\boldsymbol{w}}^{(k-1)})$. We assume $h_\lambda$ is concave (refer to Assumption 3.5 for the choices of $h_\lambda$). Here we find $f(\boldsymbol{w}|\hat{\boldsymbol{w}}^{(k-1)})$ by linearizing $\sum_i h_\lambda(w_i)$. Set $f_k(\boldsymbol{w}) = f(\boldsymbol{w}|\hat{\boldsymbol{w}}^{(k-1)})$ to simplify the notation and obtain

$$f_k(\boldsymbol{w}) = -\log \det(\mathcal{L}\boldsymbol{w} + \boldsymbol{J}) + \operatorname{tr}(\boldsymbol{S}\mathcal{L}\boldsymbol{w}) + \sum_i h_\lambda'(\hat{w}_i^{(k-1)}) w_i, \tag{11}$$

By minimizing $f_k(\boldsymbol{w})$, we establish a sequence $\{\hat{\boldsymbol{w}}^{(k)}\}_{k \geq 1}$ by solving a sequence of sub-problems

$$\hat{\boldsymbol{w}}^{(k)} = \arg\min_{\boldsymbol{w} \geq \boldsymbol{0}} -\log \det(\mathcal{L}\boldsymbol{w} + \boldsymbol{J}) + \operatorname{tr}(\boldsymbol{S}\mathcal{L}\boldsymbol{w}) + \sum_i z_i^{(k-1)} w_i, \tag{12}$$

where $z_i^{(k-1)} = h_\lambda'\left(\hat{w}_i^{(k-1)}\right)$, $i \in [p(p-1)/2]$. We can see $\sum_i z_i^{(k-1)} w_i$ is equivalent to $\sum_i z_i^{(k-1)} |w_i|$ because $\boldsymbol{w} \geq \boldsymbol{0}$ and $z_i^{(k-1)} \geq 0$ by Assumption 3.5. Thus the problem (12) can be viewed as a weighted $\ell_1$-norm penalized maximum likelihood estimation under the Laplacian constrained Gaussian graphical model. The iteration procedure is summarized in Algorithm 1.

To solve the problem (12), we develop a projected gradient descent algorithm with backtracking line search. To obtain $\hat{\boldsymbol{w}}^{(k)}$, the algorithm starts with $\boldsymbol{w}_0^{(k)}$ and then establishes the sequence $\{\boldsymbol{w}_t^{(k)}\}_{t\geq 0}$ by the projected gradient descent as below. In the $t$-th iteration, we update $\boldsymbol{w}_t^{(k)}$ by

$$\boldsymbol{w}_t^{(k)} = \mathcal{P}_+\left(\boldsymbol{w}_{t-1}^{(k)} - \eta \nabla f_k(\boldsymbol{w}_{t-1}^{(k)})\right), \tag{13}$$

where $\mathcal{P}_+(\boldsymbol{a}) = \max(\boldsymbol{a}, \boldsymbol{0})$ and $\nabla f_k(\boldsymbol{w}_{t-1}^{(k)}) = -\mathcal{L}^*(\mathcal{L}\boldsymbol{w}_{t-1}^{(k)} + \boldsymbol{J})^{-1} + \mathcal{L}^* \boldsymbol{S} + \boldsymbol{z}^{(k-1)}$. The sequence $\{\boldsymbol{w}_t^{(k)}\}_{t\geq 0}$ will converge to $\hat{\boldsymbol{w}}^{(k)}$. Here we set $\boldsymbol{w}_0^{(k)} = \hat{\boldsymbol{w}}^{(k-1)}$ which is the limit point of the sequence $\{\boldsymbol{w}_t^{(k-1)}\}_{t\geq 0}$. To establish the theoretical results in Section 3.3, the initial point $\hat{\boldsymbol{w}}^{(0)}$ of Algorithm 1 is chosen such that $|\mathrm{supp}^+(\hat{\boldsymbol{w}}^{(0)})| \leq s$, where $s$ is the number of nonzero edges in the true graph. In addition, the proposed method can be extended to estimate other structured matrices such as Hankel matrices with the usage of Hankel linear operator [4, 57].

---

**Algorithm 1** Nonconvex Graph Learning (NGL)

---

**Input:** Sample covariance $\boldsymbol{S}$, $\lambda$, $\hat{\boldsymbol{w}}^{(0)}$;
    $k \leftarrow 1$;
1: **while** Stopping criteria not met **do**
2:    Update $z_i^{(k-1)} = h_\lambda'(\hat{w}_i^{(k-1)})$, for $i = 1, \ldots, p(p-1)/2$;
3:    Update $\hat{\boldsymbol{w}}^{(k)} = \arg\min_{\boldsymbol{w}\geq\boldsymbol{0}} -\log\det(\mathcal{L}\boldsymbol{w} + \boldsymbol{J}) + \mathrm{tr}(\boldsymbol{S}\mathcal{L}\boldsymbol{w}) + \sum_i z_i^{(k-1)} w_i$;
4:    $k \leftarrow k + 1$;
5: **end while**
**Output:** $\hat{\boldsymbol{w}}^{(k)}$.

---

### 3.3 Theoretical Results

Before we present the theoretical results, we first list the assumptions needed for establishing our theorems.

We denote the true graph weights by $\boldsymbol{w}^\star \in \mathbb{R}^{p(p-1)/2}$, which are non-negative, i.e., $\boldsymbol{w}^\star \geq \boldsymbol{0}$. Let $\mathcal{S}^\star = \{i \in [p(p-1)/2] \,|\, w_i^\star > 0\}$ be the support set of $\boldsymbol{w}^\star$ and $s$ be the number of the nonzero weights, i.e., $|\mathcal{S}^\star| = s$. Let $\lambda_{\max}(\mathcal{L}\boldsymbol{w}^\star)$ denote the maximum eigenvalue of $\mathcal{L}\boldsymbol{w}^\star$. We impose some mild conditions on the sparsity-promoting function $h_\lambda$ in Assumption 3.5 and the true graph weights $\boldsymbol{w}^\star$ in Assumption 3.6.

**Assumption 3.5.** The function $h_\lambda : \mathbb{R} \to \mathbb{R}$ satisfies the following conditions:

1. $h_\lambda(0) = 0$, and $h_\lambda'(x)$ is monotone and Lipschitz continuous for $x \in [0, +\infty)$;

2. There exists a $\gamma > 0$ such that $h_\lambda'(x) = 0$ for $x \geq \gamma\lambda$;

3. $h_\lambda'(x) = \lambda$ for $x \leq 0$ and $h_\lambda'(c\lambda) \geq \lambda/2$, where $c = (2 + \sqrt{2})\lambda_{\max}^2(\mathcal{L}\boldsymbol{w}^\star)$ is a constant.

**Assumption 3.6.** The graph weights $\boldsymbol{w}^\star$ represent a connected graph. The minimal nonzero graph weight satisfies $\min_{i\in\mathcal{S}^\star} w_i^\star \geq (c + \gamma)\lambda \gtrsim \lambda$, where $c$ and $\gamma$ are defined in Assumption 3.5.

**Remark 3.7.** In Assumption 3.5, the conditions on $h_\lambda(x)$ are mainly made over $x \in [0, +\infty)$ because of the nonnegativity of $\boldsymbol{w}$ in the optimization (12). In Assumption 3.5, the first two conditions are made to promote sparsity and unbiasedness [34], and hold for a variety of nonconvex sparsity-promoting functions including MCP [60] and SCAD [16]. In the third condition, we specify $h_\lambda'(x)$ for $x \leq 0$ only for theoretical analysis. The condition $h_\lambda'(c\lambda) \geq \lambda/2$ can always hold by tuning parameters due to the conditions $h_\lambda'(0) = \lambda$ and $h_\lambda'(\gamma\lambda) = 0$. In Assumption 3.6, the conditions on the true graph weights $\boldsymbol{w}^\star$ are mild. In our theorems, the regularization parameter $\lambda$ is taken with the order $\sqrt{\log p/n}$ that could be very small when the sample size $n$ increases. The assumptions on the minimal magnitude of signals are often employed in the analysis of nonconvex optimization [17, 52].

In the following theorem, the choice of the regularization parameter $\lambda$ is set according to a user-defined parameter $\alpha > 2$. A larger $\alpha$ yields a larger probability with which the claims hold, but also leads to a more stringent requirement on the number of samples.

**Theorem 3.8.** *Under Assumptions 3.5 and 3.6, take the regularization parameter* $\lambda = \sqrt{4\alpha c_0^{-1} \log p/n}$ *for some* $\alpha > 2$. *If the sample size* $n$ *is lower bounded by*

$$n \geq \max(94\alpha c_0^{-1} \lambda_{\max}^2(\mathcal{L}\boldsymbol{w}^\star)s \log p, 8\alpha \log p),$$

*then with probability at least* $1 - 1/p^{\alpha-2}$, *the sequence* $\hat{\boldsymbol{w}}^{(k)}$ *returned by Algorithm 1 satisfies*

$$\left\|\hat{\boldsymbol{w}}^{(k)} - \boldsymbol{w}^\star\right\| \leq \underbrace{2(3\sqrt{2}+4)\lambda_{\max}^2(\mathcal{L}\boldsymbol{w}^\star)\sqrt{\alpha c_0^{-1} s \log p/n}}_{\text{Statistical error}} + \underbrace{\left(\frac{3}{2+\sqrt{2}}\right)^k\left\|\hat{\boldsymbol{w}}^{(0)} - \boldsymbol{w}^\star\right\|}_{\text{Optimization error}},$$

*where* $c_0 = 1/\big(8\left\|\mathcal{L}^*(\mathcal{L}\boldsymbol{w}^\star + \boldsymbol{J})^{-1}\right\|_{\max}^2\big)$ *is a constant.*

The statement in Theorem 3.8 holds with overwhelming probability. Theorem 3.8 shows that the estimation error between the estimated and true graph weights is bounded by two terms, i.e., the optimization error and statistical error. The optimization error, $(\frac{3}{2+\sqrt{2}})^k\|\hat{\boldsymbol{w}}^{(0)} - \boldsymbol{w}^\star\|$, decays to zero at a linear rate with respect to the iteration number $k$. The statistical error is with the order of $\sqrt{s \log p/n}$, and a large sample size $n$ will lead to a small statistical error. We can see the statistical error is independent of $k$, implying that it will not decrease during iterations in the algorithm.

**Corollary 3.9.** *Under the same assumptions and conditions as stated in Theorem 3.8, the sequence* $\hat{\boldsymbol{w}}^{(k)}$ *returned by Algorithm 1 satisfies*

$$\left\|\mathcal{L}\hat{\boldsymbol{w}}^{(k)} - \mathcal{L}\boldsymbol{w}^\star\right\|_{\mathrm{F}} \leq \underbrace{4(2\sqrt{2}+3)\lambda_{\max}^2(\mathcal{L}\boldsymbol{w}^\star)\sqrt{\alpha c_0^{-1} s \log p/n}}_{\text{Statistical error}} + \underbrace{\left(\frac{3}{2+\sqrt{2}}\right)^k\left\|\mathcal{L}\hat{\boldsymbol{w}}^{(0)} - \mathcal{L}\boldsymbol{w}^\star\right\|_{\mathrm{F}}}_{\text{Optimization error}},$$

*with probability at least* $1 - 1/p^{\alpha-2}$. *Moreover, if* $k \geq \lceil 4\log(4\alpha c_0^{-1})\rceil$, *then the estimation error is dominated by the statistical error and we obtain*

$$\left\|\mathcal{L}\hat{\boldsymbol{w}}^{(k)} - \mathcal{L}\boldsymbol{w}^\star\right\|_{\mathrm{F}} \lesssim \sqrt{s \log p/n},$$

*where* $c_0 = 1/\big(8\left\|\mathcal{L}^*(\mathcal{L}\boldsymbol{w}^\star + \boldsymbol{J})^{-1}\right\|_{\max}^2\big)$ *is a constant.*

Corollary 3.9 presents the estimation error between the estimated and true precision matrices in the Laplacian constrained GGM. The order of statistical error is upper bounded by $\sqrt{s \log p/n}$, which matches the order of minimax lower bound [5, 42, 43] for the Gaussian graphical model. Yet it is still unknown if $\sqrt{s \log p/n}$ is the minimax rate of convergence for estimating sparse precision matrices under the Laplacian constrained GGM. Furthermore, the proposed estimation method can achieve the order of $\sqrt{s \log p/n}$ by solving only $\lceil 4\log(4\alpha c_0^{-1})\rceil$ sub-problems. Notice that $\lceil 4\log(4\alpha c_0^{-1})\rceil$ is independent of the dimension size $p$ and sample size $n$.

## 4 Experimental Results

In this section, we conduct numerical simulations on both synthetic data and real-world data to verify the performance of the proposed method. We use relative error (RE) and F-score (FS) to evaluate the performance of the algorithms, which are defined as

$$\mathsf{RE} = \frac{\left\|\hat{\boldsymbol{\Theta}} - \boldsymbol{\Theta}^\star\right\|_{\mathrm{F}}}{\left\|\boldsymbol{\Theta}^\star\right\|_{\mathrm{F}}}, \qquad \mathsf{FS} = \frac{2\mathsf{tp}}{2\mathsf{tp} + \mathsf{fp} + \mathsf{fn}}, \tag{14}$$

where $\hat{\boldsymbol{\Theta}} = \mathcal{L}\hat{\boldsymbol{w}}$ and $\boldsymbol{\Theta}^\star = \mathcal{L}\boldsymbol{w}^\star$ denote the estimated and true precision matrices, respectively. The true positive number is denoted as $\mathsf{tp}$, i.e., the case that there is an actual edge and the algorithm detects it, the false positive is denoted as $\mathsf{fp}$, i.e., the case that there is no actual edge but algorithm detects one, and the false negative is denoted as $\mathsf{fn}$, i.e., the case that the algorithm failed to detect an actual edge. The F-score takes values in $[0, 1]$, where $1$ indicates perfect structure recovery. For our algorithm, we test two nonconvex penalties, MCP and SCAD, defined respectively by

$$h'_{\mathsf{MCP},\lambda}(x) = \left\{\begin{array}{ll} \lambda - \frac{x}{\gamma} & x \in [0, \gamma\lambda], \\ 0 & x \in [\gamma\lambda, \infty), \end{array}\right. \quad h'_{\mathsf{SCAD},\lambda}(x) = \left\{\begin{array}{ll} \lambda & x \in [0, \lambda], \\ (\gamma\lambda - x)/\gamma - 1 & x \in [\lambda, \gamma\lambda], \\ 0 & x \in [\gamma\lambda, \infty), \end{array}\right.$$

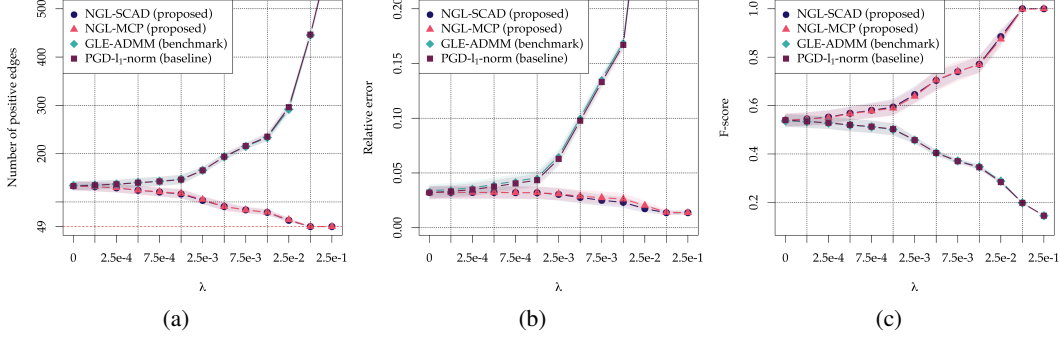

Figure 2: Performance measures (a) Number of positive edges, (b) Relative error and (c) F-score as a function of regularization parameter $\lambda$ in learning random Barabasi-Albert graphs. The true number of positive edges is 49 and the sample size ratio is $n/p = 100$.

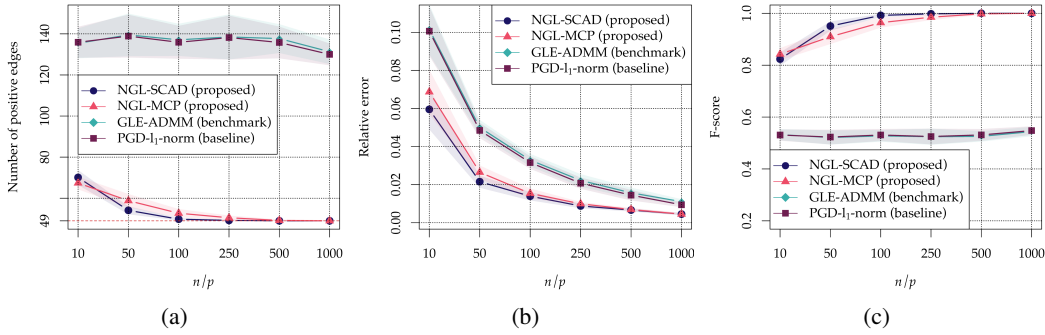

Figure 3: Performance measures (a) Number of positive edges, (b) Relative error and (c) F-score as a function of the sample size ratio of $n/p$ in learning random Barabasi-Albert graphs. The true number of positive edges is 49. The regularization parameter $\lambda$ for each algorithm is fine-tuned.

where we only define $h'_{\mathsf{MCP},\lambda}(x)$ and $h'_{\mathsf{SCAD},\lambda}(x)$ for $x \geq 0$ because of the nonnegativity constraint in (12). We set $\gamma$ equal to $1.01$ in $h'_{\mathsf{MCP},\lambda}(x)$ and $2.01$ in $h'_{\mathsf{SCAD},\lambda}(x)$ for all the experiments. The compared methods include the state-of-the-art GLE-ADMM algorithm [62] and the baseline projected gradient descent with the $\ell_1$-norm.

## 4.1 Synthetic Data

We generate the data matrix $\boldsymbol{X} \in \mathbb{R}^{p \times n}$ with each column of $\boldsymbol{X}$ independently sampled from the LGMRF with $\boldsymbol{\Theta} = \mathcal{L}\boldsymbol{w}^\star$, where $\boldsymbol{w}^\star$ is the true weights from a random Barabasi-Albert graph of degree 1. The number of nodes in the Barabasi-Albert graph is $p = 50$, and the weights associated with edges are uniformly sampled from $U(2,5)$. The sample covariance matrix is constructed by $\boldsymbol{S} = \frac{1}{n}\boldsymbol{X}\boldsymbol{X}^\top$, where $n$ is the number of samples. The curves in Figures 2 and 3 are the results of an average of 100 Monte Carlo realizations and the shaded areas around the curves represent the one-standard deviation confidence interval.

Figure 2 presents the results of the random Barabasi-Albert graphs learned by GLE-ADMM [62], the projected gradient descent with $\ell_1$-norm and the proposed method with the two regularizers. It is observed that both GLE-ADMM and the baseline projected gradient descent with $\ell_1$-norm achieve the best performance in terms of sparsity, relative error and F-score when $\lambda = 0$, which defies the purpose of introducing the $\ell_1$-norm regularizer. In contrast, the proposed methods enhance sparsity when increasing $\lambda$. It is observed that, with $\lambda$ equal to $0.1$ or $0.25$, the proposed methods achieve the true number of nonzero edges, and an F-score of $1$, implying that all the zero and nonzero edges of the true graph are correctly identified by the proposed methods.

Figure 3 shows that the proposed methods always outperform both GLE-ADMM and the baseline projected gradient descent with $\ell_1$-norm in terms of sparsity, relative error, and F-score under different samples size ratios.

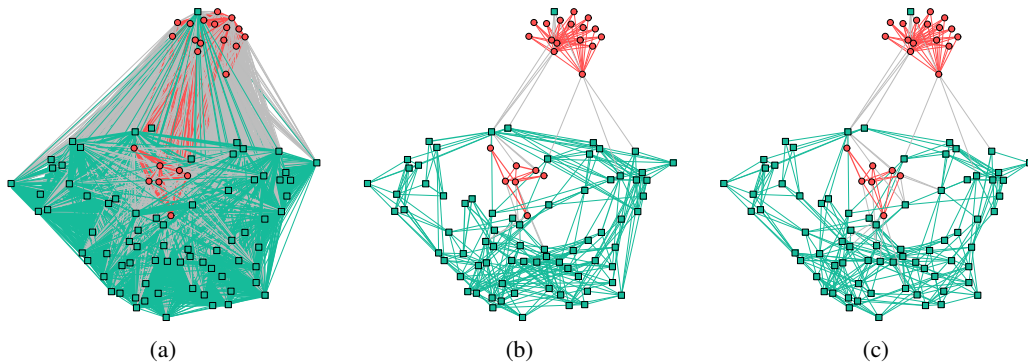

<center>(a)             (b)             (c)</center>

Figure 4: The learned graphs using the 2019-nCoV data set by (a) GLE-ADMM, (b) NGL-SCAD (proposed method), and (c) NGL-MCP (proposed method). The computational time for GLE-ADMM, NGL-SCAD and NGL-MCP are 2.9, 0.7 and 0.8 seconds, respectively, conducted on a PC with a 2.8 GHz Inter Core i7 CPU and 16 GB RAM. The regularization parameter is set as $\lambda_{\mathsf{ADMM}} = 0$, $\lambda_{\mathsf{SCAD}} = 0.6$, and $\lambda_{\mathsf{MCP}} = 1.2$.

## 4.2 Real-world Data

We conduct numerical experiments on the 2019-nCoV data set[1] from 98 anonymous Chinese patients affected by the outbreak of 2019-nCoV on early February, 2020. The features include age (integer), gender (categorical), and location (categorical). The label is a binary variable representing the life status of patients, alive (green) or no longer alive (red). Our goal is to construct a graph from the data features. To this end, we first pre-process the feature matrix so as to transform the categorical features into numerical ones via one-hot-encoding, and obtain a feature matrix $\boldsymbol{X}$ with the dimension $98 \times 32$, i.e., $p = 98$ and $n = 32$. We then compute the sample covariance matrix and learn the graphs.

Figure 4 shows that the benchmark GLE-ADMM[2] is unable to impose sparsity, diminishing interpretation capabilities of the graph severely. On the other hand, the proposed methods NGL-SCAD and NGL-MCP obtain a sparse graph with clearer connections. Note that the 2019-nCoV data consists of two groups, red and green nodes. It is natural to assume that the nodes belonging to different groups are dissimilar from each other, while the nodes in the same group are similar. In this sense, the performance of our learned graphs in Figure 4 are significant, because most connections are between nodes within the same group, and only a few connections (gray edges) are between nodes from distinct groups. The learned graphs possibly provide guidance on priority setting in health care because green nodes (patients alive) that have stronger connections with red nodes (patients that passed away) may suffer a higher health risk.

It is worth mentioning that the real-world data sets may not exactly follow the Laplacian constrained GGM. In the case of the data following Laplacian constrained GGM, the formulation (3) can be viewed as a regularized maximum likelihood estimation of the precision matrix. In a more general setting with non-Gaussian distribution, (3) can be related to the regularized log-determinant Bregman divergence optimization, and the learned graph weights can quantify the similarity between nodes. This is because the trace term in (3) can be written as Laplacian quadratic [12, 22, 24], which tends to assign a large weight between nodes if their signal values are similar.

## 5 Conclusions

In this paper, we have considered learning a sparse graph under the Laplacian constrained Gaussian graphical model. We have proved that a large regularization parameter of the $\ell_1$-norm leads to learning a complete graph. Then we have proposed a nonconvex penalized maximum likelihood method by solving a sequence of weighted $\ell_1$-norm regularized sub-problems, and have established the order of the statistical error. A projected gradient descent algorithm has been designed to solve the sub-problems. Numerical results involving synthetic and real-world data sets demonstrate the effectiveness of the proposed method.

## Broader Impact

This paper provides an unexpected behavior of the $\ell_1$ norm in learning sparse graphs, which may greatly benefit the community of signal processing and machine learning over graphs. This paper further provides a solution to solve the issue with theoretical guarantees.

## Acknowledgments

We would like to thank the anonymous reviewers for their helpful comments. This work was supported by the Hong Kong GRF 16207019 research grant.

## Footnotes

[1] 2019-nCoV data is available in a queryable format via the R package nCov2019 which lives on GitHub: https://github.com/GuangchuangYu/nCov2019.

[2] The code for GLE-ADMM lives at https://github.com/dppalomar/spectralGraphTopology.

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
