[Supplementary Material]

# Supplementary Material: Nonconvex Sparse Graph Learning under Laplacian Constrained Graphical Model

Jiaxi Ying [*]          José Vinícius de M. Cardoso [†]          Daniel P. Palomar [‡]

**Abstract**

In this supplementary material, Section 1 provides the proofs of Theorems 3.1, 3.4, 3.8, and Corollary 3.9. The proofs of technical lemmas are collected in Section 2. Section 3 presents the additional experimental results.

## 1   Proofs of Theorems

This section contains the proofs of Theorems 3.1, 3.4, 3.8, and Corollary 3.9. Before starting to prove the theorems, we first present some definitions and technical lemmas.

Recall that the optimization problem we aim to solve is

$$\min_{\boldsymbol{w} \geq \boldsymbol{0}} -\log \det(\mathcal{L}\boldsymbol{w} + \boldsymbol{J}) + \operatorname{tr}(\boldsymbol{S}\mathcal{L}\boldsymbol{w}) + \sum_i h_\lambda(w_i). \tag{1}$$

Let $F(\boldsymbol{w}) = -\log \det(\mathcal{L}\boldsymbol{w} + \boldsymbol{J}) + \operatorname{tr}(\boldsymbol{S}\mathcal{L}\boldsymbol{w}) + \sum_i h_\lambda(w_i)$. The feasible set of (1) is $\mathcal{S}_{\boldsymbol{w}} = \{\boldsymbol{w} \,|\, \boldsymbol{w} \geq \boldsymbol{0}, \boldsymbol{w} \in \operatorname{dom}(F)\}$, where $\operatorname{dom}(F)$ denotes the domain of the function $F$. One can verify that

$$\operatorname{dom}(F) = \{\boldsymbol{w} \in \mathbb{R}^{p(p-1)/2} \,|\, \det(\mathcal{L}\boldsymbol{w} + \boldsymbol{J}) > 0\}. \tag{2}$$

The set $\mathcal{S}_{\boldsymbol{w}}$ can be equivalently written as

$$\mathcal{S}_{\boldsymbol{w}} = \{\boldsymbol{w} \in \mathbb{R}^{p(p-1)/2} \,|\, \boldsymbol{w} \geq \boldsymbol{0}, (\mathcal{L}\boldsymbol{w} + \boldsymbol{J}) \in \mathcal{S}_{++}^p\}, \tag{3}$$

which is due to the reason that $\mathcal{L}\boldsymbol{w} + \boldsymbol{J}$ must be positive semi-definite, because $\mathcal{L}\boldsymbol{w}$ is positive semi-definite for any $\boldsymbol{w} \geq \boldsymbol{0}$ following from (27), while the matrix $\boldsymbol{J}$ is rank one, and the nonzero eigenvalue of $\boldsymbol{J}$ is 1 whose eigenvector is orthogonal to the row and column spaces of $\mathcal{L}\boldsymbol{w}$. The condition $\det(\mathcal{L}\boldsymbol{w} + \boldsymbol{J}) > 0$ in (2) implies that $\mathcal{L}\boldsymbol{w} + \boldsymbol{J}$ is non-singular. The non-singularity and positive semi-definiteness of $\mathcal{L}\boldsymbol{w} + \boldsymbol{J}$ together lead to $(\mathcal{L}\boldsymbol{w} + \boldsymbol{J}) \in \mathcal{S}_{++}^p$.

Next, we will show that $\mathcal{S}_{\boldsymbol{w}}$ is a convex set. For any $\boldsymbol{x}_1, \boldsymbol{x}_2 \in \mathcal{S}_{\boldsymbol{w}}$, define $\boldsymbol{x}_t = t\boldsymbol{x}_1 + (1-t)\boldsymbol{x}_2$, $t \in [0,1]$. It is clear that $\boldsymbol{x}_t \geq \boldsymbol{0}$. Since $\mathcal{S}_{++}^p$ is a convex cone, one has

$$\mathcal{L}\boldsymbol{x}_t + \boldsymbol{J} = t(\mathcal{L}\boldsymbol{x}_1 + \boldsymbol{J}) + (1-t)(\mathcal{L}\boldsymbol{x}_2 + \boldsymbol{J}) \in \mathcal{S}_{++}^p, \tag{4}$$

indicating that $\boldsymbol{x}_t \in \mathcal{S}_{\boldsymbol{w}}$ and thus $\mathcal{S}_{\boldsymbol{w}}$ is convex.

[*]Department of Electronic and Computer Engineering, The Hong Kong University of Science and Technology, Clear Water Bay, Hong Kong SAR; Email: `jx.ying@connect.ust.hk`.

[†]Department of Electronic and Computer Engineering, The Hong Kong University of Science and Technology, Clear Water Bay, Hong Kong SAR; Email: `jvdmc@connect.ust.hk`.

[‡]Department of Electronic and Computer Engineering, Department of Industrial Engineering and Data Analytics, The Hong Kong University of Science and Technology, Clear Water Bay, Hong Kong SAR; Email: `palomar@ust.hk`.

Recall that the sequence $\{\hat{\boldsymbol{w}}^{(k)}\}_{k\geq 1}$ is established by solving

$$\hat{\boldsymbol{w}}^{(k)} = \arg\min_{\boldsymbol{w}\geq\boldsymbol{0}} -\log\det(\mathcal{L}\boldsymbol{w}+\boldsymbol{J}) + \operatorname{tr}\left(\boldsymbol{S}\mathcal{L}\boldsymbol{w}\right) + \sum_i z_i^{(k-1)} w_i, \tag{5}$$

where $z_i^{(k-1)} = h'_\lambda(\hat{w}_i^{(k-1)})$, $i \in [p(p-1)/2]$, and $h_\lambda$ satisfies the conditions in Assumption 3.5. Let $\|\boldsymbol{x}\|_{\max} = \max_i |x_i|$ and $\|\boldsymbol{x}\|_{\min} = \min_i |x_i|$. We present the technical lemmas in Section 1.1 for establishing the theorems.

## 1.1 Technical Lemmas

**Lemma 1.1.** *Let* $f(\boldsymbol{w}) = -\log\det(\mathcal{L}\boldsymbol{w}+\boldsymbol{J})$, *with* $\boldsymbol{w} \in \mathbb{R}^{p(p-1)/2}$. *Then for any* $\boldsymbol{x} \in \mathbb{R}^{p(p-1)/2}$, *we have*

$$\boldsymbol{x}^\top \nabla^2 f(\boldsymbol{w})\boldsymbol{x} = vec(\mathcal{L}\boldsymbol{x})^\top \left((\mathcal{L}\boldsymbol{w}+\boldsymbol{J})^{-1} \otimes (\mathcal{L}\boldsymbol{w}+\boldsymbol{J})^{-1}\right) vec(\mathcal{L}\boldsymbol{x}).$$

**Lemma 1.2.** *For any given* $\boldsymbol{w} \in \mathbb{R}^{p(p-1)/2}$ *satisfying* $(\mathcal{L}\boldsymbol{w}+\boldsymbol{J}) \in \mathcal{S}_{++}^p$, *there must exist an unique* $\boldsymbol{x} \in \mathbb{R}^{p(p-1)/2}$ *such that*

$$\mathcal{L}\boldsymbol{x} + \frac{1}{b}\boldsymbol{J} = (\mathcal{L}\boldsymbol{w}+b\boldsymbol{J})^{-1} \tag{6}$$

*holds for any* $b \neq 0$, *where* $\boldsymbol{J} = \frac{1}{p}\boldsymbol{1}_{p\times p}$, *in which* $\boldsymbol{1}_{p\times p} \in \mathbb{R}^{p\times p}$ *with each element equals 1.*

**Lemma 1.3.** *Let* $\mathcal{G} = \mathcal{L}^*\mathcal{L} : \mathbb{R}^{p(p-1)/2} \to \mathbb{R}^{p(p-1)/2}$, $\boldsymbol{x} \mapsto \mathcal{L}^*\mathcal{L}\boldsymbol{x}$. *For any* $\boldsymbol{x} \in \mathbb{R}^{p(p-1)/2}$, $\mathcal{G}\boldsymbol{x} = \boldsymbol{M}\boldsymbol{x}$ *with* $\boldsymbol{M} \in \mathbb{R}^{\frac{p(p-1)}{2}\times\frac{p(p-1)}{2}}$ *satisfying*

$$M_{kl} = \begin{cases} 4 & l = k, \\ 1 & l \in (\Omega_i \cup \Omega_j)\backslash k, \\ 0 & Otherwise, \end{cases}$$

*where* $i, j \in [p]$ *satisfying* $k = i - j + \frac{j-1}{2}(2p - j)$ *and* $i > j$, *and* $\Omega_t$ *is an index set defined by*

$$\Omega_t := \left\{ l \in [p(p-1)/2] \,|\, [\mathcal{L}\boldsymbol{x}]_{tt} = \sum_l x_l \right\}, \quad t \in [p].$$

*Furthermore, we have* $\lambda_{\min}(\boldsymbol{M}) = 2$ *and* $\lambda_{\max}(\boldsymbol{M}) = 2p$.

**Lemma 1.4.** *Take* $\lambda = \sqrt{4\alpha c_0^{-1} \log p/n}$ *and suppose* $n \geq 94\alpha c_0^{-1}\lambda_{\max}^2(\mathcal{L}\boldsymbol{w}^\star)s\log p$ *for some* $\alpha > 2$, *where* $c_0$ *is a constant defined in Lemma 1.8. Let*

$$\hat{\boldsymbol{w}} = \arg\min_{\boldsymbol{w}\geq\boldsymbol{0}} -\log\det(\mathcal{L}\boldsymbol{w}+\boldsymbol{J}) + \operatorname{tr}(\mathcal{L}\boldsymbol{w}\boldsymbol{S}) + \boldsymbol{z}^\top\boldsymbol{w},$$

*where* $\boldsymbol{z}$ *obeys* $0 \leq z_i \leq \lambda$ *for* $i \in [p(p-1)/2]$. *If* $\left\|\mathcal{L}^*\left((\mathcal{L}\boldsymbol{w}^\star+\boldsymbol{J})^{-1} - \boldsymbol{S}\right)\right\|_{\max} \leq \lambda/2 \leq \|\boldsymbol{z}_{\mathcal{E}^c}\|_{\min}$ *holds with the set* $\mathcal{E}$ *satisfying* $\mathcal{S}^\star \subseteq \mathcal{E}$ *and* $|\mathcal{E}| \leq 2s$, *then* $\hat{\boldsymbol{w}}$ *obeys*

$$\|\mathcal{L}\hat{\boldsymbol{w}} - \mathcal{L}\boldsymbol{w}^\star\|_{\mathrm{F}} \leq 2\sqrt{2}\lambda_{\max}^2(\mathcal{L}\boldsymbol{w}^\star)\left(\|\boldsymbol{z}_{\mathcal{S}^\star}\| + \left\|\left(\mathcal{L}^*\left((\mathcal{L}\boldsymbol{w}^\star+\boldsymbol{J})^{-1} - \boldsymbol{S}\right)\right)_{\mathcal{E}}\right\|\right) \leq 2(1+\sqrt{2})\lambda_{\max}^2(\mathcal{L}\boldsymbol{w}^\star)\sqrt{s}\lambda,$$

*where* $\mathcal{S}^\star$ *is the support of* $\boldsymbol{w}^\star$.

**Lemma 1.5.** *Under Assumption 3.5, take $\lambda = \sqrt{4\alpha c_0^{-1}\log p/n}$ and suppose $n \geq 94\alpha c_0^{-1}\lambda_{\max}^2(\mathcal{L}\boldsymbol{w}^\star)s\log p$ for some $\alpha > 2$, where $c_0$ is a constant defined in Lemma 1.8. Define the set $\mathcal{E}^{(k)}$ by*

$$\mathcal{E}^{(k)} = \{\mathcal{S}^\star \cup \mathcal{S}^{(k)}\}, \quad with \quad \mathcal{S}^{(k)} = \{i \in [p(p-1)/2] \,|\, \hat{w}_i^{(k-1)} \geq b\}, \tag{7}$$

*where $\hat{\boldsymbol{w}}^{(k)}$ for $k \geq 1$ is defined in (5), $\mathcal{S}^\star$ is the support of $\boldsymbol{w}^\star$ with $|\mathcal{S}^\star| \leq s$ and $b = (2+\sqrt{2})\lambda_{\max}^2(\mathcal{L}\boldsymbol{w}^\star)\lambda$ is a constant. If $\left\|\mathcal{L}^*\big((\mathcal{L}\boldsymbol{w}^\star + \boldsymbol{J})^{-1} - \boldsymbol{S}\big)\right\|_{\max} \leq \lambda/2$ holds and $\hat{\boldsymbol{w}}^{(0)}$ satisfies $|\mathrm{supp}^+(\hat{\boldsymbol{w}}^{(0)})| \leq s$, then $\mathcal{E}^{(k)}$ obeys $|\mathcal{E}^{(k)}| \leq 2s$, for any $k \geq 1$.*

**Lemma 1.6.** *Under Assumptions 3.5 and 3.6, take $\lambda = \sqrt{4\alpha c_0^{-1}\log p/n}$ and suppose $n \geq 94\alpha c_0^{-1}\lambda_{\max}^2(\mathcal{L}\boldsymbol{w}^\star)s\log p$ for some $\alpha > 2$, where $c_0$ is a constant defined in Lemma 1.8. If $\left\|\mathcal{L}^*\big((\mathcal{L}\boldsymbol{w}^\star + \boldsymbol{J})^{-1} - \boldsymbol{S}\big)\right\|_{\max} \leq \lambda/2$ holds and $\hat{\boldsymbol{w}}^{(0)}$ satisfies $|\mathrm{supp}^+(\hat{\boldsymbol{w}}^{(0)})| \leq s$, then for any $k \geq 1$, $\hat{\boldsymbol{w}}^{(k)}$ defined in (5) obeys*

$$\left\|\hat{\boldsymbol{w}}^{(k)} - \boldsymbol{w}^\star\right\| \leq 2\lambda_{\max}^2(\mathcal{L}\boldsymbol{w}^\star)\left\|\big(\mathcal{L}^*\big((\mathcal{L}\boldsymbol{w}^\star + \boldsymbol{J})^{-1} - \boldsymbol{S}\big)\big)_{\mathcal{S}^\star}\right\| + \frac{3}{2+\sqrt{2}}\left\|\hat{\boldsymbol{w}}^{(k-1)} - \boldsymbol{w}^\star\right\|,$$

*and*

$$\left\|\mathcal{L}\hat{\boldsymbol{w}}^{(k)} - \mathcal{L}\boldsymbol{w}^\star\right\|_{\mathrm{F}} \leq 2\sqrt{2}\lambda_{\max}^2(\mathcal{L}\boldsymbol{w}^\star)\left\|\big(\mathcal{L}^*\big((\mathcal{L}\boldsymbol{w}^\star + \boldsymbol{J})^{-1} - \boldsymbol{S}\big)\big)_{\mathcal{S}^\star}\right\| + \frac{3}{2+\sqrt{2}}\left\|\mathcal{L}\hat{\boldsymbol{w}}^{(k-1)} - \mathcal{L}\boldsymbol{w}^\star\right\|_{\mathrm{F}}.$$

**Lemma 1.7.** *Take $\lambda = \sqrt{4\alpha c_0^{-1}\log p/n}$ and suppose $n \geq 8\alpha\log p$ for some $\alpha > 2$, where $c_0$ is a constant defined in Lemma 1.8. Then one has*

$$\mathbb{P}\big[\left\|\mathcal{L}^*\big((\mathcal{L}\boldsymbol{w}^\star + \boldsymbol{J})^{-1} - \boldsymbol{S}\big)\right\|_{\max} \leq \lambda/2\big] \geq 1 - 1/p^{\alpha-2}.$$

**Lemma 1.8.** *Consider a zero-mean random vector $\boldsymbol{X} = [X_1, \ldots, X_p]^\top \in \mathbb{R}^p$ is a LGMRF with precision matrix $\mathcal{L}\boldsymbol{w}^\star \in \mathcal{S}_L$. Given $n$ i.i.d samples $\boldsymbol{X}^{(1)}, \ldots, \boldsymbol{X}^{(n)}$, the associated sample covariance matrix $\boldsymbol{S} = \frac{1}{n}\sum_{k=1}^n \boldsymbol{X}^{(k)}\boldsymbol{X}^{(k)T}$ satisfies, for $t \in [0, t_0]$,*

$$\mathbb{P}\big[\,|[\mathcal{L}^*\boldsymbol{S}]_i - \big(\mathcal{L}^*(\mathcal{L}\boldsymbol{w}^\star + \boldsymbol{J})^{-1}\big)_i| \geq t\big] \leq 2\exp(-c_0 n t^2), \quad for \ \ i = 1, \ldots, p(p-1)/2,$$

*where $t_0 = \left\|\mathcal{L}^*(\mathcal{L}\boldsymbol{w}^\star + \boldsymbol{J})^{-1}\right\|_{\max}$ and $c_0 = 1/\big(8\left\|\mathcal{L}^*(\mathcal{L}\boldsymbol{w}^\star + \boldsymbol{J})^{-1}\right\|_{\max}^2\big)$ are two constants.*

**Lemma 1.9.** *Let $f(\boldsymbol{w}) = -\log\det(\mathcal{L}\boldsymbol{w} + \boldsymbol{J})$. Define a local region of $\boldsymbol{w}^\star$ by*

$$\mathcal{B}_{\boldsymbol{M}}(\boldsymbol{w}^\star; r) = \{\boldsymbol{w}|\boldsymbol{w} \in \mathbb{B}_{\boldsymbol{M}}(\boldsymbol{w}^\star; r) \cap \mathcal{S}_{\boldsymbol{w}}\}.$$

*where $\mathbb{B}_{\boldsymbol{M}}(\boldsymbol{w}^\star; r) = \{\boldsymbol{w} \in \mathbb{R}^{p(p-1)/2} \,|\, \|\boldsymbol{w} - \boldsymbol{w}^\star\|_{\boldsymbol{M}} \leq r\}$, and $\mathcal{S}_{\boldsymbol{w}} = \{\boldsymbol{w}\,|\,\boldsymbol{w} \geq \boldsymbol{0}, (\mathcal{L}\boldsymbol{w} + \boldsymbol{J}) \in \mathcal{S}_{++}^p\}$. Then for any $\boldsymbol{w}_1, \boldsymbol{w}_2 \in \mathcal{B}_{\boldsymbol{M}}(\boldsymbol{w}^\star; r)$, we have*

$$\langle \nabla f(\boldsymbol{w}_1) - \nabla f(\boldsymbol{w}_2), \boldsymbol{w}_1 - \boldsymbol{w}_2 \rangle \geq (\|\mathcal{L}\boldsymbol{w}^\star\|_2 + r)^{-2}\|\mathcal{L}\boldsymbol{w}_1 - \mathcal{L}\boldsymbol{w}_2\|_{\mathrm{F}}^2.$$

**Lemma 1.10.** *[7] Suppose a positive matrix $\boldsymbol{A} \in \mathbb{R}^{p \times p}$ is diagonally scaled such that $A_{ii} = 1$, $i = 1, \ldots, p$, and $0 < A_{ij} < 1$, $i \neq j$. Let $y$ and $x$ be the lower and upper bounds satisfying*

$$0 < y \leq A_{ij} \leq x < 1, \quad \forall\, i \neq j,$$

*and define $s$ by*

$$x^2 = sy + (1-s)y^2.$$

*Then the inverse matrix of $\boldsymbol{A}$ exists and $\boldsymbol{A}$ is an inverse $M$-matrix if $s^{-1} \geq p - 2$ with $p > 3$.*

**Lemma 1.11.** *[6] (Sub-exponential tail bound) Suppose $X$ is sub-exponential with parameters $(\upsilon, \alpha)$. Then*

$$\mathbb{P}[X - \mu \geq t] \leq \begin{cases} e^{-\frac{t^2}{2\upsilon^2}} & t \in [0, \frac{\upsilon^2}{\alpha}], \\ e^{-\frac{t}{2\alpha}} & t \in (\frac{\upsilon^2}{\alpha}, +\infty). \end{cases}$$

## 1.2  Proof of Theorem 3.1

*Proof.* The $\ell_1$-norm regularized maximum likelihood estimation under the Laplacian constrained Gaussian graphical model can be formulated as

$$\min_{\boldsymbol{\Theta} \in \mathcal{S}_L} - \log \det(\boldsymbol{\Theta} + \boldsymbol{J}) + \operatorname{tr}(\boldsymbol{\Theta S}) + \lambda \sum_{i>j} |\Theta_{ij}|, \tag{8}$$

where $\boldsymbol{S}$ is the sample covariance matrix with the samples generated from LGMRF. As a result of Theorem 3.4, the optimization (8) can be equivalently written as

$$\min_{\boldsymbol{w} \geq 0} - \log \det(\mathcal{L}\boldsymbol{w} + \boldsymbol{J}) + \operatorname{tr}(\mathcal{L}\boldsymbol{w S}) + \lambda \|\boldsymbol{w}\|_1. \tag{9}$$

Due to the non-negativity constraint $\boldsymbol{w} \geq \boldsymbol{0}$, (9) can be further rewritten as

$$\min_{\boldsymbol{w} \geq 0} - \log \det(\mathcal{L}\boldsymbol{w} + \boldsymbol{J}) + \langle \mathcal{L}^* \boldsymbol{S} + \lambda \boldsymbol{1}, \boldsymbol{w} \rangle, \tag{10}$$

where $\boldsymbol{1} = [1, \ldots, 1]^\top$.

We first prove that the optimization (10) has one global minimizer if $\lambda > 0$. Let $f(\boldsymbol{w}) = - \log \det(\mathcal{L}\boldsymbol{w} + \boldsymbol{J}) + \langle \mathcal{L}^* \boldsymbol{S} + \lambda \boldsymbol{1}, \boldsymbol{w} \rangle$. The feasible set of (10) is $\mathcal{S}_{\boldsymbol{w}} = \{\boldsymbol{w} \in \mathbb{R}^{p(p-1)/2} \,|\, \boldsymbol{w} \geq \boldsymbol{0}, (\mathcal{L}\boldsymbol{w} + \boldsymbol{J}) \in \mathcal{S}_{++}^p\}$, which is the same with the feasible set of (1). For any $\boldsymbol{w} \in \mathcal{S}_{\boldsymbol{w}}$, the minimum eigenvalue of $\nabla^2 f(\boldsymbol{w})$ can be lower bounded by

$$
\begin{aligned}
\lambda_{\min}\left(\nabla^2 f(\boldsymbol{w})\right) &= \inf_{\|\boldsymbol{x}\|=1} \boldsymbol{x}^\top \nabla^2 f(\boldsymbol{w}) \boldsymbol{x} \\
&= \inf_{\|\boldsymbol{x}\|=1} (\operatorname{vec}(\mathcal{L}\boldsymbol{x}))^\top \left((\mathcal{L}\boldsymbol{w} + \boldsymbol{J})^{-1} \otimes (\mathcal{L}\boldsymbol{w} + \boldsymbol{J})^{-1}\right) \operatorname{vec}(\mathcal{L}\boldsymbol{x}) \\
&\geq \inf_{\|\boldsymbol{x}\|=1} \frac{(\operatorname{vec}(\mathcal{L}\boldsymbol{x}))^\top \left((\mathcal{L}\boldsymbol{w} + \boldsymbol{J})^{-1} \otimes (\mathcal{L}\boldsymbol{w} + \boldsymbol{J})^{-1}\right) \operatorname{vec}(\mathcal{L}\boldsymbol{x})}{(\operatorname{vec}(\mathcal{L}\boldsymbol{x}))^\top \operatorname{vec}(\mathcal{L}\boldsymbol{x})} \cdot \inf_{\|\boldsymbol{x}\|=1} \|\mathcal{L}\boldsymbol{x}\|_{\mathrm{F}}^2 \\
&\geq \lambda_{\min}\left((\mathcal{L}\boldsymbol{w} + \boldsymbol{J})^{-1} \otimes (\mathcal{L}\boldsymbol{w} + \boldsymbol{J})^{-1}\right) \cdot \inf_{\|\boldsymbol{x}\|=1} \|\mathcal{L}\boldsymbol{x}\|_{\mathrm{F}}^2 \\
&= \lambda_{\min}\left((\mathcal{L}\boldsymbol{w} + \boldsymbol{J})^{-1} \otimes (\mathcal{L}\boldsymbol{w} + \boldsymbol{J})^{-1}\right) \cdot \inf_{\|\boldsymbol{x}\|=1} \boldsymbol{x}^\top \boldsymbol{M} \boldsymbol{x} \\
&= 2\lambda_{\min}\left((\mathcal{L}\boldsymbol{w} + \boldsymbol{J})^{-1}\right)^2 \\
&> 0,
\end{aligned}
$$

where the second equality is due to Lemma 1.1; the third equality follows from Lemma 1.3; the last equality follows from the property of Kronecker product that the eigenvalues of $\boldsymbol{A} \otimes \boldsymbol{B}$ are $\lambda_i \mu_j$ for $i, j \in [p]$, where $\lambda_i$ and $\mu_j$ are the eigenvalues of $\boldsymbol{A} \in \mathbb{R}^{p \times p}$ and $\boldsymbol{B} \in \mathbb{R}^{p \times p}$, respectively, and $\lambda_{\min}(\boldsymbol{M}) = 2$ following from Lemma 1.3; the last inequality follows from (3). Therefore, the optimization (10) is strictly convex, and thus (10) has at most one global minimizer.

The existence of minimizers of (10) can be guaranteed by the coercivity of $f(\boldsymbol{w})$. The function $f(\boldsymbol{w})$ can be lower bounded by

$$
\begin{aligned}
f(\boldsymbol{w}) &= - \log\left(\prod_{i=1}^{p} \lambda_i(\mathcal{L}\boldsymbol{w} + \boldsymbol{J})\right) + \langle \mathcal{L}^* \boldsymbol{S} + \lambda \boldsymbol{1}, \boldsymbol{w} \rangle \\
&= - \log\left(\prod_{i=2}^{p} \lambda_i(\mathcal{L}\boldsymbol{w})\right) + \langle \mathcal{L}^* \boldsymbol{S} + \lambda \boldsymbol{1}, \boldsymbol{w} \rangle \\
&\geq -(p-1) \log\left(\sum_{i=1}^{p} \lambda_i(\mathcal{L}\boldsymbol{w})\right) + \langle \mathcal{L}^* \boldsymbol{S} + \lambda \boldsymbol{1}, \boldsymbol{w} \rangle + (p-1) \log(p-1)
\end{aligned}
$$

$$= -(p-1)\log\left(\sum_{i=1}^{p}[\mathcal{L}\boldsymbol{w}]_{ii}\right) + \langle\mathcal{L}^*\boldsymbol{S} + \lambda\mathbf{1}, \boldsymbol{w}\rangle + (p-1)\log(p-1)$$

$$= -(p-1)\log\left(2\sum_{t=1}^{p(p-1)/2} w_t\right) + \langle\mathcal{L}^*\boldsymbol{S} + \lambda\mathbf{1}, \boldsymbol{w}\rangle + (p-1)\log(p-1)$$

$$\geq -(p-1)\log\left(\sum_{t=1}^{p(p-1)/2} w_t\right) + \lambda\sum_{t=1}^{p(p-1)/2} w_t + (p-1)\log\frac{p-1}{2}, \tag{11}$$

where the second equality follows from (34) with $b = 1$; the forth equality follows from $\mathcal{L}\boldsymbol{w}\cdot\mathbf{1} = \mathbf{0}$; the last inequality holds because $\boldsymbol{w}\geq\mathbf{0}$, and $\mathcal{L}^*\boldsymbol{S}\geq\mathbf{0}$, which follows from (70); the first inequality holds because the smallest eigenvalue $\lambda_1(\mathcal{L}\boldsymbol{w}) = 0$ and

$$\frac{a_1 + a_2 + \ldots + a_n}{n} \geq \sqrt[n]{a_1\cdot a_2\cdots a_n}$$

holds for any non-negative real numbers of $a_1,\ldots,a_n$. A function $g:\Omega\to\mathbb{R}\cup\{+\infty\}$ is called coercive over $\Omega$, if every sequence $\boldsymbol{x}_k\in\Omega$ with $\|\boldsymbol{x}_k\|\to+\infty$ obeys $\lim_{k\to\infty} g(\boldsymbol{x}_k) = +\infty$, where $\Omega\subset\mathbb{R}^n$. Let

$$h(z) = -(p-1)\log z + \lambda z + (p-1)\log\frac{p-1}{2}.$$

A simple calculation yields $\lim_{z\to+\infty} h(z) = +\infty$ if $\lambda > 0$. For any sequence $\boldsymbol{w}_k\in\mathrm{cl}(\mathcal{S}_{\boldsymbol{w}})$ with $\|\boldsymbol{w}_k\|\to+\infty$, where $\mathrm{cl}(\mathcal{S}_{\boldsymbol{w}})$ is the closure of $\mathcal{S}_{\boldsymbol{w}}$, one has $\sum_{t=1}^{p(p-1)/2}[\boldsymbol{w}_k]_t\to+\infty$, because $\sum_{t=1}^{p(p-1)/2}[\boldsymbol{w}_k]_t\geq\|\boldsymbol{w}_k\|$. Then one obtains

$$\lim_{k\to\infty} f(\boldsymbol{w}_k) \geq \lim_{k\to\infty} h\left(\sum_{t=1}^{p(p-1)/2}[\boldsymbol{w}_k]_t\right) = \lim_{z\to+\infty} h(z) = +\infty,$$

where the first inequality follows from (11). Hence, $f(\boldsymbol{w})$ is coercive over $\mathrm{cl}(\mathcal{S}_{\boldsymbol{w}})$. Following from the Extreme Value Theorem in [1], if $\Omega\subset\mathbb{R}^n$ is non-empty and closed, and $g:\Omega\to\mathbb{R}\cup\{+\infty\}$ is lower semi-continuous and coercive, then the optimization $\min_{\boldsymbol{x}\in\Omega} g(\boldsymbol{x})$ has at least one global minimizer. Therefore, by the coercivity of $f(\boldsymbol{w})$, (10) has at least one global minimizer in $\mathrm{cl}(\mathcal{S}_{\boldsymbol{w}})$.

Let $\Omega_A = \{\boldsymbol{w}\in\mathbb{R}^{p(p-1)/2}\,|\,\boldsymbol{w}\geq\mathbf{0}\}$ and $\Omega_B = \{\boldsymbol{w}\in\mathbb{R}^{p(p-1)/2}\,|\,(\mathcal{L}\boldsymbol{w} + \boldsymbol{J})\in\mathcal{S}_{++}^p\}$. $\Omega_A$ is a closed set and $\Omega_B$ is an open set. Then $\mathcal{S}_{\boldsymbol{w}}$ can be rewritten as $\mathcal{S}_{\boldsymbol{w}} = \Omega_A\cap\Omega_B$. Consider the set $V := \mathrm{cl}(\mathcal{S}_{\boldsymbol{w}})\setminus\mathcal{S}_{\boldsymbol{w}}$, we have

$$V \subseteq \left\{\mathrm{cl}(\Omega_A)\cap\mathrm{cl}(\Omega_B)\right\}\setminus\left\{\Omega_A\cap\Omega_B\right\} = \Omega_A\cap\partial\Omega_B, \tag{12}$$

where $\partial\Omega_B$ is the boundary of $\Omega_B$. Notice that every matrix on the boundary of the set of positive definite matrices is positive semi-definite and has zero determinant. Hence, one has $\partial\Omega_B = \{\boldsymbol{w}\in\mathbb{R}^{p(p-1)/2}\,|\,(\mathcal{L}\boldsymbol{w} + \boldsymbol{J})\in\mathcal{S}_+^p, \det(\mathcal{L}\boldsymbol{w} + \boldsymbol{J}) = 0\}$. As a result, for any $\boldsymbol{w}_b\in\mathrm{cl}(\mathcal{S}_{\boldsymbol{w}})\setminus\mathcal{S}_{\boldsymbol{w}}$, $f(\boldsymbol{w}_b) = +\infty$. Therefore, (10) has at least one global minimizer and the minimizer must belong to the set $\mathcal{S}_{\boldsymbol{w}}$. On the other hand, by the strict convexity of $f(\boldsymbol{w})$, (10) has at most one global minimizer in $\mathcal{S}_{\boldsymbol{w}}$. Totally, we conclude that (10) has an unique global minimizer in $\mathcal{S}_{\boldsymbol{w}}$ if $\lambda > 0$.

We prove the theorem through the KKT conditions. The Lagrangian of the optimization (10) is

$$L(\boldsymbol{w}, \boldsymbol{v}) = -\log\det(\mathcal{L}\boldsymbol{w} + \boldsymbol{J}) + \langle\mathcal{L}^*\boldsymbol{S} + \lambda\mathbf{1}, \boldsymbol{w}\rangle - \boldsymbol{v}^\top\boldsymbol{w},$$

where $\boldsymbol{v}$ is a KKT multiplier. Let $(\hat{\boldsymbol{w}}, \hat{\boldsymbol{v}})$ be any pair of points that satisfies the KKT conditions. Then we have

$$-\mathcal{L}^*\left((\mathcal{L}\hat{\boldsymbol{w}} + \boldsymbol{J})^{-1}\right) + \mathcal{L}^*\boldsymbol{S} + \lambda\mathbf{1} - \hat{\boldsymbol{v}} = \mathbf{0}; \tag{13}$$

$$\hat{w}_i\hat{v}_i = 0, \text{ for } i = 1,\ldots,p(p-1)/2; \tag{14}$$

$$\hat{\boldsymbol{w}}\geq\mathbf{0},\ \hat{\boldsymbol{v}}\geq\mathbf{0}; \tag{15}$$

As we know, for any convex optimization with differentiable objective and constraint functions, any point that satisfies the KKT conditions (under Slater's constraint qualification) must be primal and dual optimal. Therefore, $\hat{\boldsymbol{w}}$ must obey $\hat{\boldsymbol{w}} = \arg\min_{\boldsymbol{w} \geq \boldsymbol{0}} f(\boldsymbol{w})$. Note that the pair of points $(\hat{\boldsymbol{w}}, \hat{\boldsymbol{v}})$ that satisfies the KKT conditions is unique. To prove the optimal solution $\hat{\boldsymbol{w}} > \boldsymbol{0}$ holds for (10), we can equivalently prove that the KKT conditions (13)-(15) hold for $(\hat{\boldsymbol{w}} > \boldsymbol{0}, \hat{\boldsymbol{v}} = \boldsymbol{0})$. It is further equivalent to prove that

$$\mathcal{L}^* \left( (\mathcal{L}\hat{\boldsymbol{w}} + \boldsymbol{J})^{-1} \right) = \mathcal{L}^* \boldsymbol{S} + \lambda \boldsymbol{1}. \tag{16}$$

holds for $\hat{\boldsymbol{w}} > \boldsymbol{0}$. Following from Lemma 1.2 with the fact that $\hat{\boldsymbol{w}} \in \mathcal{S}_{\boldsymbol{w}}$, there must exist an unique $\boldsymbol{x}$ such that

$$\mathcal{L}\boldsymbol{x} + \frac{1}{b}\boldsymbol{J} = (\mathcal{L}\hat{\boldsymbol{w}} + b\boldsymbol{J})^{-1} \tag{17}$$

holds for any $b \neq 0$. Thus one has

$$\mathcal{L}^* \left( (\mathcal{L}\hat{\boldsymbol{w}} + \boldsymbol{J})^{-1} \right) = \mathcal{L}^* \left( \mathcal{L}\boldsymbol{x} + \boldsymbol{J} \right) = \mathcal{L}^* \mathcal{L}\boldsymbol{x}, \tag{18}$$

where the first equality follows from (17) with $b = 1$; the second equality holds because $\boldsymbol{J} \in \mathcal{N}(\mathcal{L}^*)$ where $\mathcal{N}(\mathcal{L}^*)$ is the null space of $\mathcal{L}^*$ defined by $\mathcal{N}(\mathcal{L}^*) := \{\boldsymbol{X} \in \mathbb{R}^{p \times p} \mid \mathcal{L}^* \boldsymbol{X} = \boldsymbol{0}\}$. Combining (16) and (18) yields

$$\boldsymbol{x} = (\mathcal{L}^* \mathcal{L})^{-1} (\mathcal{L}^* \boldsymbol{S} + \lambda \boldsymbol{1}), \tag{19}$$

where $\mathcal{L}^* \mathcal{L}$ is invertible according to Lemma 1.3. Recall that $\boldsymbol{S}$ is the sample covariance matrix defined by

$$\boldsymbol{S} = \sum_{k=1}^{n} \boldsymbol{X}^{(k)} \left( \boldsymbol{X}^{(k)} \right)^{\top},$$

where $\boldsymbol{X}^{(1)}, \ldots, \boldsymbol{X}^{(n)}$ are the samples independently drawn from the LGMRF in Definition 2.1. According to the density function of LGMRF, we get $\boldsymbol{1}^{\top} \boldsymbol{X}^{(k)} = 0$ for $k = 1, \ldots, n$. Therefore, $\boldsymbol{S}$ is symmetric and obeys $\boldsymbol{S} \cdot \boldsymbol{1} = \boldsymbol{0}$. It is easy to verify that $\boldsymbol{S} \in \mathcal{R}(\mathcal{L})$, where $\mathcal{R}(\mathcal{L})$ is the range space of $\mathcal{L}$ defined by $\mathcal{R}(\mathcal{L}) := \{\mathcal{L}\boldsymbol{y} \mid \boldsymbol{y} \in \mathbb{R}^{p(p-1)/2}\}$. Hence, there must exist a $\boldsymbol{y} \in \mathbb{R}^{p(p-1)/2}$ such that $\boldsymbol{S} = \mathcal{L}\boldsymbol{y}$. Thus $\mathcal{L}^* \boldsymbol{S} = \mathcal{L}^* \mathcal{L}\boldsymbol{y}$. One further obtains

$$\boldsymbol{y} = (\mathcal{L}^* \mathcal{L})^{-1} \mathcal{L}^* \boldsymbol{S}.$$

Then a simple calculation yields

$$\mathcal{L}(\mathcal{L}^* \mathcal{L})^{-1} \mathcal{L}^* \boldsymbol{S} = \mathcal{L}\boldsymbol{y} = \boldsymbol{S}. \tag{20}$$

Next, we construct a matrix $\boldsymbol{X} = \mathcal{L}\boldsymbol{x} + a\boldsymbol{J}$ with $a > 0$ and have

$$\begin{aligned} \mathcal{L}\boldsymbol{x} + a\boldsymbol{J} &= \mathcal{L}(\mathcal{L}^* \mathcal{L})^{-1} (\mathcal{L}^* \boldsymbol{S} + \lambda \boldsymbol{1}) + a\boldsymbol{J} \\ &= \boldsymbol{S} + \lambda \mathcal{L}(\mathcal{L}^* \mathcal{L})^{-1} \boldsymbol{1} + a\boldsymbol{J} \\ &= \boldsymbol{S} + \frac{\lambda}{2p} \mathcal{L}\boldsymbol{1} + a\boldsymbol{J}, \end{aligned} \tag{21}$$

where the first equality follows from (19); the second equality follows from (20); the third equality holds because $\mathcal{L}^* \mathcal{L}\boldsymbol{1} = 2p\boldsymbol{1}$ and then one has $(\mathcal{L}^* \mathcal{L})^{-1}\boldsymbol{1} = \frac{1}{2p}\boldsymbol{1}$.

Let $\hat{\boldsymbol{X}} = \boldsymbol{D}^{-\frac{1}{2}} \boldsymbol{X} \boldsymbol{D}^{-\frac{1}{2}}$ be the normalized matrix of $\boldsymbol{X}$, where $\boldsymbol{D}$ is a diagonal matrix containing the diagonals of $\boldsymbol{X}$. Notice that each diagonal element of $\hat{\boldsymbol{X}}$ is 1. Next, we will prove that, under some conditions, $\hat{\boldsymbol{X}}$ is an inverse $M$-matrix, that is, $(\hat{\boldsymbol{X}})^{-1}$ is a $M$-matrix. We say $\boldsymbol{A} \in \mathbb{R}^{p \times p}$ is an $M$-matrix if

$$\boldsymbol{A} = s\boldsymbol{I} - \boldsymbol{B}, \tag{22}$$

where $\boldsymbol{B} \in \mathbb{R}^{p \times p}$ is an element-wise non-negative matrix and $s > \rho(\boldsymbol{B})$, the spectral radius of $\boldsymbol{B}$. According to (21), one has

$$X_{ij} = S_{ij} - \frac{\lambda}{2p} + \frac{a}{p}, \quad \text{for } i \neq j,$$

and

$$X_{ii} = S_{ii} + \frac{p-1}{2p}\lambda + \frac{a}{p}, \quad i = 1, \dots, p.$$

Define $\widetilde{S}_{ij} = \max_{i \neq j} S_{ij}$, $\bar{S}_{ij} = \min_{i \neq j} S_{ij}$, $\widetilde{S}_{kk} = \max_k S_{kk}$ and $\bar{S}_{kk} = \min_k S_{kk}$. By the definition of $\hat{\boldsymbol{X}}$, the lower bound $y$ and upper bound $x$ of the elements off the diagonal of $\hat{\boldsymbol{X}}$ can be obtained as below,

$$
\begin{aligned}
\hat{X}_{ij} = \frac{X_{ij}}{\sqrt{X_{ii} X_{jj}}} &= \frac{S_{ij} - \frac{\lambda}{2p} + \frac{a}{p}}{\sqrt{S_{ii} + \frac{p-1}{2p}\lambda + \frac{a}{p}} \cdot \sqrt{S_{jj} + \frac{p-1}{2p}\lambda + \frac{a}{p}}} \\
&\geq \frac{\bar{S}_{ij} - \frac{\lambda}{2p} + \frac{a}{p}}{\widetilde{S}_{kk} + \frac{p-1}{2p}\lambda + \frac{a}{p}} =: y, \quad \forall \, i \neq j.
\end{aligned}
\tag{23}
$$

and

$$
\hat{X}_{ij} \leq \frac{\widetilde{S}_{ij} - \frac{\lambda}{2p} + \frac{a}{p}}{\bar{S}_{kk} + \frac{p-1}{2p}\lambda + \frac{a}{p}} =: x, \quad \forall \, i \neq j.
\tag{24}
$$

Define $s$ by $x^2 = sy + (1-s)y^2$. According to Lemma 1.10, if $0 < y \leq x < 1$ and $s^{-1} \geq p - 2$ with $p > 3$, then $\hat{\boldsymbol{X}}$ is an inverse $M$-matrix. We can see, provided that $y = \frac{1}{p+1}$ and the inequalities

$$
0 < y \leq x \leq \sqrt{2}y < 1,
\tag{25}
$$

hold, then one has

$$
s^{-1} = \frac{y - y^2}{x^2 - y^2} \geq \frac{1 - y}{y} > p - 2,
$$

where the first inequality follows from $x \leq \sqrt{2}y$. Therefore, if $y = \frac{1}{p+1}$ and (25) holds, then $\hat{\boldsymbol{X}}$ is an inverse $M$-matrix.

Next, we prove that if $a = \widetilde{S}_{kk} - (p+1)\bar{S}_{ij} + \lambda$ with $\lambda \geq 2(\sqrt{2}+1)(p+1)(\widetilde{S}_{kk} - \bar{S}_{ij})$, then $y = \frac{1}{p+1}$ and (25) holds. Substituting $a = \widetilde{S}_{kk} - (p+1)\bar{S}_{ij} + \lambda$ into (23) yields $y = \frac{1}{p+1}$. Then it is clear that $y > 0$ and $\sqrt{2}y < 1$. Comparing $x$ and $y$ defined in (23) and (24), respectively, one has $y \leq x$. A simple algebra yields

$$
\begin{aligned}
x &= \frac{\widetilde{S}_{ij} - \frac{\lambda}{2p} + \frac{a}{p}}{\bar{S}_{kk} + \frac{p-1}{2p}\lambda + \frac{a}{p}} \leq \frac{p\widetilde{S}_{kk} - \frac{\lambda}{2} + a}{p\bar{S}_{ij} + \frac{p-1}{2}\lambda + a} \\
&= \frac{(p+1)(\widetilde{S}_{kk} - \bar{S}_{ij}) + \frac{\lambda}{2}}{\widetilde{S}_{kk} - \bar{S}_{ij} + \frac{(p+1)\lambda}{2}} \leq \frac{\sqrt{2}}{p+1} = \sqrt{2}y,
\end{aligned}
$$

where the first inequality follows from $\widetilde{S}_{ij} \leq \widetilde{S}_{kk}$ and $\bar{S}_{kk} \geq \bar{S}_{ij}$ because $\boldsymbol{S} \in \mathcal{S}_+^p$ and $\boldsymbol{S} \cdot \boldsymbol{1} = \boldsymbol{0}$. It is easy to verify that $\lambda \geq 2(\sqrt{2}+1)(p+1)(\widetilde{S}_{kk} - \bar{S}_{ij})$ is large enough to establish the second inequality. Therefore, all the inequalities in (25) hold.

Consequently, by Lemma 1.10, we conclude that $\hat{\boldsymbol{X}}$ is an inverse $M$-matrix when $a = \widetilde{S}_{kk} - (p+1)\bar{S}_{ij} + \lambda$ with $\lambda \geq 2(\sqrt{2}+1)(p+1)(\widetilde{S}_{kk} - \bar{S}_{ij})$. Therefore, $\hat{\boldsymbol{X}}^{-1} = \boldsymbol{D}^{\frac{1}{2}}\boldsymbol{X}^{-1}\boldsymbol{D}^{\frac{1}{2}}$ is an $M$-matrix. Notice that the

elements off the diagonal of an $M$-matrix are non-positive according to (22). As a result, the elements off the diagonal of $\boldsymbol{D}^{\frac{1}{2}}\boldsymbol{X}^{-1}\boldsymbol{D}^{\frac{1}{2}}$ are non-positive, implying that the elements off the diagonal of $\boldsymbol{X}^{-1}$ are also non-positive, because $\boldsymbol{X} = \mathcal{L}\boldsymbol{x} + a\boldsymbol{J}$ is positive definite and thus the diagonal elements of $\boldsymbol{D}$ are positive. The application of (17) with $b = \frac{1}{a}$ yields

$$\boldsymbol{X}^{-1} = [\mathcal{L}\boldsymbol{x} + a\boldsymbol{J}]^{-1} = \mathcal{L}\hat{\boldsymbol{w}} + \frac{1}{a}\boldsymbol{J}.$$

One further obtains

$$[\mathcal{L}\hat{\boldsymbol{w}} + \frac{1}{a}\boldsymbol{J}]_{ij} = -\hat{w}_k + \frac{1}{ap} \leq 0, \quad \forall\, i \neq j,$$

where $k = i - j + \frac{i-1}{2}(2p - j)$. Therefore, we establish

$$\hat{w}_k \geq \frac{1}{ap} = \frac{1}{(\widetilde{S}_{kk} - (p+1)\bar{S}_{ij} + \lambda)p} > 0, \quad \forall\, k,$$

concluding that (16) holds for $\hat{\boldsymbol{w}} > \boldsymbol{0}$. Note that $a = \widetilde{S}_{kk} - (p+1)\bar{S}_{ij} + \lambda > 0$ because $\widetilde{S}_{kk} > 0, \lambda > 0$, and $\bar{S}_{ij} \leq 0$, where $\bar{S}_{ij} \leq 0$ following from $\boldsymbol{S} \cdot \boldsymbol{1} = \boldsymbol{0}$ and each diagonal element $S_{ii} \geq 0$ since $\boldsymbol{S}$ is positive semi-definite. □

## 1.3 Proofs of Theorem 3.4

*Proof.* Let $\boldsymbol{x} \in \mathbb{R}^{p(p-1)/2}$. According to the definition of $\mathcal{L}$, $\mathcal{L}\boldsymbol{x}$ must obey $[\mathcal{L}\boldsymbol{x}]_{ij} = [\mathcal{L}\boldsymbol{x}]_{ji}$, for any $i \neq j$ and $(\mathcal{L}\boldsymbol{x}) \cdot \boldsymbol{1} = \boldsymbol{0}$.

Next, we will show that $\mathcal{L}\boldsymbol{x}$ is positive semi-definite for any $\boldsymbol{x} \geq \boldsymbol{0}$ by the Gershgorin circle theorem [5]. Given a matrix $\boldsymbol{X} \in \mathbb{R}^{p \times p}$ with entries $X_{ij}$. Let $R_i(\boldsymbol{X}) = \sum_{j \neq i}|X_{ij}|$ be the sum of the absolute values of the non-diagonal entries in the $i$-th row. Then a Gershgorin disc is the disc $D(X_{ii}, R_i(\boldsymbol{X}))$ centered at $X_{ii}$ on the complex plane with radius $R_i(\boldsymbol{X})$. Gershgorin circle theorem [3] shows that each eigenvalue of $\boldsymbol{X}$ lies within at least one of the Gershgorin discs. For any $\boldsymbol{x} \geq \boldsymbol{0}$, $R_i(\mathcal{L}\boldsymbol{x}) = [\mathcal{L}\boldsymbol{x}]_{ii}$ holds for each $i$ because $(\mathcal{L}\boldsymbol{x}) \cdot \boldsymbol{1} = \boldsymbol{0}$ and $[\mathcal{L}\boldsymbol{x}]_{ij} \leq 0$ for any $i \neq j$. For any given eigenvalue $\lambda$ of $\mathcal{L}\boldsymbol{x}$, by Gershgorin circle theorem, there must exist one Gershgorin disc $D([\mathcal{L}\boldsymbol{x}]_{ii}, R_i(\mathcal{L}\boldsymbol{x}))$ such that

$$|\lambda - [\mathcal{L}\boldsymbol{x}]_{ii}| \leq R_i(\boldsymbol{X}) = [\mathcal{L}\boldsymbol{x}]_{ii}, \tag{26}$$

indicating that $\lambda \geq 0$. Note that the eigenvalues of $\mathcal{L}\boldsymbol{x}$ are real since $\mathcal{L}\boldsymbol{x}$ is symmetric. Therefore, one has

$$\mathcal{L}\boldsymbol{x} \in \mathcal{S}_+^p, \quad \forall \boldsymbol{x} \geq \boldsymbol{0}. \tag{27}$$

Finally, we will prove that $\text{rank}(\mathcal{L}\boldsymbol{x}) = p - 1 \Leftrightarrow (\mathcal{L}\boldsymbol{x} + \boldsymbol{J}) \in \mathcal{S}_{++}^p$, for any $\boldsymbol{x} \geq \boldsymbol{0}$. On one hand, if $\text{rank}(\mathcal{L}\boldsymbol{x}) = p - 1$, then $\mathcal{L}\boldsymbol{x} + \boldsymbol{J}$ admits the eigenvalue decomposition $\boldsymbol{U}\boldsymbol{\Lambda}\boldsymbol{U}^\top$, where $\boldsymbol{U} = [\boldsymbol{U}_s \; \frac{1}{\sqrt{p}}\boldsymbol{1}]$ and $\boldsymbol{\Lambda}$ is a diagonal matrix with the diagonal elements $[\lambda_2, \ldots, \lambda_p, 1]$. Here $\lambda_{i=2}^p$ are the nonzero eigenvalues of $\mathcal{L}\boldsymbol{x}$ and $\boldsymbol{U}_s$ is a $p \times (p-1)$ matrix whose columns are the corresponding eigenvectors of $\mathcal{L}\boldsymbol{x}$. Note that the nonzero eigenvalue $\lambda_{i=2}^p > 0$ because $\mathcal{L}\boldsymbol{x} \in \mathcal{S}_+^p$. Therefore, one has $(\mathcal{L}\boldsymbol{x} + \boldsymbol{J}) \in \mathcal{S}_{++}^p$. On the other hand, if $(\mathcal{L}\boldsymbol{x} + \boldsymbol{J}) \in \mathcal{S}_{++}^p$, then $\text{rank}(\mathcal{L}\boldsymbol{x}) \geq \text{rank}(\mathcal{L}\boldsymbol{x} + \boldsymbol{J}) - \text{rank}(\boldsymbol{J}) = p - 1$ because $\mathcal{L}\boldsymbol{x} + \boldsymbol{J}$ is full rank and $\boldsymbol{J}$ is rank one. Furthermore, $\text{rank}(\mathcal{L}\boldsymbol{x}) \leq p - 1$ because $(\mathcal{L}\boldsymbol{x}) \cdot \boldsymbol{1} = \boldsymbol{0}$. Therefore, we conclude that $\text{rank}(\mathcal{L}\boldsymbol{x}) = p - 1$, completing the proof. □

## 1.4 Proofs of Theorem 3.8 and Corollary 3.9

*Proof.* We first prove Theorem 3.8. Take the regularization parameter $\lambda = \sqrt{4\alpha c_0^{-1} \log p/n}$ for some $\alpha > 2$, and the sample size

$$n \geq \max\left(94\alpha c_0^{-1}\lambda_{\max}^2(\mathcal{L}\boldsymbol{w}^\star)s \log p, 8\alpha \log p\right), \tag{28}$$

where $c_0$ is a constant defined in Lemma 1.8. Notice that the sample size $n$ in (28) satisfies the conditions on the number of samples in Lemmas 1.4, 1.5, 1.6 and 1.7. We choose the initial point $\hat{\boldsymbol{w}}^{(0)}$ of Algorithm 1 satisfying $|\mathrm{supp}^+(\hat{\boldsymbol{w}}^{(0)})| \leq s$.

Define an event $\mathcal{J} = \left\{ \left\| \mathcal{L}^*\big( (\mathcal{L}\boldsymbol{w}^\star + \boldsymbol{J})^{-1} - \boldsymbol{S}\big) \right\|_{\max} \leq \lambda/2 \right\}$. According to Lemma 1.7, $\left\| \mathcal{L}^*\big( (\mathcal{L}\boldsymbol{w}^\star + \boldsymbol{J})^{-1} - \boldsymbol{S}\big) \right\|_{\max} \leq \lambda/2$ holds with probability at least $1 - 1/p^{\alpha-2}$. Under the event $\mathcal{J}$, one applies Lemma 1.6 and obtains, for any $k \geq 1$,

$$\left\| \hat{\boldsymbol{w}}^{(k)} - \boldsymbol{w}^\star \right\| \leq 2\lambda_{\max}^2(\mathcal{L}\boldsymbol{w}^\star) \left\| \big( \mathcal{L}^*\big( (\mathcal{L}\boldsymbol{w}^\star + \boldsymbol{J})^{-1} - \boldsymbol{S}\big) \big)_{\mathcal{S}^\star} \right\| + \frac{3}{2+\sqrt{2}} \left\| \hat{\boldsymbol{w}}^{(k-1)} - \boldsymbol{w}^\star \right\|,$$

By induction, if $d_k \leq a_0 + \rho d_{k-1}$ for any $k \geq 1$ with $\rho \in [0,1)$, then

$$d_k \leq \frac{1-\rho^k}{1-\rho} a_0 + \rho^k d_0. \tag{29}$$

Taking $a_0 = 2\lambda_{\max}^2(\mathcal{L}\boldsymbol{w}^\star) \left\| \big( \mathcal{L}^*\big( (\mathcal{L}\boldsymbol{w}^\star + \boldsymbol{J})^{-1} - \boldsymbol{S}\big) \big)_{\mathcal{S}^\star} \right\|$, $\rho = \frac{3}{2+\sqrt{2}}$ and $d_k = \left\| \hat{\boldsymbol{w}}^{(k)} - \boldsymbol{w}^\star \right\|$, one obtains

$$\left\| \hat{\boldsymbol{w}}^{(k)} - \boldsymbol{w}^\star \right\| \leq 2\sqrt{2}(\sqrt{2}+1)^2 \lambda_{\max}^2(\mathcal{L}\boldsymbol{w}^\star) \left\| \big( \mathcal{L}^*\big( (\mathcal{L}\boldsymbol{w}^\star + \boldsymbol{J})^{-1} - \boldsymbol{S}\big) \big)_{\mathcal{S}^\star} \right\| + \left( \frac{3}{2+\sqrt{2}} \right)^k \left\| \hat{\boldsymbol{w}}^{(0)} - \boldsymbol{w}^\star \right\|.$$

Under the event $\mathcal{J}$, $\left\| \big( \mathcal{L}^*\big( (\mathcal{L}\boldsymbol{w}^\star + \boldsymbol{J})^{-1} - \boldsymbol{S}\big) \big)_{\mathcal{S}^\star} \right\|$ can be bounded by

$$\left\| \big( \mathcal{L}^*\big( (\mathcal{L}\boldsymbol{w}^\star + \boldsymbol{J})^{-1} - \boldsymbol{S}\big) \big)_{\mathcal{S}^\star} \right\| \leq \sqrt{s}\lambda/2 \leq \sqrt{\alpha c_0^{-1} s \log p/n}. \tag{30}$$

Therefore, under the event $\mathcal{J}$, which holds with probability at least $1 - 1/p^{\alpha-2}$, one has

$$\left\| \hat{\boldsymbol{w}}^{(k)} - \boldsymbol{w}^\star \right\| \leq 2(3\sqrt{2}+4)\lambda_{\max}^2(\mathcal{L}\boldsymbol{w}^\star)\sqrt{\alpha c_0^{-1} s \log p/n} + \left( \frac{3}{2+\sqrt{2}} \right)^k \left\| \hat{\boldsymbol{w}}^{(0)} - \boldsymbol{w}^\star \right\|.$$

Next, we prove Corollary 3.9. Under the event $\mathcal{J}$, one applies Lemma 1.6 and obtains,

$$\left\| \mathcal{L}\hat{\boldsymbol{w}}^{(k)} - \mathcal{L}\boldsymbol{w}^\star \right\|_{\mathrm{F}} \leq 2\sqrt{2}\lambda_{\max}^2(\mathcal{L}\boldsymbol{w}^\star) \left\| \big( \mathcal{L}^*\big( (\mathcal{L}\boldsymbol{w}^\star + \boldsymbol{J})^{-1} - \boldsymbol{S}\big) \big)_{\mathcal{S}^\star} \right\| + \frac{3}{2+\sqrt{2}} \left\| \mathcal{L}\hat{\boldsymbol{w}}^{(k-1)} - \mathcal{L}\boldsymbol{w}^\star \right\|_{\mathrm{F}} \tag{31}$$

holds for any $k \geq 1$. Taking $a_0 = 2\sqrt{2}\lambda_{\max}^2(\mathcal{L}\boldsymbol{w}^\star) \left\| \big( \mathcal{L}^*\big( (\mathcal{L}\boldsymbol{w}^\star + \boldsymbol{J})^{-1} - \boldsymbol{S}\big) \big)_{\mathcal{S}^\star} \right\|$, $\rho = \frac{3}{2+\sqrt{2}}$ and $d_k = \left\| \mathcal{L}\hat{\boldsymbol{w}}^{(k)} - \mathcal{L}\boldsymbol{w}^\star \right\|_{\mathrm{F}}$, by (29) one has

$$\left\| \mathcal{L}\hat{\boldsymbol{w}}^{(k)} - \mathcal{L}\boldsymbol{w}^\star \right\|_{\mathrm{F}} \leq 4(\sqrt{2}+1)^2 \lambda_{\max}^2(\mathcal{L}\boldsymbol{w}^\star) \left\| \big( \mathcal{L}^*\big( (\mathcal{L}\boldsymbol{w}^\star + \boldsymbol{J})^{-1} - \boldsymbol{S}\big) \big)_{\mathcal{S}^\star} \right\| + \left( \frac{3}{2+\sqrt{2}} \right)^k \left\| \mathcal{L}\hat{\boldsymbol{w}}^{(0)} - \mathcal{L}\boldsymbol{w}^\star \right\|_{\mathrm{F}}.$$

Similarly, according to (30), one obtains

$$\left\| \mathcal{L}\hat{\boldsymbol{w}}^{(k)} - \mathcal{L}\boldsymbol{w}^\star \right\|_{\mathrm{F}} \leq 4(3+2\sqrt{2})\lambda_{\max}^2(\mathcal{L}\boldsymbol{w}^\star)\sqrt{\alpha c_0^{-1} s \log p/n} + \left( \frac{3}{2+\sqrt{2}} \right)^k \left\| \mathcal{L}\hat{\boldsymbol{w}}^{(0)} - \mathcal{L}\boldsymbol{w}^\star \right\|_{\mathrm{F}}$$

holds at least $1 - 1/p^{\alpha-2}$.

Alternative to (29), one obtains

$$d_k \leq \frac{1-\rho^{k-1}}{1-\rho} a_0 + \rho^{k-1} d_1,$$

and correspondingly establishes

$$\left\| \mathcal{L}\hat{\boldsymbol{w}}^{(k)} - \mathcal{L}\boldsymbol{w}^\star \right\|_{\mathrm{F}} \leq 4(3+2\sqrt{2})\lambda_{\max}^2(\mathcal{L}\boldsymbol{w}^\star)\sqrt{\alpha c_0^{-1} s \log s/n} + \left( \frac{3}{2+\sqrt{2}} \right)^{k-1} \left\| \mathcal{L}\hat{\boldsymbol{w}}^{(1)} - \mathcal{L}\boldsymbol{w}^\star \right\|_{\mathrm{F}}. \tag{32}$$

To apply Lemma 1.4, we first check the necessary conditions of the lemma. Let $\boldsymbol{z}^{(0)}$ satisfy $z_i^{(0)} = h_\lambda'(\hat{w}_i^{(0)})$, $i \in [p(p-1)/2]$. Notice that $z_i^{(0)} \in [0, \lambda]$ for $i \in [p(p-1)/2]$ by Assumption 3.5. According to (7), $\mathcal{E}^{(1)} = \{\mathcal{S}^\star \cup \mathcal{S}^{(1)}\}$, where $\mathcal{S}^{(1)} = \{i \in [p(p-1)/2] \mid w_i^{(0)} \geq b\}$. For any $i \in \{\mathcal{S}^{(1)}\}^c$, one has

$$z_i^{(0)} = h_\lambda'(\hat{w}_i^{(0)}) \geq h_\lambda'(b) \geq \frac{\lambda}{2},$$

where the first inequality holds because $\hat{w}_i^{(0)} < b$ for any $i \in \{\mathcal{S}^{(1)}\}^c$, and $h_\lambda'$ is non-increasing according to Assumption 3.5; the second inequality directly follows from Assumption 3.5. Hence one has

$$\left\| \boldsymbol{z}_{\{\mathcal{E}^{(1)}\}^c}^{(0)} \right\|_{\min} \geq \left\| \boldsymbol{z}_{\{\mathcal{S}^{(1)}\}^c}^{(0)} \right\|_{\min} \geq \lambda/2.$$

One also obtains $|\mathcal{E}^{(1)}| < 2s$ by Lemma 1.5, and $\mathcal{S}^\star \subseteq \mathcal{E}^{(1)}$ by the definition of $\mathcal{E}^{(1)}$. Therefore, one can apply Lemma 1.4 with $\mathcal{E} = \mathcal{E}^{(1)}$ and $\boldsymbol{z} = \boldsymbol{z}^{(0)}$ and obtains

$$\left\| \mathcal{L}\hat{\boldsymbol{w}}^{(1)} - \mathcal{L}\boldsymbol{w}^\star \right\|_{\mathrm{F}} \leq 2(1 + \sqrt{2})\lambda_{\max}^2(\mathcal{L}\boldsymbol{w}^\star)\sqrt{s}\lambda.$$

If $t \geq \log_{\frac{2+\sqrt{2}}{3}}(\lambda\sqrt{n/\log p}) = \log\left(\sqrt{4\alpha c_0^{-1}}\right)/\log\frac{2+\sqrt{2}}{3}$, a simple algebra yields

$$\left(\frac{3}{2+\sqrt{2}}\right)^{t-1} \left\| \mathcal{L}\hat{\boldsymbol{w}}^{(1)} - \mathcal{L}\boldsymbol{w}^\star \right\|_{\mathrm{F}} \leq \frac{2(3\sqrt{2}+4)}{3}(\lambda\sqrt{n/\log p})^{-1}\lambda_{\max}^2(\mathcal{L}\boldsymbol{w}^\star)\sqrt{s}\lambda \lesssim \sqrt{s\log p/n}. \quad (33)$$

Taking $k \geq \lceil 4\log(4\alpha c_0^{-1})\rceil \geq \log\left(\sqrt{4\alpha c_0^{-1}}\right)/\log\frac{2+\sqrt{2}}{3}$, and combining (32) and (33) together, we can conclude that

$$\left\| \mathcal{L}\hat{\boldsymbol{w}}^{(k)} - \mathcal{L}\boldsymbol{w}^\star \right\|_{\mathrm{F}} \lesssim \sqrt{s\log p/n},$$

completing the proof. □

# 2 Proof of Technical Lemmas

This section contains the proofs of technical lemmas used in Section 1.

## 2.1 Proof of Lemma 1.1

*Proof.* The gradient of $f(\boldsymbol{w})$ is $\nabla f(\boldsymbol{w}) = -\mathcal{L}^*(\mathcal{L}\boldsymbol{w} + \boldsymbol{J})^{-1}$ and its Hessian matrix is $\nabla^2 f(\boldsymbol{w})$ with the $k$-th column being

$$\begin{aligned} \left[\nabla^2 f(\boldsymbol{w})\right]_{:,k} &= \frac{\partial(\nabla f(\boldsymbol{w}))}{\partial w_k} = -\mathcal{L}^*\left(\frac{\partial(\mathcal{L}\boldsymbol{w} + \boldsymbol{J})^{-1}}{\partial w_k}\right) \\ &= \mathcal{L}^*\left((\mathcal{L}\boldsymbol{w} + \boldsymbol{J})^{-1}\frac{\partial(\mathcal{L}\boldsymbol{w} + \boldsymbol{J})}{\partial w_k}(\mathcal{L}\boldsymbol{w} + \boldsymbol{J})^{-1}\right) \\ &= \mathcal{L}^*\left((\mathcal{L}\boldsymbol{w} + \boldsymbol{J})^{-1}\boldsymbol{A}_k(\mathcal{L}\boldsymbol{w} + \boldsymbol{J})^{-1}\right), \end{aligned}$$

where $\boldsymbol{A}_k \in \mathbb{R}^{p \times p}$ is a matrix with $[\boldsymbol{A}_k]_{ii} = [\boldsymbol{A}_k]_{jj} = 1$, $[\boldsymbol{A}_k]_{ij} = [\boldsymbol{A}_k]_{ji} = -1$ and zeros for the other elements, in which $i, j \in \mathbb{Z}^+$ obeying $k = i - j + \frac{j-1}{2}(2p - j)$ and $i > j$. Therefore, $\nabla^2 f(\boldsymbol{w})$ can be written as

$$\nabla^2 f(\boldsymbol{w}) = [\mathcal{L}^*\boldsymbol{B}_1, \mathcal{L}^*\boldsymbol{B}_2, \ldots, \mathcal{L}^*\boldsymbol{B}_{p(p-1)/2}],$$

where $\boldsymbol{B}_k = (\mathcal{L}\boldsymbol{w} + \boldsymbol{J})^{-1}\boldsymbol{A}_k(\mathcal{L}\boldsymbol{w} + \boldsymbol{J})^{-1}$, for $k = 1, 2, \ldots, p(p-1)/2$. Then one has

$$
\begin{aligned}
\boldsymbol{x}^\top \nabla^2 f(\boldsymbol{w})\boldsymbol{x} &= \boldsymbol{x}^\top [\mathcal{L}^* \boldsymbol{B}_1, \mathcal{L}^* \boldsymbol{B}_2, \ldots, \mathcal{L}^* \boldsymbol{B}_{p(p-1)/2}]\boldsymbol{x} \\
&= \boldsymbol{x}^\top \mathcal{L}^* \left( \sum_{k=1}^{p(p-1)/2} x_k \boldsymbol{B}_k \right) \\
&= \boldsymbol{x}^\top \mathcal{L}^* \left( (\mathcal{L}\boldsymbol{w} + \boldsymbol{J})^{-1} \left( \sum_{k=1}^{p(p-1)/2} x_k \boldsymbol{A}_k \right) (\mathcal{L}\boldsymbol{w} + \boldsymbol{J})^{-1} \right) \\
&= \boldsymbol{x}^\top \mathcal{L}^* \left( (\mathcal{L}\boldsymbol{w} + \boldsymbol{J})^{-1}\mathcal{L}\boldsymbol{x}(\mathcal{L}\boldsymbol{w} + \boldsymbol{J})^{-1} \right) \\
&= \left\langle \mathcal{L}\boldsymbol{x}, (\mathcal{L}\boldsymbol{w} + \boldsymbol{J})^{-1}\mathcal{L}\boldsymbol{x}(\mathcal{L}\boldsymbol{w} + \boldsymbol{J})^{-1} \right\rangle \\
&= \operatorname{vec}(\mathcal{L}\boldsymbol{x})^\top \operatorname{vec}\left( (\mathcal{L}\boldsymbol{w} + \boldsymbol{J})^{-1}\mathcal{L}\boldsymbol{x}(\mathcal{L}\boldsymbol{w} + \boldsymbol{J})^{-1} \right) \\
&= \operatorname{vec}(\mathcal{L}\boldsymbol{x})^\top \left( (\mathcal{L}\boldsymbol{w} + \boldsymbol{J})^{-1} \otimes (\mathcal{L}\boldsymbol{w} + \boldsymbol{J})^{-1} \right) \operatorname{vec}(\mathcal{L}\boldsymbol{x}),
\end{aligned}
$$

where the forth equality follows from the definition of $\mathcal{L}$, and the last equality follows from the property of Kronecker product that $\operatorname{vec}(\boldsymbol{ABC}) = \left( \boldsymbol{C}^\top \otimes \boldsymbol{A} \right) \operatorname{vec}(\boldsymbol{B})$. $\qquad \square$

## 2.2 Proof of Lemma 1.2

*Proof.* Let $\boldsymbol{X} = \mathcal{L}\boldsymbol{w} + b\boldsymbol{J}$ with any $b \neq 0$ and any given $\boldsymbol{w} \in \mathbb{R}^{p(p-1)/2}$ obeying $(\mathcal{L}\boldsymbol{w} + \boldsymbol{J}) \in \mathcal{S}_{++}^p$. It is easy to verify that the column spaces as well as row spaces of $\mathcal{L}\boldsymbol{w}$ and $\boldsymbol{J}$ are orthogonal with each other. Hence $\boldsymbol{X}$ admits the eigenvalue decomposition

$$
\boldsymbol{X} = \mathcal{L}\boldsymbol{w} + b\boldsymbol{J} = \begin{bmatrix} \boldsymbol{U} & \boldsymbol{u} \end{bmatrix} \begin{bmatrix} \boldsymbol{\Lambda} & \boldsymbol{0} \\ \boldsymbol{0} & b \end{bmatrix} \begin{bmatrix} \boldsymbol{U} & \boldsymbol{u} \end{bmatrix}^\top, \tag{34}
$$

where $\mathcal{L}\boldsymbol{w} = \boldsymbol{U\Lambda U}^\top$ and $b\boldsymbol{J} = b\boldsymbol{uu}^\top$ with $\boldsymbol{u} = \frac{1}{\sqrt{p}}\boldsymbol{1}_p$, in which $\boldsymbol{1}_p \in \mathbb{R}^p$ with each element equal to 1. Notice that $\boldsymbol{\Lambda}$ is non-singular. $\boldsymbol{X}^{-1}$ admits the eigenvalue decomposition

$$
\boldsymbol{X}^{-1} = \begin{bmatrix} \boldsymbol{U} & \boldsymbol{u} \end{bmatrix} \begin{bmatrix} \boldsymbol{\Lambda}^{-1} & \boldsymbol{0} \\ \boldsymbol{0} & \frac{1}{b} \end{bmatrix} \begin{bmatrix} \boldsymbol{U} & \boldsymbol{u} \end{bmatrix}^\top = \boldsymbol{U\Lambda}^{-1}\boldsymbol{U}^\top + \frac{1}{b}\boldsymbol{J},
$$

It is easy to check that $\boldsymbol{U\Lambda}^{-1}\boldsymbol{U}^\top$ is symmetric and $\boldsymbol{U\Lambda}^{-1}\boldsymbol{U}^\top \cdot \boldsymbol{1}_p = \boldsymbol{0}$. Therefore, there must exist a $\boldsymbol{x}$ such that $\mathcal{L}\boldsymbol{x} = \boldsymbol{U\Lambda}^{-1}\boldsymbol{U}^\top$. One further obtains $x_k = [\boldsymbol{U\Lambda}^{-1}\boldsymbol{U}^\top]_{ij}$, for $k = 1, \ldots, p(p-1)/2$, where $i, j \in \mathbb{Z}^+$ satisfying $k = i - j + \frac{j-1}{2}(2p - j)$ and $i > j$. Hence such $\boldsymbol{x}$ is fixed and unique for a given $\boldsymbol{w}$. Note that $\boldsymbol{x}$ is independent of $b$, and thus $\mathcal{L}\boldsymbol{x} + \frac{1}{b}\boldsymbol{J} = (\mathcal{L}\boldsymbol{w} + b\boldsymbol{J})^{-1}$ holds for any $b \neq 0$, completing the proof. $\qquad \square$

## 2.3 Proof of Lemma 1.3

*Proof.* Define an index set $\Omega_t$ by

$$
\Omega_t := \left\{ l \in [p(p-1)/2] \,|\, [\mathcal{L}\boldsymbol{x}]_{tt} = \sum_l x_l \right\}, \quad t \in [p]. \tag{35}
$$

According to the definition of $\mathcal{L}$, for any $\boldsymbol{x} \in \mathbb{R}^{p(p-1)/2}$, one obtains that $\mathcal{L}\boldsymbol{x} \in \mathbb{R}^{p \times p}$ obeys

$$
[\mathcal{L}\boldsymbol{x}]_{ij} = \begin{cases} -x_k & i > j, \\ [\mathcal{L}\boldsymbol{x}]_{ji} & i < j, \\ \sum_{l \in \Omega_i} x_l & i = j, \end{cases} \tag{36}
$$

where $k = i - j + \frac{j-1}{2}(2p - j)$. By the definition of $\mathcal{L}^*$, one further obtains that $\mathcal{L}^* \mathcal{L} \boldsymbol{x} \in \mathbb{R}^{p(p-1)/2}$ satisfies

$$[\mathcal{L}^* \mathcal{L} \boldsymbol{x}]_k = [\mathcal{L} \boldsymbol{x}]_{ii} + [\mathcal{L} \boldsymbol{x}]_{jj} + 2x_k = \sum_{l \in \Omega_i \cup \Omega_j} x_l + 2x_k = \sum_{l \in \Omega_i \cup \Omega_j \backslash k} x_l + 4x_k, \tag{37}$$

where $i, j \in [p]$ satisfying $k = i - j + \frac{j-1}{2}(2p - j)$ and $i > j$. The last equality holds because

$$\sum_{l \in \Omega_i \cup \Omega_j} x_l = [\mathcal{L} \boldsymbol{x}]_{ii} + [\mathcal{L} \boldsymbol{x}]_{jj} = -\sum_{m \neq i} [\mathcal{L} \boldsymbol{x}]_{im} - \sum_{m \neq j} [\mathcal{L} \boldsymbol{x}]_{jm} = 2x_k - \sum_{m \neq i,j} [\mathcal{L} \boldsymbol{x}]_{im} - \sum_{m \neq i,j} [\mathcal{L} \boldsymbol{x}]_{jm},$$

where the last equality follows from $[\mathcal{L} \boldsymbol{x}]_{ij} = -x_k$ according to (36).

According to (37), we conclude that there exists a matrix $\boldsymbol{M} \in \mathbb{R}^{\frac{p(p-1)}{2} \times \frac{p(p-1)}{2}}$ such that $\mathcal{L}^* \mathcal{L} \boldsymbol{x} = \boldsymbol{M} \boldsymbol{x}$ for any $\boldsymbol{x} \in \mathbb{R}^{p(p-1)/2}$, and $\boldsymbol{M}$ obeys

$$M_{kl} = \begin{cases} 4 & l = k, \\ 1 & l \in (\Omega_i \cup \Omega_j) \backslash k, \\ 0 & \text{Otherwise}, \end{cases} \tag{38}$$

where $i, j \in [p]$ satisfying $k = i - j + \frac{j-1}{2}(2p - j)$ and $i > j$. Note that we use the fact that $\{\Omega_i \backslash k\} \cap \{\Omega_j \backslash k\} = \varnothing$ with $k = i - j + \frac{j-1}{2}(2p - j)$.

Finally, we will compute the minimum and maximum eigenvalues of $\boldsymbol{M}$. To compute the minimum eigenvalue of $\boldsymbol{M}$, one has

$$\lambda_{\min}(\boldsymbol{M}) = \inf_{\boldsymbol{x} \neq \boldsymbol{0}} \frac{\boldsymbol{x}^\top \boldsymbol{M} \boldsymbol{x}}{\|\boldsymbol{x}\|^2} = \inf_{\boldsymbol{x} \neq \boldsymbol{0}} \frac{\|\mathcal{L} \boldsymbol{x}\|_{\mathrm{F}}^2}{\|\boldsymbol{x}\|^2} = \inf_{\boldsymbol{x} \neq \boldsymbol{0}} \frac{2 \sum_{k=1}^{p(p-1)/2} x_k^2 + \sum_{i=1}^{p} ([\mathcal{L} \boldsymbol{x}]_{ii})^2}{\|\boldsymbol{x}\|^2} \geq 2.$$

with equality when $[\mathcal{L} \boldsymbol{x}]_{11} = \ldots = [\mathcal{L} \boldsymbol{x}]_{pp} = 0$, which can be written as $\boldsymbol{Q} \boldsymbol{x} = \boldsymbol{0}$ with $\boldsymbol{Q} \in \mathbb{R}^{p \times \frac{p(p-1)}{2}}$. Obviously, there must exist a nonzero solution to $\boldsymbol{Q} \boldsymbol{x} = \boldsymbol{0}$, and thus $\lambda_{\min}(\boldsymbol{M}) = 2$. To compute the maximum eigenvalue of $\boldsymbol{M}$, one has

$$\begin{aligned}
\lambda_{\max}(\boldsymbol{M}) &= \sup_{\boldsymbol{x} \neq \boldsymbol{0}} \frac{\boldsymbol{x}^\top \boldsymbol{M} \boldsymbol{x}}{\|\boldsymbol{x}\|^2} = \sup_{\boldsymbol{x} \neq \boldsymbol{0}} \frac{2 \sum_{k=1}^{p(p-1)/2} x_k^2 + \sum_{t=1}^{p} ([\mathcal{L} \boldsymbol{x}]_{tt})^2}{\|\boldsymbol{x}\|^2} \\
&= \sup_{\boldsymbol{x} \neq \boldsymbol{0}} \frac{4 \sum_{k=1}^{p(p-1)/2} x_k^2 + \sum_{t=1}^{p} \sum_{i,j \in \Omega_t,\ i \neq j} x_i x_j}{\|\boldsymbol{x}\|^2} \\
&\leq \sup_{\boldsymbol{x} \neq \boldsymbol{0}} \frac{4 \sum_{k=1}^{p(p-1)/2} x_k^2 + \frac{1}{2} \sum_{t=1}^{p} \sum_{i,j \in \Omega_t,\ i \neq j} (x_i^2 + x_j^2)}{\|\boldsymbol{x}\|^2} \\
&= \sup_{\boldsymbol{x} \neq \boldsymbol{0}} \frac{4 \sum_{k=1}^{p(p-1)/2} x_k^2 + \sum_{t=1}^{p} (|\Omega_t| - 1) \sum_{i \in \Omega_t} x_i^2}{\|\boldsymbol{x}\|^2} \\
&= \sup_{\boldsymbol{x} \neq \boldsymbol{0}} \frac{4 \sum_{k=1}^{p(p-1)/2} x_k^2 + (p - 2) \sum_{t=1}^{p} \sum_{i \in \Omega_t} x_i^2}{\|\boldsymbol{x}\|^2} \\
&= 2p,
\end{aligned}$$

with equality when each element of $\boldsymbol{x}$ is equal with each other, and thus $\lambda_{\max}(\boldsymbol{M}) = 2p$. The last second equality is obtained by plugging $|\Omega_t| = p - 1$ with $t \in [p]$, which is easy to verify according to the definition of $\Omega_t$ in (35); the last equality follows from $\sum_{t=1}^{p} \sum_{i \in \Omega_t} x_i^2 = 2 \sum_{k=1}^{p(p-1)/2} x_k^2$, because for any $k \in [p(p-1)/2]$, $k \in \Omega_t$ only holds with $t = \{i, j\}$, where $i, j \in [p]$ obeying $k = i - j + \frac{j-1}{2}(2p - j)$ and $i > j$. $\qquad \square$

## 2.4 Proof of Lemma 1.4

*Proof.* Take $\lambda = \sqrt{4\alpha c_0^{-1}\log p/n}$ and $n \geq 94\alpha c_0^{-1}\lambda_{\max}^2(\mathcal{L}\boldsymbol{w}^\star)s\log p$. Define a local region

$$\mathcal{B}_{\boldsymbol{M}}(\boldsymbol{w}^\star;\lambda_{\max}(\mathcal{L}\boldsymbol{w}^\star)) = \{\boldsymbol{w}\,|\,\boldsymbol{w} \in \mathbb{B}_{\boldsymbol{M}}(\boldsymbol{w}^\star;\lambda_{\max}(\mathcal{L}\boldsymbol{w}^\star)) \cap \mathcal{S}_{\boldsymbol{w}}\},$$

where $\mathbb{B}_{\boldsymbol{M}}(\boldsymbol{w}^\star;r) = \{\boldsymbol{w} \in \mathbb{R}^{p(p-1)/2}\,|\,\|\boldsymbol{w}-\boldsymbol{w}^\star\|_{\boldsymbol{M}} \leq r\}$, in which $\|\boldsymbol{x}\|_{\boldsymbol{M}}^2 = \langle\boldsymbol{x},\boldsymbol{M}\boldsymbol{x}\rangle = \|\mathcal{L}\boldsymbol{x}\|_{\mathrm{F}}^2$ with $\boldsymbol{M} \succ \boldsymbol{0}$ defined in Lemma 1.3, and $\mathcal{S}_{\boldsymbol{w}} = \{\boldsymbol{w} \in \mathbb{R}^{p(p-1)/2}\,|\,\boldsymbol{w} \geq \boldsymbol{0},(\mathcal{L}\boldsymbol{w}+\boldsymbol{J}) \in \mathcal{S}_{++}^p\}$. It is easy to check that $\boldsymbol{w}^\star \in \mathcal{B}_{\boldsymbol{M}}(\boldsymbol{w}^\star;\lambda_{\max}(\mathcal{L}\boldsymbol{w}^\star))$.

Recall that $\hat{\boldsymbol{w}}$ minimizes the optimization

$$\min_{\boldsymbol{w}\geq\boldsymbol{0}} -\log\det(\mathcal{L}\boldsymbol{w}+\boldsymbol{J}) + \mathrm{tr}\,(\mathcal{L}\boldsymbol{w}\boldsymbol{S}) + \boldsymbol{z}^\top\boldsymbol{w}, \tag{39}$$

where $0 \leq z_i \leq \lambda$ for $i \in [p(p-1)/2]$. We can see the optimization problems (1) and (39) have the same feasible set. Therefore, $\mathcal{S}_{\boldsymbol{w}}$ is also the feasible set of (39) and thus $\hat{\boldsymbol{w}} \in \mathcal{S}_{\boldsymbol{w}}$ must hold.

Next, we will prove that $\hat{\boldsymbol{w}} \in \mathcal{B}_{\boldsymbol{M}}(\boldsymbol{w}^\star;\lambda_{\max}(\mathcal{L}\boldsymbol{w}^\star))$. We first construct an intermediate estimator,

$$\boldsymbol{w}_t = \boldsymbol{w}^\star + t(\hat{\boldsymbol{w}} - \boldsymbol{w}^\star), \tag{40}$$

where $t$ is taken such that $\|\boldsymbol{w}_t - \boldsymbol{w}^\star\|_{\boldsymbol{M}} = \lambda_{\max}(\mathcal{L}\boldsymbol{w}^\star)$ if $\|\hat{\boldsymbol{w}} - \boldsymbol{w}^\star\|_{\boldsymbol{M}} > \lambda_{\max}(\mathcal{L}\boldsymbol{w}^\star)$, and $t = 1$ otherwise. Hence $\|\boldsymbol{w}_t - \boldsymbol{w}^\star\|_{\boldsymbol{M}} \leq \lambda_{\max}(\mathcal{L}\boldsymbol{w}^\star)$ always holds and $t \in [0,1]$. One further has $\boldsymbol{w}_t \in \mathcal{S}_{\boldsymbol{w}}$ because both $\boldsymbol{w}^\star, \hat{\boldsymbol{w}} \in \mathcal{S}_{\boldsymbol{w}}$ and $\mathcal{S}_{\boldsymbol{w}}$ is a convex set as shown in (4). Therefore, we conclude that $\boldsymbol{w}_t \in \mathcal{B}_{\boldsymbol{M}}(\boldsymbol{w}^\star;\lambda_{\max}(\mathcal{L}\boldsymbol{w}^\star))$. Applying Lemma 1.9 with $\boldsymbol{w}_1 = \boldsymbol{w}_t$, $\boldsymbol{w}_2 = \boldsymbol{w}^*$ and $r = \lambda_{\max}(\mathcal{L}\boldsymbol{w}^\star)$ yields

$$t\langle -\mathcal{L}^*\left(\mathcal{L}\boldsymbol{w}_t+\boldsymbol{J}\right)^{-1} + \mathcal{L}^*\left(\mathcal{L}\boldsymbol{w}^\star+\boldsymbol{J}\right)^{-1},\,\hat{\boldsymbol{w}}-\boldsymbol{w}^*\rangle \geq (2\lambda_{\max}(\mathcal{L}\boldsymbol{w}^\star))^{-2}\|\mathcal{L}\boldsymbol{w}_t - \mathcal{L}\boldsymbol{w}^*\|_{\mathrm{F}}^2. \tag{41}$$

Let $q(a) = -\log\det\left(\mathcal{L}(\boldsymbol{w}^\star + a(\hat{\boldsymbol{w}}-\boldsymbol{w}^\star)) + \boldsymbol{J}\right) + a\langle\mathcal{L}^*(\mathcal{L}\boldsymbol{w}^\star+\boldsymbol{J})^{-1},\hat{\boldsymbol{w}}-\boldsymbol{w}^\star\rangle$ and $a \in [0,1]$. One has

$$q'(a) = \langle -\mathcal{L}^*\left(\mathcal{L}\boldsymbol{w}_a+\boldsymbol{J}\right)^{-1} + \mathcal{L}^*\left(\mathcal{L}\boldsymbol{w}^\star+\boldsymbol{J}\right)^{-1},\,\hat{\boldsymbol{w}}-\boldsymbol{w}^\star\rangle, \tag{42}$$

and

$$q''(a) = \left\langle\mathcal{L}^*\left((\mathcal{L}\boldsymbol{w}_a+\boldsymbol{J})^{-1}(\mathcal{L}\hat{\boldsymbol{w}}-\mathcal{L}\boldsymbol{w}^\star)(\mathcal{L}\boldsymbol{w}_a+\boldsymbol{J})^{-1}\right),\,\hat{\boldsymbol{w}}-\boldsymbol{w}^\star\right\rangle = \mathrm{tr}\,(\boldsymbol{ABAB})\,,$$

where $\boldsymbol{w}_a = \boldsymbol{w}^\star + a(\hat{\boldsymbol{w}}-\boldsymbol{w}^\star)$, $\boldsymbol{A} = (\mathcal{L}\boldsymbol{w}_a+\boldsymbol{J})^{-1}$ and $\boldsymbol{B} = (\mathcal{L}\hat{\boldsymbol{w}}-\mathcal{L}\boldsymbol{w}^\star)$. Note that $\boldsymbol{A}$ is symmetric and positive definite because $\boldsymbol{w}_t \in \mathcal{S}_{\boldsymbol{w}}$ and $\boldsymbol{B}$ is symmetric. Let $\boldsymbol{C} = \boldsymbol{AB}$. According to Theorem 1 in [2], all the eigenvalues of a matrix $\boldsymbol{X} \in \mathbb{R}^{p\times p}$ are real if there exists a symmetric and positive definite matrix $\boldsymbol{Y} \in \mathbb{R}^{p\times p}$ such that $\boldsymbol{XY}$ are symmetric. It is easy to check that the matrix $\boldsymbol{CA}$ is symmetric with $\boldsymbol{A}$ symmetric and positive definite, and thus all the eigenvalues of $\boldsymbol{C}$ are real. Suppose $\lambda_1,\ldots,\lambda_p$ are the eigenvalues of $\boldsymbol{C}$. Then the eigenvalues of $\boldsymbol{CC}$ are $\lambda_1^2,\ldots,\lambda_p^2$. Therefore, $q''(a) = \sum_{i=1}^p \lambda_i^2 \geq 0$, implying that $q'(a)$ is non-decreasing with the increase of $a$. Then one obtains

$$t\langle\mathcal{L}^*\left(\mathcal{L}\boldsymbol{w}^\star+\boldsymbol{J}\right)^{-1} - \mathcal{L}^*\left(\mathcal{L}\hat{\boldsymbol{w}}+\boldsymbol{J}\right)^{-1},\hat{\boldsymbol{w}}-\boldsymbol{w}^\star\rangle = tq'(1) \geq tq'(t) \geq (2\lambda_{\max}(\mathcal{L}\boldsymbol{w}^\star))^{-2}\|\mathcal{L}\boldsymbol{w}_t - \mathcal{L}\boldsymbol{w}^*\|_{\mathrm{F}}^2. \tag{43}$$

where the first inequality holds because $q'(a)$ is non-decreasing and $t \leq 1$, and the second inequality follows from (41).

The Lagrangian of the optimization (39) is

$$L(\boldsymbol{w},\boldsymbol{\nu}) = -\log\det(\mathcal{L}\boldsymbol{w}+\boldsymbol{J}) + \mathrm{tr}\,(\mathcal{L}\boldsymbol{w}\boldsymbol{S}) + \boldsymbol{z}^\top\boldsymbol{w} - \boldsymbol{v}^\top\boldsymbol{w},$$

where $\boldsymbol{v}$ is a KKT multiplier. Let $(\hat{\boldsymbol{w}},\hat{\boldsymbol{v}})$ be the primal and dual optimal point. Then $(\hat{\boldsymbol{w}},\hat{\boldsymbol{v}})$ must satisfy the KKT conditions as below

$$-\mathcal{L}^*\left((\mathcal{L}\hat{\boldsymbol{w}}+\boldsymbol{J})^{-1}\right) + \mathcal{L}^*\boldsymbol{S} + \boldsymbol{z} - \hat{\boldsymbol{v}} = \boldsymbol{0}; \tag{44}$$

$$\hat{w}_i\hat{v}_i = 0,\ \text{for } i = 1,\ldots,p(p-1)/2; \tag{45}$$

$$\hat{\boldsymbol{w}} \geq \boldsymbol{0},\ \hat{\boldsymbol{v}} \geq \boldsymbol{0}; \tag{46}$$

According to (44), one has

$$\left\langle -\mathcal{L}^*\big(\mathcal{L}\hat{\boldsymbol{w}}+\boldsymbol{J}\big)^{-1}+\mathcal{L}^*\boldsymbol{S},\,\hat{\boldsymbol{w}}-\boldsymbol{w}^\star\right\rangle = \left\langle \hat{\boldsymbol{v}}-\boldsymbol{z},\hat{\boldsymbol{w}}-\boldsymbol{w}^\star\right\rangle. \tag{47}$$

Substituting (47) into (43) yields

$$\|\mathcal{L}\boldsymbol{w}_t-\mathcal{L}\boldsymbol{w}^\star\|_{\mathrm{F}}^2 \le 4t\lambda_{\max}^2(\mathcal{L}\boldsymbol{w}^\star)\Big(\langle\hat{\boldsymbol{v}}-\boldsymbol{z},\hat{\boldsymbol{w}}-\boldsymbol{w}^\star\rangle+\big\langle\mathcal{L}^*\big((\mathcal{L}\boldsymbol{w}^\star+\boldsymbol{J})^{-1}-\boldsymbol{S}\big),\hat{\boldsymbol{w}}-\boldsymbol{w}^\star\big\rangle\Big)$$

$$= 4t\lambda_{\max}^2(\mathcal{L}\boldsymbol{w}^\star)\Big(\underbrace{\langle\hat{\boldsymbol{v}},\hat{\boldsymbol{w}}-\boldsymbol{w}^\star\rangle}_{\text{term I}}-\underbrace{\langle\boldsymbol{z},\hat{\boldsymbol{w}}-\boldsymbol{w}^\star\rangle}_{\text{term II}}+\underbrace{\big\langle\mathcal{L}^*\big((\mathcal{L}\boldsymbol{w}^\star+\boldsymbol{J})^{-1}-\boldsymbol{S}\big),\hat{\boldsymbol{w}}-\boldsymbol{w}^\star\big\rangle}_{\text{term III}}\Big). \tag{48}$$

Next we will bound term I, II and III, respectively. The term I can be directly bounded by

$$\langle\hat{\boldsymbol{v}},\hat{\boldsymbol{w}}-\boldsymbol{w}^\star\rangle = -\langle\hat{\boldsymbol{v}},\boldsymbol{w}^\star\rangle \le 0. \tag{49}$$

where the equality follows from (45) and the inequality follows from $\hat{\boldsymbol{v}}\ge\boldsymbol{0}$ in (46) and $\boldsymbol{w}^\star\ge\boldsymbol{0}$.

For term II, we separate the support of $\boldsymbol{z}$ into two parts, $\mathcal{S}^\star$ and its complementary set $\{\mathcal{S}^\star\}^c$, where $\mathcal{S}^\star$ is the support of $\boldsymbol{w}^\star$ with $|\mathcal{S}^\star|\le s$. Take a set $\mathcal{E}$ satisfying $\mathcal{S}^\star\subseteq\mathcal{E}$ and $|\mathcal{E}|\le 2s$. A simple algebra yields

$$\langle\boldsymbol{z},\hat{\boldsymbol{w}}-\boldsymbol{w}^\star\rangle = \langle\boldsymbol{z}_{\mathcal{S}^\star},(\hat{\boldsymbol{w}}-\boldsymbol{w}^\star)_{\mathcal{S}^\star}\rangle + \langle\boldsymbol{z}_{\{\mathcal{S}^\star\}^c},(\hat{\boldsymbol{w}}-\boldsymbol{w}^\star)_{\{\mathcal{S}^\star\}^c}\rangle$$

$$= \langle\boldsymbol{z}_{\mathcal{S}^\star},(\hat{\boldsymbol{w}}-\boldsymbol{w}^\star)_{\mathcal{S}^\star}\rangle + \langle\boldsymbol{z}_{\{\mathcal{S}^\star\}^c},\hat{\boldsymbol{w}}_{\{\mathcal{S}^\star\}^c}\rangle$$

$$\ge -\|\boldsymbol{z}_{\mathcal{S}^\star}\|\,\|(\hat{\boldsymbol{w}}-\boldsymbol{w}^\star)_{\mathcal{S}^\star}\| + \langle\boldsymbol{z}_{\{\mathcal{S}^\star\}^c},\hat{\boldsymbol{w}}_{\{\mathcal{S}^\star\}^c}\rangle$$

$$\ge -\|\boldsymbol{z}_{\mathcal{S}^\star}\|\,\|(\hat{\boldsymbol{w}}-\boldsymbol{w}^\star)_{\mathcal{S}^\star}\| + \langle\boldsymbol{z}_{\mathcal{E}^c},\hat{\boldsymbol{w}}_{\mathcal{E}^c}\rangle, \tag{50}$$

where the first inequality follows from Cauchy–Schwarz inequality and the second inequality follows from $\boldsymbol{z}\ge\boldsymbol{0}$, $\hat{\boldsymbol{w}}\ge\boldsymbol{0}$ and $\mathcal{E}^c\subseteq\{\mathcal{S}^\star\}^c$.

For term III, we separate the support of $\mathcal{L}^*\big((\mathcal{L}\boldsymbol{w}^\star+\boldsymbol{J})^{-1}-\boldsymbol{S}\big)$ into parts, $\mathcal{E}$ and $\mathcal{E}^c$. Then one has

$$\big\langle\mathcal{L}^*\big((\mathcal{L}\boldsymbol{w}^\star+\boldsymbol{J})^{-1}-\boldsymbol{S}\big),\hat{\boldsymbol{w}}-\boldsymbol{w}^\star\big\rangle = \big\langle\big(\mathcal{L}^*\big((\mathcal{L}\boldsymbol{w}^\star+\boldsymbol{J})^{-1}-\boldsymbol{S}\big)\big)_{\mathcal{E}},(\hat{\boldsymbol{w}}-\boldsymbol{w}^\star)_{\mathcal{E}}\big\rangle$$

$$+ \big\langle\big(\mathcal{L}^*\big((\mathcal{L}\boldsymbol{w}^\star+\boldsymbol{J})^{-1}-\boldsymbol{S}\big)\big)_{\mathcal{E}^c},(\hat{\boldsymbol{w}}-\boldsymbol{w}^\star)_{\mathcal{E}^c}\big\rangle$$

$$\le \Big\|\big(\mathcal{L}^*\big((\mathcal{L}\boldsymbol{w}^\star+\boldsymbol{J})^{-1}-\boldsymbol{S}\big)\big)_{\mathcal{E}}\Big\|\,\|(\hat{\boldsymbol{w}}-\boldsymbol{w}^\star)_{\mathcal{E}}\|$$

$$+ \big\langle\big(\mathcal{L}^*\big((\mathcal{L}\boldsymbol{w}^\star+\boldsymbol{J})^{-1}-\boldsymbol{S}\big)\big)_{\mathcal{E}^c},\hat{\boldsymbol{w}}_{\mathcal{E}^c}\big\rangle. \tag{51}$$

Substituting (49), (50) and (51) into (48) yields

$$\|\mathcal{L}\boldsymbol{w}_t-\mathcal{L}\boldsymbol{w}^\star\|_{\mathrm{F}}^2 \le 4t\lambda_{\max}^2(\mathcal{L}\boldsymbol{w}^\star)\Big(\Big\|\big(\mathcal{L}^*\big((\mathcal{L}\boldsymbol{w}^\star+\boldsymbol{J})^{-1}-\boldsymbol{S}\big)\big)_{\mathcal{E}}\Big\|\,\|(\hat{\boldsymbol{w}}-\boldsymbol{w}^\star)_{\mathcal{E}}\| \tag{52}$$

$$\big\langle\big(\mathcal{L}^*\big((\mathcal{L}\boldsymbol{w}^\star+\boldsymbol{J})^{-1}-\boldsymbol{S}\big)\big)_{\mathcal{E}^c}-\boldsymbol{z}_{\mathcal{E}^c},\hat{\boldsymbol{w}}_{\mathcal{E}^c}\big\rangle + \|\boldsymbol{z}_{\mathcal{S}^\star}\|\,\|(\hat{\boldsymbol{w}}-\boldsymbol{w}^\star)_{\mathcal{S}^\star}\|\Big).$$

Notice that the inequality

$$\big\langle\big(\mathcal{L}^*\big((\mathcal{L}\boldsymbol{w}^\star+\boldsymbol{J})^{-1}-\boldsymbol{S}\big)\big)_{\mathcal{E}^c}-\boldsymbol{z}_{\mathcal{E}^c},\hat{\boldsymbol{w}}_{\mathcal{E}^c}\big\rangle \le 0, \tag{53}$$

holds because $\hat{\boldsymbol{w}}\ge\boldsymbol{0}$, $\|\boldsymbol{z}_{\mathcal{E}^c}\|_{\min}\ge\lambda/2$ and

$$\Big\|\big(\mathcal{L}^*\big((\mathcal{L}\boldsymbol{w}^\star+\boldsymbol{J})^{-1}-\boldsymbol{S}\big)\big)_{\mathcal{E}^c}\Big\|_{\max} \le \Big\|\mathcal{L}^*\big((\mathcal{L}\boldsymbol{w}^\star+\boldsymbol{J})^{-1}-\boldsymbol{S}\big)\Big\|_{\max} \le \frac{\lambda}{2},$$

where the last inequality follows from the conditions in Lemma 1.4. Combining (52) and (53) together yields

$$\|\mathcal{L}\boldsymbol{w}_t-\mathcal{L}\boldsymbol{w}^\star\|_{\mathrm{F}}^2 \le 4t\lambda_{\max}^2(\mathcal{L}\boldsymbol{w}^\star)\Big(\|\boldsymbol{z}_{\mathcal{S}^\star}\|\,\|(\hat{\boldsymbol{w}}-\boldsymbol{w}^\star)_{\mathcal{S}^\star}\| + \Big\|\big(\mathcal{L}^*\big((\mathcal{L}\boldsymbol{w}^\star+\boldsymbol{J})^{-1}-\boldsymbol{S}\big)\big)_{\mathcal{E}}\Big\|\,\|(\hat{\boldsymbol{w}}-\boldsymbol{w}^\star)_{\mathcal{E}}\|\Big)$$

$$\le 4t\lambda_{\max}^2(\mathcal{L}\boldsymbol{w}^\star)\Big(\|\boldsymbol{z}_{\mathcal{S}^\star}\| + \Big\|\big(\mathcal{L}^*\big((\mathcal{L}\boldsymbol{w}^\star+\boldsymbol{J})^{-1}-\boldsymbol{S}\big)\big)_{\mathcal{E}}\Big\|\Big)\|\hat{\boldsymbol{w}}-\boldsymbol{w}^\star\|, \tag{54}$$

where the last inequality follows from $\|\hat{\boldsymbol{w}} - \boldsymbol{w}^\star\| \geq \|(\hat{\boldsymbol{w}} - \boldsymbol{w}^\star)_{\mathcal{E}}\| \geq \|(\hat{\boldsymbol{w}} - \boldsymbol{w}^\star)_{\mathcal{S}^\star}\|$.

On the other hand, one has

$$\|\mathcal{L}\boldsymbol{w}_t - \mathcal{L}\boldsymbol{w}^\star\|_{\mathrm{F}} = t\,\|\mathcal{L}\hat{\boldsymbol{w}} - \mathcal{L}\boldsymbol{w}^\star\|_{\mathrm{F}} \geq t\Big(\sum_{i\neq j}\big([\mathcal{L}\hat{\boldsymbol{w}} - \mathcal{L}\boldsymbol{w}^\star]_{ij}\big)^2\Big)^{\frac{1}{2}} = \sqrt{2}t\,\|\hat{\boldsymbol{w}} - \boldsymbol{w}^\star\|. \tag{55}$$

Combining (54) and (55) together yields

$$\|\mathcal{L}\boldsymbol{w}_t - \mathcal{L}\boldsymbol{w}^\star\|_{\mathrm{F}} \leq 2\sqrt{2}\lambda_{\max}^2(\mathcal{L}\boldsymbol{w}^\star)\Big(\|\boldsymbol{z}_{\mathcal{S}^\star}\| + \big\|\big(\mathcal{L}^*\big((\mathcal{L}\boldsymbol{w}^\star + \boldsymbol{J})^{-1} - \boldsymbol{S}\big)\big)_{\mathcal{E}}\big\|\Big). \tag{56}$$

Recall that $\|\boldsymbol{z}\|_{\max} \leq \lambda$ and $|\mathcal{S}^\star| \leq s$. Thus one has

$$\|\boldsymbol{z}_{\mathcal{S}^\star}\| \leq \sqrt{s}\lambda. \tag{57}$$

One also has

$$\big\|\big(\mathcal{L}^*\big((\mathcal{L}\boldsymbol{w}^\star + \boldsymbol{J})^{-1} - \boldsymbol{S}\big)\big)_{\mathcal{E}}\big\| \leq \Big(|\mathcal{E}|\,\big\|\mathcal{L}^*\big((\mathcal{L}\boldsymbol{w}^\star + \boldsymbol{J})^{-1} - \boldsymbol{S}\big)\big\|_{\max}^2\Big)^{\frac{1}{2}} \leq \frac{\sqrt{2}}{2}\sqrt{s}\lambda. \tag{58}$$

Substituting (57) and (58) into (56) yields

$$\|\mathcal{L}\boldsymbol{w}_t - \mathcal{L}\boldsymbol{w}^\star\|_{\mathrm{F}} \leq 2(\sqrt{2}+1)\lambda_{\max}^2(\mathcal{L}\boldsymbol{w}^\star)\sqrt{s}\lambda < \lambda_{\max}(\mathcal{L}\boldsymbol{w}^\star), \tag{59}$$

which implies that $t = 1$ in (40), i.e., $\boldsymbol{w}_t = \hat{\boldsymbol{w}}$. The last inequality is established by plugging $\lambda = \sqrt{4\alpha c_0^{-1}\log p/n}$ with $n \geq 94\alpha c_0^{-1}\lambda_{\max}^2(\mathcal{L}\boldsymbol{w}^\star)s\log p$. Therefore, we conclude that

$$\|\mathcal{L}\hat{\boldsymbol{w}} - \mathcal{L}\boldsymbol{w}^\star\|_{\mathrm{F}} \leq 2\sqrt{2}\lambda_{\max}^2(\mathcal{L}\boldsymbol{w}^\star)\Big(\|\boldsymbol{z}_{\mathcal{S}^\star}\| + \big\|\big(\mathcal{L}^*\big((\mathcal{L}\boldsymbol{w}^\star + \boldsymbol{J})^{-1} - \boldsymbol{S}\big)\big)_{\mathcal{E}}\big\|\Big) \leq 2(1+\sqrt{2})\lambda_{\max}^2(\mathcal{L}\boldsymbol{w}^\star)\sqrt{s}\lambda,$$

where the first inequality is established by (56) with $t = 1$, and the second inequality is established by plugging (57) and (58). □

## 2.5 Proof of Lemma 1.5

*Proof.* Recall that $\mathcal{E}^{(k)} = \{\mathcal{S}^\star \cup \mathcal{S}^{(k)}\}$ and $\mathcal{S}^{(k)} = \{i \in [p(p-1)/2] \mid \hat{w}_i^{(k-1)} \geq b\}$ with $b = (2+\sqrt{2})\lambda_{\max}^2(\mathcal{L}\boldsymbol{w}^\star)\lambda$.

We prove $|\mathcal{E}^{(k)}| \leq 2s$ holds by induction. For $k = 1$, $\forall i \notin \mathrm{supp}^+(\hat{\boldsymbol{w}}^{(0)})$, i.e., $w_i^{(0)} \leq 0$, one has $w_i^{(0)} < b$, implying that $i \notin \mathcal{S}^{(1)}$. In other words, $\mathcal{S}^{(1)} \subseteq \mathrm{supp}^+(\hat{\boldsymbol{w}}^{(0)})$. Then one has

$$|\mathcal{E}^{(1)}| = |\mathcal{S}^\star \cup \mathcal{S}^{(1)}| \leq |\mathcal{S}^\star \cup \mathrm{supp}^+(\hat{\boldsymbol{w}}^{(0)})| \leq s + s = 2s.$$

Therefore, $|\mathcal{E}^{(k)}| \leq 2s$ holds for $k = 1$.

Assume $|\mathcal{E}^{(k-1)}| \leq 2s$ holds for some $k \geq 2$. We separate the set $\mathcal{E}^{(k)}$ into two parts, $\mathcal{S}^\star$ and $\mathcal{S}^{(k)}\backslash\mathcal{S}^\star$. For any $i \in \mathcal{S}^{(k)}\backslash\mathcal{S}^\star$, one has $\hat{w}_i^{(k-1)} \geq b$, and further obtains

$$\sqrt{|\mathcal{S}^{(k)}\backslash\mathcal{S}^\star|} \leq \sqrt{\sum_{i\in\mathcal{S}^{(k)}\backslash\mathcal{S}^\star}\Big(\frac{\hat{w}_i^{(k-1)}}{b}\Big)^2} = \frac{\big\|\hat{\boldsymbol{w}}_{\mathcal{S}^{(k)}\backslash\mathcal{S}^\star}^{(k-1)}\big\|}{b} = \frac{\big\|(\hat{\boldsymbol{w}}^{(k-1)} - \boldsymbol{w}^\star)_{\mathcal{S}^{(k)}\backslash\mathcal{S}^\star}\big\|}{b}$$

$$\leq \frac{\big\|\hat{\boldsymbol{w}}_{\mathcal{E}^{(k)}}^{(k-1)} - \boldsymbol{w}^\star\big\|}{b} \leq \frac{\big\|\hat{\boldsymbol{w}}^{(k-1)} - \boldsymbol{w}^\star\big\|}{b}. \tag{60}$$

Let $\boldsymbol{z}^{(k-2)}$ satisfy $z_i^{(k-2)} = h_\lambda'(\hat{w}_i^{(k-2)})$, $i \in [p(p-1)/2]$. By Assumption 3.5, one has $z_i^{(k-2)} \in [0, \lambda]$ for $i \in [p(p-1)/2]$. For any $i \in \{\mathcal{S}^{(k-1)}\}^c$, one further has

$$z_i^{(k-2)} = h_\lambda'(\hat{w}_i^{(k-2)}) \geq h_\lambda'(b) \geq \frac{\lambda}{2}, \tag{61}$$

where the first inequality holds because $\hat{w}_i^{(k-2)} < b$ for any $i \in \left\{\mathcal{S}^{(k-1)}\right\}^c$ by the definition of $\mathcal{S}^{(k-1)}$, and $h'_\lambda$ is non-increasing by Assumption 3.5; the second inequality follows from Assumption 3.5. Therefore, one obtains

$$\left\|\boldsymbol{z}_{\{\mathcal{E}^{(k-1)}\}^c}^{(k-2)}\right\|_{\min} \geq \left\|\boldsymbol{z}_{\{\mathcal{S}^{(k-1)}\}^c}^{(k-2)}\right\|_{\min} \geq \lambda/2,$$

where the first inequality follows from $\left\{\mathcal{E}^{(k-1)}\right\}^c \subseteq \left\{\mathcal{S}^{(k-1)}\right\}^c$ and the second inequality follows from (61). One also has $|\mathcal{E}^{(k-1)}| \leq 2s$ and $\mathcal{S}^\star \subseteq \mathcal{E}^{(k-1)}$. Hence we can apply Lemma 1.4 with $\mathcal{E} = \mathcal{E}^{(k-1)}$ and $\boldsymbol{z} = \boldsymbol{z}^{(k-2)}$ and obtain

$$\left\|\hat{\boldsymbol{w}}^{(k-1)} - \boldsymbol{w}^\star\right\| \leq \frac{\sqrt{2}}{2}\left\|\mathcal{L}\hat{\boldsymbol{w}}^{(k-1)} - \mathcal{L}\boldsymbol{w}^\star\right\|_{\mathrm{F}} \leq (2+\sqrt{2})\lambda_{\max}^2(\mathcal{L}\boldsymbol{w}^\star)\sqrt{s}\lambda, \tag{62}$$

where the first inequality holds with the proof similar to (69). Combining (60) and (62) together yields

$$\sqrt{|\mathcal{S}^{(k)}\backslash\mathcal{S}^\star|} \leq \frac{(2+\sqrt{2})\lambda_{\max}^2(\mathcal{L}\boldsymbol{w}^\star)\sqrt{s}\lambda}{b} = \sqrt{s},$$

where the last equality follows from $b = (2+\sqrt{2})\lambda_{\max}^2(\mathcal{L}\boldsymbol{w}^\star)\lambda$. Therefore, one gets

$$|\mathcal{E}^{(k)}| = |\mathcal{S}^\star \cup \mathcal{S}^{(k)}\backslash\mathcal{S}^\star| = |\mathcal{S}^\star| + |\mathcal{S}^{(k)}\backslash\mathcal{S}^\star| \leq s + s = 2s,$$

completing the induction. □

## 2.6  Proof of Lemma 1.6

*Proof.* For any $k \geq 1$, one has $|\mathcal{E}^{(k)}| \leq 2s$ by Lemma 1.5. According to the definition of $\mathcal{E}^{(k)}$ in (7), one has $\mathcal{S}^\star \subseteq \mathcal{E}^{(k)}$. Let $z_i^{(k-1)} = h'_\lambda(\hat{w}_i^{(k-1)})$, $i \in [p(p-1)/2]$. By Assumption 3.5, one has $z_i^{(k-1)} \in [0, \lambda]$ for $i \in [p(p-1)/2]$. For any $i \in \left\{\mathcal{S}^{(k)}\right\}^c$, one has

$$z_i^{(k-1)} = h'_\lambda(\hat{w}_i^{(k-1)}) \geq h'_\lambda(b) \geq \frac{\lambda}{2},$$

where the first inequality holds because $\hat{w}_i^{(k-1)} < b$ for any $i \in \left\{\mathcal{S}^{(k)}\right\}^c$ by the definition of $\mathcal{S}^{(k)}$ in (7), and $h'_\lambda$ is non-increasing by Assumption 3.5; the second inequality follows from Assumption 3.5. Therefore, $\left\|\boldsymbol{z}_{\{\mathcal{E}^{(k)}\}^c}^{(k-1)}\right\|_{\min} \geq \left\|\boldsymbol{z}_{\{\mathcal{S}^{(k)}\}^c}^{(k-1)}\right\|_{\min} \geq \lambda/2$. Applying Lemma 1.4 with $\mathcal{E} = \mathcal{E}^{(k)}$ and $\boldsymbol{z} = \boldsymbol{z}^{(k-1)}$ yields

$$\left\|\mathcal{L}\hat{\boldsymbol{w}}^{(k)} - \mathcal{L}\boldsymbol{w}^\star\right\|_{\mathrm{F}} \leq 2\sqrt{2}\lambda_{\max}^2(\mathcal{L}\boldsymbol{w}^\star)\left(\left\|\boldsymbol{z}_{\mathcal{S}^\star}^{(k-1)}\right\| + \left\|\left(\mathcal{L}^*\left((\mathcal{L}\boldsymbol{w}^\star + \boldsymbol{J})^{-1} - \boldsymbol{S}\right)\right)_{\mathcal{E}^{(k)}}\right\|\right). \tag{63}$$

We will show that the term $\left\|\boldsymbol{z}_{\mathcal{S}^\star}^{(k-1)}\right\|$ in (63) can be bounded in terms of $\left\|\hat{\boldsymbol{w}}^{(k-1)} - \boldsymbol{w}^\star\right\|$. For any given $\boldsymbol{w} \in \mathbb{R}^{p(p-1)/2}$, if $|w_i^\star - w_i| \geq b$, then one has

$$0 \leq h'_\lambda(w_i) \leq \lambda \leq \lambda b^{-1}|w_i^\star - w_i|,$$

where $b = (2+\sqrt{2})\lambda_{\max}^2(\mathcal{L}\boldsymbol{w}^\star)\lambda$, and the first two inequalities follows from Assumption 3.5. Otherwise, one has $w_i^\star - w_i \leq |w_i^\star - w_i| \leq b$, then $0 \leq h'_\lambda(w_i) \leq h'_\lambda(w_i^\star - b)$ because $h'_\lambda$ is non-increasing. Totally, one has

$$h'_\lambda(w_i) \leq \lambda b^{-1}|w_i^\star - w_i| + h'_\lambda(w_i^\star - b), \quad \forall i \in [p(p-1)/2]. \tag{64}$$

Collecting the indices $i \in \mathcal{S}^\star$ together and applying (64) with $\boldsymbol{w} = \hat{\boldsymbol{w}}^{(k-1)}$ yields

$$\left\|\boldsymbol{z}_{\mathcal{S}^\star}^{(k-1)}\right\| = \left\|h'_\lambda(\hat{\boldsymbol{w}}_{\mathcal{S}^\star}^{(k-1)})\right\| \leq \frac{\lambda}{b}\left\|\hat{\boldsymbol{w}}_{\mathcal{S}^\star}^{(k-1)} - \boldsymbol{w}_{\mathcal{S}^\star}^\star\right\| + \left\|h'_\lambda(\boldsymbol{w}_{\mathcal{S}^\star}^\star - \boldsymbol{b})\right\| \leq \frac{\lambda}{b}\left\|\hat{\boldsymbol{w}}^{(k-1)} - \boldsymbol{w}^\star\right\| + \left\|h'_\lambda(\boldsymbol{w}_{\mathcal{S}^\star}^\star - \boldsymbol{b})\right\|, \tag{65}$$

where $h'_\lambda(\hat{\boldsymbol{w}}^{(k-1)}_{\mathcal{S}^\star}) = (h'_\lambda(\hat{w}_i^{(k-1)}))_{i\in\mathcal{S}^\star}$, and $\boldsymbol{b} = [b,\ldots,b]^\top$ is a constant vector. Combining (63) and (65) together yields

$$\left\|\mathcal{L}\hat{\boldsymbol{w}}^{(k)} - \mathcal{L}\boldsymbol{w}^\star\right\|_{\mathrm{F}} \le 2\sqrt{2}\lambda_{\max}^2(\mathcal{L}\boldsymbol{w}^\star)\left(\left\|\left(\mathcal{L}^*\left((\mathcal{L}\boldsymbol{w}^\star + \boldsymbol{J})^{-1} - \boldsymbol{S}\right)\right)_{\mathcal{E}^{(k)}}\right\| + \left\|h'_\lambda(\boldsymbol{w}^\star_{\mathcal{S}^\star} - \boldsymbol{b})\right\|\right)$$
$$+ 2\sqrt{2}\frac{\lambda}{b}\lambda_{\max}^2(\mathcal{L}\boldsymbol{w}^\star)\left\|\hat{\boldsymbol{w}}^{(k-1)} - \boldsymbol{w}^\star\right\|. \qquad (66)$$

By separating the set $\mathcal{E}^{(k)}$ into two parts, $\mathcal{S}^\star$ and $\mathcal{S}^{(k)}\backslash\mathcal{S}^\star$, one has

$$\left\|\left(\mathcal{L}^*\left((\mathcal{L}\boldsymbol{w}^\star + \boldsymbol{J})^{-1} - \boldsymbol{S}\right)\right)_{\mathcal{E}^{(k)}}\right\| \le \left\|\left(\mathcal{L}^*\left((\mathcal{L}\boldsymbol{w}^\star + \boldsymbol{J})^{-1} - \boldsymbol{S}\right)\right)_{\mathcal{S}^\star}\right\| + \left\|\left(\mathcal{L}^*\left((\mathcal{L}\boldsymbol{w}^\star + \boldsymbol{J})^{-1} - \boldsymbol{S}\right)\right)_{\mathcal{S}^{(k)}\backslash\mathcal{S}^\star}\right\|.$$

We will show that the term $\left\|\left(\mathcal{L}^*\left((\mathcal{L}\boldsymbol{w}^\star + \boldsymbol{J})^{-1} - \boldsymbol{S}\right)\right)_{\mathcal{S}^{(k)}\backslash\mathcal{S}^\star}\right\|$ can be bounded in terms of $\left\|\hat{\boldsymbol{w}}^{(k-1)} - \boldsymbol{w}^\star\right\|$.

$$\left\|\left(\mathcal{L}^*\left((\mathcal{L}\boldsymbol{w}^\star + \boldsymbol{J})^{-1} - \boldsymbol{S}\right)\right)_{\mathcal{S}^{(k)}\backslash\mathcal{S}^\star}\right\| \le \sqrt{|\mathcal{E}^{(k)}\backslash\mathcal{S}^\star|}\left\|\left(\mathcal{L}^*\left((\mathcal{L}\boldsymbol{w}^\star + \boldsymbol{J})^{-1} - \boldsymbol{S}\right)\right)_{\mathcal{S}^{(k)}\backslash\mathcal{S}^\star}\right\|_{\max}$$
$$\le \sqrt{|\mathcal{E}^{(k)}\backslash\mathcal{S}^\star|}\left\|\mathcal{L}^*\left((\mathcal{L}\boldsymbol{w}^\star + \boldsymbol{J})^{-1} - \boldsymbol{S}\right)\right\|_{\max}$$
$$\le \frac{1}{b}\left\|\hat{\boldsymbol{w}}^{(k-1)} - \boldsymbol{w}^\star\right\|\left\|\mathcal{L}^*\left((\mathcal{L}\boldsymbol{w}^\star + \boldsymbol{J})^{-1} - \boldsymbol{S}\right)\right\|_{\max}$$
$$\le \frac{\lambda}{2b}\left\|\hat{\boldsymbol{w}}^{(k-1)} - \boldsymbol{w}^\star\right\|,$$

where the last second equality follows from (60). Thus one has

$$\left\|\left(\mathcal{L}^*\left((\mathcal{L}\boldsymbol{w}^\star + \boldsymbol{J})^{-1} - \boldsymbol{S}\right)\right)_{\mathcal{E}^{(k)}}\right\| \le \left\|\left(\mathcal{L}^*\left((\mathcal{L}\boldsymbol{w}^\star + \boldsymbol{J})^{-1} - \boldsymbol{S}\right)\right)_{\mathcal{S}^\star}\right\| + \frac{\lambda}{2b}\left\|\hat{\boldsymbol{w}}^{(k-1)} - \boldsymbol{w}^\star\right\|. \qquad (67)$$

Substituting (67) into (66) yields

$$\left\|\mathcal{L}\hat{\boldsymbol{w}}^{(k)} - \mathcal{L}\boldsymbol{w}^\star\right\|_{\mathrm{F}} \le 2\sqrt{2}\lambda_{\max}^2(\mathcal{L}\boldsymbol{w}^\star)\left(\left\|\left(\mathcal{L}^*\left((\mathcal{L}\boldsymbol{w}^\star + \boldsymbol{J})^{-1} - \boldsymbol{S}\right)\right)_{\mathcal{S}^\star}\right\| + \left\|h'_\lambda(\boldsymbol{w}^\star_{\mathcal{S}^\star} - \boldsymbol{b})\right\|\right)$$
$$+ 3\sqrt{2}\frac{\lambda}{b}\lambda_{\max}^2(\mathcal{L}\boldsymbol{w}^\star)\left\|\hat{\boldsymbol{w}}^{(k-1)} - \boldsymbol{w}^\star\right\|$$
$$= 2\sqrt{2}\lambda_{\max}^2(\mathcal{L}\boldsymbol{w}^\star)\left\|\left(\mathcal{L}^*\left((\mathcal{L}\boldsymbol{w}^\star + \boldsymbol{J})^{-1} - \boldsymbol{S}\right)\right)_{\mathcal{S}^\star}\right\| + \frac{3\sqrt{2}}{2 + \sqrt{2}}\left\|\hat{\boldsymbol{w}}^{(k-1)} - \boldsymbol{w}^\star\right\|$$
$$\le 2\sqrt{2}\lambda_{\max}^2(\mathcal{L}\boldsymbol{w}^\star)\left\|\left(\mathcal{L}^*\left((\mathcal{L}\boldsymbol{w}^\star + \boldsymbol{J})^{-1} - \boldsymbol{S}\right)\right)_{\mathcal{S}^\star}\right\| + \frac{3}{2 + \sqrt{2}}\left\|\mathcal{L}\hat{\boldsymbol{w}}^{(k-1)} - \mathcal{L}\boldsymbol{w}^\star\right\|_{\mathrm{F}}, \qquad (68)$$

where the equality is established by plugging $b = (2+\sqrt{2})\lambda_{\max}^2(\mathcal{L}\boldsymbol{w}^\star)\lambda$ and following from $\left\|h'_\lambda(\boldsymbol{w}^\star_{\mathcal{S}^\star} - \boldsymbol{b})\right\| = 0$ because $\left\|\boldsymbol{w}^\star_{\mathcal{S}^\star}\right\|_{\min} - b \ge \gamma\lambda$ and $h'_\lambda(x) = 0$ for any $x \ge \gamma\lambda$ following from Assumption 3.6; the last inequality follows from

$$\left\|\mathcal{L}\hat{\boldsymbol{w}}^{(k)} - \mathcal{L}\boldsymbol{w}^\star\right\|_{\mathrm{F}} = \left(2\sum_{i=1}^{p(p-1)/2}(\hat{w}_i^{(k)} - w_i^\star)^2 + \sum_{j=1}^{p}([\mathcal{L}\hat{\boldsymbol{w}}^{(k)} - \mathcal{L}\boldsymbol{w}^\star]_{jj})^2\right)^{\frac{1}{2}} \ge \sqrt{2}\left\|\hat{\boldsymbol{w}}^{(k)} - \boldsymbol{w}^\star\right\|. \qquad (69)$$

Similarly, one also obtains

$$\left\|\hat{\boldsymbol{w}}^{(k)} - \boldsymbol{w}^\star\right\| \le \frac{\sqrt{2}}{2}\left\|\mathcal{L}\hat{\boldsymbol{w}}^{(k)} - \mathcal{L}\boldsymbol{w}^\star\right\|_{\mathrm{F}}$$
$$\le 2\lambda_{\max}^2(\mathcal{L}\boldsymbol{w}^\star)\left\|\left(\mathcal{L}^*\left((\mathcal{L}\boldsymbol{w}^\star + \boldsymbol{J})^{-1} - \boldsymbol{S}\right)\right)_{\mathcal{S}^\star}\right\| + \frac{3}{2 + \sqrt{2}}\left\|\hat{\boldsymbol{w}}^{(k-1)} - \boldsymbol{w}^\star\right\|.$$

$\square$

## 2.7   Proof of Lemma 1.7

*Proof.* We apply Lemma 1.8 with $t = \lambda/2$ and union sum bound, then get

$$\mathbb{P}\big[\big\|\mathcal{L}^*\big((\mathcal{L}\boldsymbol{w}^\star + \boldsymbol{J})^{-1} - \boldsymbol{S}\big)\big\|_{\max} \geq \lambda/2\big] \leq p(p-1)\exp(-\frac{1}{4}c_0 n\lambda^2) \leq p^2\exp(-\frac{1}{4}c_0 n\lambda^2),$$

for any $\lambda \leq 2t_0$, where $t_0 = \big\|\mathcal{L}^*(\mathcal{L}\boldsymbol{w}^\star + \boldsymbol{J})^{-1}\big\|_{\max}$ and $c_0 = 1/(8\big\|\mathcal{L}^*(\mathcal{L}\boldsymbol{w}^\star + \boldsymbol{J})^{-1}\big\|_{\max}^2)$. Take $\lambda = \sqrt{4\alpha c_0^{-1}\log p/n}$ for some $\alpha > 2$. To guarantee $\lambda \leq 2t_0$, one takes $n \geq 8\alpha\log p$. By calculation, we establish

$$\mathbb{P}\big[\big\|\mathcal{L}^*\big((\mathcal{L}\boldsymbol{w}^\star + \boldsymbol{J})^{-1} - \boldsymbol{S}\big)\big\|_{\max} \leq \lambda/2\big] \geq 1 - p^2\exp(-\frac{1}{4}c_0 n\lambda^2) \geq 1 - 1/p^{\alpha-2},$$

completing the proof. □

## 2.8   Proof of Lemma 1.8

*Proof.* The LGMRF is a constrained GMRF model with $\mathbf{1}^\top \boldsymbol{X} = 0$ and we can follow the method called *conditioning by Kriging* [4] to sample LGMRF. More specifically, to sample $\boldsymbol{X}$ for LGMRF with precision matrix $\mathcal{L}\boldsymbol{w}^\star$, we could first sample from a unconstrained GMRF $\tilde{\boldsymbol{X}} \sim N(\mathbf{0}, (\mathcal{L}\boldsymbol{w}^\star + \boldsymbol{J})^{-1})$ and then correct for the constraint $\mathbf{1}^\top \boldsymbol{X} = 0$ by

$$\boldsymbol{X}^{(k)} = \tilde{\boldsymbol{X}}^{(k)} - \frac{1}{p}\mathbf{1}\mathbf{1}^\top \tilde{\boldsymbol{X}}^{(k)}, \quad \text{for } k = 1, \ldots, n.$$

For any $i \in [p(p-1)/2]$, one has

$$
\begin{aligned}
[\mathcal{L}^*\boldsymbol{S}]_i &= [\mathcal{L}^*(\frac{1}{n}\sum_{k=1}^{n}\boldsymbol{X}^{(k)}\boldsymbol{X}^{(k)T})]_i = \frac{1}{n}\sum_{k=1}^{n}[\mathcal{L}^*(\boldsymbol{X}^{(k)}\boldsymbol{X}^{(k)T})]_i = \frac{1}{n}\sum_{k=1}^{n}\big(X_a^{(k)} - X_b^{(k)}\big)^2 \\
&= \frac{1}{n}\sum_{k=1}^{n}\left(\Big(\tilde{X}_a^{(k)} - (\frac{1}{p}\mathbf{1}\mathbf{1}^\top\tilde{\boldsymbol{X}}^{(k)})_a\Big) - \Big(\tilde{X}_b^{(k)} - (\frac{1}{p}\mathbf{1}\mathbf{1}^\top\tilde{\boldsymbol{X}}^{(k)})_b\Big)\right)^2 \\
&= \frac{1}{n}\sum_{k=1}^{n}\big(\tilde{X}_a^{(k)} - \tilde{X}_b^{(k)}\big)^2,
\end{aligned}
\tag{70}
$$

where the indices $a$ and $b$ obey $i = a - b + (b-1)(2p-b)/2$ and $a > b$.

Let $\tilde{\boldsymbol{\Sigma}} = (\mathcal{L}\boldsymbol{w}^\star + \boldsymbol{J})^{-1}$ and thus $\tilde{\boldsymbol{X}} \sim N(\mathbf{0}, \tilde{\boldsymbol{\Sigma}})$. We first introduce two auxiliary random variables $Y_{k,i} := \tilde{X}_a^{(k)} - \tilde{X}_b^{(k)}$ and $Z_{k,i} := Y_{k,i}^2$. Together with (70), one has

$$\frac{1}{n}\sum_{k=1}^{n}Z_{k,i} = [\mathcal{L}^*\boldsymbol{S}]_i. \tag{71}$$

We can see $Y_{k,i} \sim N(0, \sigma_i^2)$ because of the fact that any linear combination of $p$ components in $\tilde{\boldsymbol{X}}$ has a univariate normal distribution. The variance of $Y_{k,i}$ is

$$
\begin{aligned}
\sigma_i^2 &= \mathbb{E}\big[(Y_{k,i} - \mathbb{E}(Y_{k,i}))^2\big] = \mathbb{E}\big[(\tilde{X}_a^{(k)} - \tilde{X}_b^{(k)})^2\big] \\
&= \tilde{\Sigma}_{aa} + \tilde{\Sigma}_{bb} - \tilde{\Sigma}_{ab} - \tilde{\Sigma}_{ba} = \big(\mathcal{L}^*\tilde{\boldsymbol{\Sigma}}\big)_i.
\end{aligned}
\tag{72}
$$

Therefore $Z_{k,i}/\sigma_i^2 \sim \chi^2(1)$ and $\mathbb{E}[Z_{k,i}/\sigma_i^2] = 1$. We say a random variable $X$ is *sub-exponential* if there are non-negative parameters $(v, \alpha)$ such that

$$\mathbb{E}\big[\exp\big(\lambda(X - \mathbb{E}[X])\big)\big] \leq \exp(\frac{v^2\lambda^2}{2}), \text{ for all } |\lambda| < \frac{1}{\alpha}. \tag{73}$$

By checking the condition in (73), one can conclude that $Z_{k,i}/\sigma_i^2$ is is sub-exponential with parameters $(2,4)$. Furthermore, if random variables $\{Y_k\}_{k=1}^n$ are independent and sub-exponential with parameters $(v_k, \alpha_k)$, then $\sum_{k=1}^n Y_k$ is still sub-exponential with parameters $(v_*, \alpha_*)$ where

$$v_* := \sqrt{\sum_{k=1}^n v_k^2} \qquad \text{and} \qquad \alpha_* := \max_{k=1,\ldots,n} \alpha_k.$$

Thus, $\sum_{k=1}^n Z_{k,i}/\sigma_i^2$ is sub-exponential with parameters $(2\sqrt{n}, 4)$. The application of the sub-exponential tail bound in Lemma 1.11 yields

$$\mathbb{P}\left[\left|\sum_{k=1}^n Z_{k,i}/\sigma_i^2 - n\right| \geq t_0\right] \leq 2\exp\left(-\frac{t_0^2}{8n}\right), \quad \text{for} \ \ t_0 \in [0, n].$$

By taking $t_0 = nt/\max_i \sigma_i^2$, one has

$$\mathbb{P}\left[\left|\frac{1}{n}\sum_{k=1}^n Z_{k,i} - \sigma_i^2\right| \geq t\right] = \mathbb{P}\left[\left|\sum_{k=1}^n Z_{k,i}/\sigma_i^2 - n\right| \geq \frac{nt}{\sigma_i^2}\right]$$

$$\leq \mathbb{P}\left[\left|\sum_{k=1}^n Z_{k,i}/\sigma_i^2 - n\right| \geq \frac{nt}{\max_i \sigma_i^2}\right]$$

$$\leq 2\exp\left(-\frac{nt^2}{8\left(\max_i \sigma_i^2\right)^2}\right) \tag{74}$$

holds for $t \in [0, \max_i \sigma_i^2]$. Notice that $\sigma_i^2 = \left(\mathcal{L}^*(\mathcal{L}\boldsymbol{w}^\star + \boldsymbol{J})^{-1}\right)_i$ according to (72). Substituting (71) into (74) yields

$$\mathbb{P}\left[\left|[\mathcal{L}^*\boldsymbol{S}]_i - \left(\mathcal{L}^*(\mathcal{L}\boldsymbol{w}^\star + \boldsymbol{J})^{-1}\right)_i\right| \geq t\right] \leq 2\exp(-c_0 nt^2), \quad \text{for} \ \ t \in [0, t_0]. \tag{75}$$

where $t_0 = \left\|\mathcal{L}^*(\mathcal{L}\boldsymbol{w}^\star + \boldsymbol{J})^{-1}\right\|_{\max}$ and $c_0 = 1/\left(8\left\|\mathcal{L}^*(\mathcal{L}\boldsymbol{w}^\star + \boldsymbol{J})^{-1}\right\|_{\max}^2\right)$, completing the proof. $\qquad\square$

## 2.9 Proof of Lemma 1.9

*Proof.* Recall that $\mathbb{B}_{\boldsymbol{M}}(\boldsymbol{w}^\star; r) = \{\boldsymbol{w} \in \mathbb{R}^{p(p-1)/2} \mid \|\boldsymbol{w} - \boldsymbol{w}^\star\|_{\boldsymbol{M}} \leq r\}$, where $\|\boldsymbol{x}\|_{\boldsymbol{M}}^2 = \langle \boldsymbol{x}, \boldsymbol{M}\boldsymbol{x}\rangle = \|\mathcal{L}\boldsymbol{x}\|_{\mathrm{F}}^2$ with $\boldsymbol{M} \succ \boldsymbol{0}$ defined in Lemma 1.3, and $\mathcal{S}_{\boldsymbol{w}} = \{\boldsymbol{w} \mid \boldsymbol{w} \geq \boldsymbol{0}, (\mathcal{L}\boldsymbol{w} + \boldsymbol{J}) \in \mathcal{S}_{++}^p\}$ We can see $\mathcal{B}_{\boldsymbol{M}}(\boldsymbol{w}^\star; r)$ is a convex set because both $\mathbb{B}_{\boldsymbol{M}}(\boldsymbol{w}^\star; r)$ and $\mathcal{S}_{\boldsymbol{w}}$ are convex. It is easy to check that $\mathcal{S}_{\boldsymbol{w}}$ is convex (See (4) for more details). For any $\boldsymbol{w}_1, \boldsymbol{w}_2 \in \mathcal{B}_{\boldsymbol{M}}(\boldsymbol{w}^\star; r)$, by Mean Value Theorem, one obtains

$$f(\boldsymbol{w}_2) = f(\boldsymbol{w}_1) + \langle \nabla f(\boldsymbol{w}_1), \boldsymbol{w}_2 - \boldsymbol{w}_1 \rangle + \frac{1}{2}\langle \boldsymbol{w}_2 - \boldsymbol{w}_1, \nabla^2 f(\boldsymbol{w}_t)(\boldsymbol{w}_2 - \boldsymbol{w}_1)\rangle, \tag{76}$$

where $\boldsymbol{w}_t = t\boldsymbol{w}_2 + (1-t)\boldsymbol{w}_1$ with $t \in [0,1]$. For any nonzero $\boldsymbol{x} \in \mathbb{R}^{p(p-1)/2}$, one has

$$\boldsymbol{x}^\top \nabla^2 f(\boldsymbol{w}_t)\boldsymbol{x} = (\mathrm{vec}(\mathcal{L}\boldsymbol{x}))^\top \left((\mathcal{L}\boldsymbol{w}_t + \boldsymbol{J})^{-1} \otimes (\mathcal{L}\boldsymbol{w}_t + \boldsymbol{J})^{-1}\right)\mathrm{vec}(\mathcal{L}\boldsymbol{x})$$

$$= \frac{(\mathrm{vec}(\mathcal{L}\boldsymbol{x}))^\top \left((\mathcal{L}\boldsymbol{w}_t + \boldsymbol{J})^{-1} \otimes (\mathcal{L}\boldsymbol{w}_t + \boldsymbol{J})^{-1}\right)\mathrm{vec}(\mathcal{L}\boldsymbol{x})}{(\mathrm{vec}(\mathcal{L}\boldsymbol{x}))^\top \mathrm{vec}(\mathcal{L}\boldsymbol{x})} \cdot \|\mathcal{L}\boldsymbol{x}\|_{\mathrm{F}}^2$$

$$\geq \inf_{\boldsymbol{y}} \frac{(\mathrm{vec}(\mathcal{L}\boldsymbol{y}))^\top \left(((\mathcal{L}\boldsymbol{w}_t)^\dagger + \boldsymbol{J}) \otimes ((\mathcal{L}\boldsymbol{w}_t)^\dagger + \boldsymbol{J})\right)\mathrm{vec}(\mathcal{L}\boldsymbol{y})}{(\mathrm{vec}(\mathcal{L}\boldsymbol{y}))^\top \mathrm{vec}(\mathcal{L}\boldsymbol{y})} \cdot \|\mathcal{L}\boldsymbol{x}\|_{\mathrm{F}}^2$$

$$= \inf_{\boldsymbol{y}} \frac{(\mathrm{vec}(\mathcal{L}\boldsymbol{y}))^\top \left((\mathcal{L}\boldsymbol{w}_t)^\dagger \otimes (\mathcal{L}\boldsymbol{w}_t)^\dagger\right)\mathrm{vec}(\mathcal{L}\boldsymbol{y})}{(\mathrm{vec}(\mathcal{L}\boldsymbol{y}))^\top \mathrm{vec}(\mathcal{L}\boldsymbol{y})} \cdot \|\mathcal{L}\boldsymbol{x}\|_{\mathrm{F}}^2$$

$$\geq \lambda_2^2\left((\mathcal{L}\boldsymbol{w}_t)^\dagger\right) \cdot \|\mathcal{L}\boldsymbol{x}\|_{\mathrm{F}}^2, \tag{77}$$

where $\lambda_2\left((\mathcal{L}\boldsymbol{w}_t)^\dagger\right)$ denotes the second smallest eigenvalue of $(\mathcal{L}\boldsymbol{w}_t)^\dagger$ and $(\mathcal{L}\boldsymbol{w}_t)^\dagger$ is the pseudo inverse of $\mathcal{L}\boldsymbol{w}_t$. The first equality follows from Lemma 1.1; the last equality follows from $\left(\mathrm{vec}(\mathcal{L}\boldsymbol{y})\right)^\top\left((\mathcal{L}\boldsymbol{w}_t)^\dagger\otimes\boldsymbol{J}\right)\mathrm{vec}(\mathcal{L}\boldsymbol{y}) = 0$, $\left(\mathrm{vec}(\mathcal{L}\boldsymbol{y})\right)^\top\left(\boldsymbol{J}\otimes(\mathcal{L}\boldsymbol{w}_t)^\dagger\right)\mathrm{vec}(\mathcal{L}\boldsymbol{y}) = 0$, and $\left(\mathrm{vec}(\mathcal{L}\boldsymbol{y})\right)^\top(\boldsymbol{J}\otimes\boldsymbol{J})\mathrm{vec}(\mathcal{L}\boldsymbol{y}) = 0$ which are easy to verify; the last inequality holds because of the property of Kronecker product that the eigenvalues of $\boldsymbol{A}\otimes\boldsymbol{B}$ are $\lambda_i\mu_j$ with the corresponding eigenvector $\boldsymbol{a}_i\otimes\boldsymbol{b}_j$, where $\lambda_1,\ldots,\lambda_p$ are the eigenvalues of $\boldsymbol{A}\in\mathbb{R}^{p\times p}$ with the corresponding eigenvectors $\boldsymbol{a}_1,\ldots,\boldsymbol{a}_p$, and $\mu_1,\ldots,\mu_p$ are the eigenvalues of $\boldsymbol{B}\in\mathbb{R}^{p\times p}$ with the corresponding eigenvectors $\boldsymbol{b}_1,\ldots,\boldsymbol{b}_p$. Notice that there is one and only one zero eigenvalue for $\mathcal{L}\boldsymbol{w}_t$ because $\boldsymbol{w}_t\in\mathcal{S}_{\boldsymbol{w}}$. Assume $\lambda_1,\ldots,\lambda_p$ are the eigenvalues of $(\mathcal{L}\boldsymbol{w}_t)^\dagger$ with the corresponding eigenvectors $\boldsymbol{a}_1,\ldots,\boldsymbol{a}_p$. Without loss of generality, let $\lambda_1 = 0$ and then $\boldsymbol{a}_1 = \frac{1}{\sqrt{p}}\boldsymbol{1}$. By calculation, one obtains

$$(\mathrm{vec}(\mathcal{L}\boldsymbol{y}))^\top\mathrm{vec}(\boldsymbol{a}_i\otimes\boldsymbol{a}_1) = 0, \quad\text{and}\quad (\mathrm{vec}(\mathcal{L}\boldsymbol{y}))^\top\mathrm{vec}(\boldsymbol{a}_1\otimes\boldsymbol{a}_i) = 0,$$

for any $i = 1,\ldots,p$ and any $\boldsymbol{y}\in\mathbb{R}^{p(p-1)/2}$, indicating that $\mathrm{vec}(\mathcal{L}\boldsymbol{y})$ is orthogonal to all the eigenvectors of $(\mathcal{L}\boldsymbol{w}_t)^\dagger\otimes(\mathcal{L}\boldsymbol{w}_t)^\dagger$ corresponding to zero eigenvalues. The smallest nonzero eigenvalue of $(\mathcal{L}\boldsymbol{w}_t)^\dagger\otimes(\mathcal{L}\boldsymbol{w}_t)^\dagger$ is $\lambda_2^2\left((\mathcal{L}\boldsymbol{w}_t)^\dagger\right)$, establishing (77). Notice that (77) also holds with $\boldsymbol{x} = \boldsymbol{0}$. One further obtains

$$\lambda_2^2\left((\mathcal{L}\boldsymbol{w}_t)^\dagger\right) \geq \left(\|\mathcal{L}\boldsymbol{w}^\star\|_2 + (1-t)\|\mathcal{L}\boldsymbol{w}_1 - \mathcal{L}\boldsymbol{w}^\star\|_2 + t\|\mathcal{L}\boldsymbol{w}_2 - \mathcal{L}\boldsymbol{w}^\star\|_2\right)^{-2} \geq \left(\|\mathcal{L}\boldsymbol{w}^\star\|_2 + r\right)^{-2}, \tag{78}$$

where the second inequality is established by $\|\mathcal{L}\boldsymbol{x}\|_2 \leq \|\mathcal{L}\boldsymbol{x}\|_{\mathrm{F}}$ and the fact that both $\boldsymbol{w}_1, \boldsymbol{w}_2 \in \mathcal{B}_{\boldsymbol{M}}(\boldsymbol{w}^\star; r)$. Substituting (78) into (77) yields

$$\boldsymbol{x}^\top\nabla^2 f(\boldsymbol{w}_t)\boldsymbol{x} \geq \left(\|\mathcal{L}\boldsymbol{w}^\star\|_2 + r\right)^{-2}\cdot\|\mathcal{L}\boldsymbol{x}\|_{\mathrm{F}}^2. \tag{79}$$

Combining (76) and (79) yields

$$f(\boldsymbol{w}_2) \geq f(\boldsymbol{w}_1) + \langle\nabla f(\boldsymbol{w}_1), \boldsymbol{w}_2 - \boldsymbol{w}_1\rangle + \frac{1}{2}\left(\|\mathcal{L}\boldsymbol{w}^\star\|_2 + r\right)^{-2}\|\mathcal{L}\boldsymbol{w}_1 - \mathcal{L}\boldsymbol{w}_2\|_{\mathrm{F}}^2, \tag{80}$$

and

$$f(\boldsymbol{w}_1) \geq f(\boldsymbol{w}_2) + \langle\nabla f(\boldsymbol{w}_2), \boldsymbol{w}_1 - \boldsymbol{w}_2\rangle + \frac{1}{2}\left(\|\mathcal{L}\boldsymbol{w}^\star\|_2 + r\right)^{-2}\|\mathcal{L}\boldsymbol{w}_1 - \mathcal{L}\boldsymbol{w}_2\|_{\mathrm{F}}^2, \tag{81}$$

Combining (80) and (81), we establish

$$\langle\nabla f(\boldsymbol{w}_1) - \nabla f(\boldsymbol{w}_2), \boldsymbol{w}_1 - \boldsymbol{w}_2\rangle \geq \left(\|\mathcal{L}\boldsymbol{w}^\star\|_2 + r\right)^{-2}\|\mathcal{L}\boldsymbol{w}_1 - \mathcal{L}\boldsymbol{w}_2\|_{\mathrm{F}}^2, \tag{82}$$

completing the proof. □

## 3 Additional Experimental Results

In this section, we present additional numerical simulation results on synthetic data and real-world data.

### 3.1 Synthetic Data

We present additional simulation results on random Barabasi-Albert graphs. Figure 1 illustrates the histograms of the nonzero graph weights, which are learned by the $\ell_1$-norm regularization method with different regularization parameters as shown in Figure 1 of the paper. Figure 1 (d) depicts the histogram for the graph in Figure 1 (d) of the paper, which is a fully connected graph, i.e., every graph weight is strictly positive. It is observed in Figure 1 (d) that all the graph weights are very small. Therefore, a large regularization parameter will lead to learn a graph with every weight strictly positive and small. We can see that the histogram in Figure 1 (d) is significantly different from the true histogram in Figure 1 (a), implying that the estimated model will fail to identify the true relationships among the data variables.

Figure 1: Histograms of nonzero graph weights learned by the $\ell_1$-norm regularization method with different regularization parameters, corresponding to the graphs in Figure 1 of the paper. The histograms count the number of nonzero graph weights falling into each interval.

Figure 2: A sample result of learning a Barabasi-Albert graph of degree one by (b) GLE-ADMM [8], (c) NGL-SCAD (proposed) and (d) NGL-MCP (proposed). The sample size ratio is $n/p = 6$. NE denotes the number of positive edges in the graph, i.e., the number of nonzero graph weights. The regularization parameters for each method are set as $\lambda_{\mathsf{ADMM}} = 0$, $\lambda_{\mathsf{SCAD}} = \lambda_{\mathsf{MCP}} = 0.5$.

Figure 2 shows a sample result of learning a random Barabasi-Albert graph via GLE-ADMM [8], NGL-SCAD and NGL-MCP. It is observed that the graphs learned by NGL-SCAD and NGL-MCP present the connection between any two nodes correctly, while there are many incorrect connections in the graph learned by GLE-ADMM. In addition, performance measures including sparsity, relative error, and F-score also indicate a better performance of the proposed method.

Table 1 shows the running time for different numbers of nodes $p$, where the graph weight $\boldsymbol{w}$ has the dimension $p(p-1)/2$. It is observed that the computational time of the proposed NGL-SCAD is much less than that of GLE-ADMM.

## 3.2 Real-world Data

In this section, we compare the proposed method with the benchmark GLE-ADMM [8] on two real-world data sets.

We conduct experiments on the COVID-19 data set[1] provided by the Israelite Hospital Albert Einstein in Brazil. The data set contains anonymized data from patients who had samples collected to perform the test

Table 1: Comparison of computational time (seconds)

| $p$ | 50 | 100 | 500 | 1000 |
|---|---|---|---|---|
| GLE-ADMM | 0.117 | 0.676 | 57.032 | 485.465 |
| NGL-SCAD | 0.023 | 0.097 | 9.000 | 67.018 |

Figure 3: Learned graphs using the Brazilian COVID-19 data set via (a) GLE-ADMM, (b) NGL-SCAD (proposed), and (c) NGL-MCP (proposed). The computational time for GLE-ADMM, NGL-SCAD, and NGL-MCP are 9.9, 55.8 and 57.9 seconds, respectively. The regularization parameters are set as $\lambda_{\mathsf{ADMM}} = 0$, $\lambda_{\mathsf{SCAD}} = 0.1$, and $\lambda_{\mathsf{MCP}} = 0.5$.

Figure 4: Stock graphs learned via (a) GLE-ADMM, (b) NGL-SCAD (proposed), and (c) NGL-MCP (proposed). The computational time for GLE-ADMM, NGL-SCAD, and NGL-MCP are 20.5, 1.3 and 90.3 seconds, respectively. The regularization parameters are set as $\lambda_{\mathsf{ADMM}} = 0$, $\lambda_{\mathsf{SCAD}} = 0.25$, and $\lambda_{\mathsf{MCP}} = 0.5$.

for SARS-CoV-2. The features in the data set are mainly clinical coming from blood, urine, and saliva exams, e.g., hemoglobin level, platelets, red blood cells, etc. The original data set contains 108 features from 558 patients. We do not consider features that were measured for at most 10 patients due to the high number of missing values. In addition, a large number of patients had no record of any features. Finally, we end up with a data matrix of 182 patients with 57 features, i.e., $p = 182$ and $n = 57$. The remaining missing values were filled in with zeros. We then compare the proposed method with the GLE-ADMM method on this data set. It is observed in Figure 3 that the proposed NGL-SCAD and NGL-MCP output a more interpretable representation of the network, where the blue and red nodes denote patients who tested negative and positive for SARS-CoV-2, respectively.

We also conduct experiments on the data set from stocks composing the S&P 500 index. We select log-returns from 181 stocks from 4 sectors, namely: "Industrials", "Consumer Staples", "Energy", "Information Technology", during a period of 4 years from January 1st 2016 to May 20th 2020, with a total of 1101 observations. Then the data matrix $\boldsymbol{X}$ has a size of $181 \times 1101$. The graphs are learned on the basis of the sample correlation matrix. Figure 4 shows that the graphs learned by NGL-SCAD and NGL-MCP are able to vividly display the sectors modularity, whereas the graph learned via GLE-ADMM fails to do so.

Finally, we present additional experimental results of GLE-ADMM on the 2019-nCoV data set with

Figure 5: The learned graphs by GLE-ADMM with (a) $\lambda = 0$, (b) $\lambda = 0.02$ and $\lambda = 0.1$ on the 2019-nCoV data set.

different regularization parameters. It is observed in Figure 5 that the learned graphs learned by GLE-ADMM are always dense along the increase of $\lambda$, implying that the $\ell_1$-norm is not able to impose a sparse solution here.

## Footnotes

[1]The data set is freely available at: `https://www.kaggle.com/einsteindata4u/covid19`