[Reviews · NeurIPS 2020]

Review 1

Summary and Contributions: The paper studies the problem of learning Laplacian-structured Gaussian graphical models (GGM). In a slight departure from the standard setting, here the precision matrix is interpreted as the Laplacian of a sparse, connected graph. The proposed solution follows the familiar approach of log-det regularized estimation with constraint-promoting regularizers, although with some innovations. The authors make four contributions: * First, they show that the typical L1-penalty used in regular sparse GGM learning does not succeed here, in the sense that the solution to the regularized problem returns dense graphs. (This requires a surprisingly intricate proof.) * Second, they develop a majorization-minimization scheme for approximating the solution to the log-det regularization problem with non-convex sparsity penalties (such as MCP or SCAD). * Third, they derive bounds on the sample complexity required for their proposed scheme to succeed. * Fourth, they support their method by testing performance on representative datasets.

Strengths: This is a good paper that seems to make interesting theoretical contributions to the area of Laplacian GGM learning. The technique of using majorization-minimization to approximate the solution of non-convex optimization problems is not particularly new or surprising, given its long history. Still, application of this technique to this specific problem seems to be novel. The theoretical bounds on sample complexity mostly match existing results for standard GGM learning. The experimental results are limited but effective. The nCoV dataset result is topical, but I am not entirely sure if the results provide any real actionable information. The presented results seem more along the lines of a sanity check on the algorithm rather than really addressing a knowledge. gap.

Weaknesses: Given that it is a theory-oriented paper, it may be helpful to provide essential elements of the proof techniques (particularly, Theorem 3.1 and 3.8) in the main paper, what special insights were used, and why previous approaches/techniques are not directly applicable. Comparisons/contrasts with previous approaches in GGM learning that employ nonconvex penalties would put the contributions in better context; e.g. Wang-Ren-Gu AISTATS 2016.

Correctness: Proofs are probably correct; I didn't check completely. The method seems sound and the results are in line with what we expect.

Clarity: The paper is well written (modulo minor weaknesses in presentation that I have pointed out above).

Relation to Prior Work: It will be helpful to add some discussion on: * how the method differs/improves upon previous approaches for nonconvex GGM learning. Among others: - Wang-Ren-Gu AISTATS 2016. - Xu-Ma-Gu NeurIPS 2017 * what new techniques/ideas are used to establish the proofs of the main theorems

Reproducibility: Yes

Additional Feedback: Questions for the authors: * How do you initialize w in Algorithm 1? * How do you initialize the PGD routine in Step 3? * why is c0 considered to be a constant? (It seems to depend on the solution). [update after rebuttal] I continue to be supportive of the paper. I agree with R2 that an intuitive reason why the L1 norm fails to promote sparsity in this particular case is missing. The proof given in the appendix is somewhat dense, but looks to be correct as far as I can tell. My own understanding is that this has to do with the specific structure of the sparsity-regularized log-det minimization problem: * In the standard graphical LASSO of Banerjee/Freedman, if we examine the KKT conditions at the optimal points we can see that they enforce shrinkage of the entries of the precision matrix (and increasing regularization shrinks more entries). * Here, the KKT conditions are subtly different (see equations 13-15 in the appendix). Since the unknown variable is a Laplacian, a) there is an affine transformation (adding an all-ones matrix) to make the problem non-singular, and b) due to positivity the L1 norm reduces to a simple summation. The combination of these two implies that the L1 norm has no sparsifying effect in this regime. (the authors can/should probably articulate this better)


Review 2

Summary and Contributions: This paper provides a mathematical analysis of and algorithmic framework for learning Laplacian-structured graphical models. A major and surprising result shows that the l_1 norm is not effective for identifying a sparse solution, and in fact, can return arbitrarily dense graphs as the l_1 regularization parameter increases. A nonconvex algorithm is demonstrated on synthetic and real-world examples.

Strengths: - The paper provides a very interesting mathematical and algorithmic framework, with a surprising result about sparsity-inducing algorithms using the l_1 norm. I expect that this could have major implications for many researchers today.

Weaknesses: - Why should the real data example follow a Laplacian-structured graphical model? This is currently not a convincing example, because it is not clear that the framework is suited to tackle this data set. The final sentence in section 4.2 is not sufficient. - It would really make the paper great to provide some more discussion and insight into why the l_1 norm causes arbitrarily dense graphs. - I found section 3.2 hard to follow. Some separation between steps would be helpful. - The broader impact section is somewhat weak and could be expanded. Is this framework actually trustworthy enough to base health care decisions on it?

Correctness: The claims and methods are correct to the best of my knowledge.

Clarity: The paper is not badly written but neither is it clean and polished. Some sections are more confusing, such as 3.2. Sections 2.2 and 5 are written in a formulaic and not very elegant way. The introduction is quite clear.

Relation to Prior Work: Yes.

Reproducibility: Yes

Additional Feedback: UPDATE: As discussed during reviews and response, it would really strengthen the paper to provide some intuition behind this surprising result, and I look forward to reading the revision. Thanks for the helpful response.


Review 3

Summary and Contributions: This paper proposes NGL, a new algorithm for problem of learning a sparse graph. It proves that the L1-norm is not effective in imposing a sparse solution in this problem,and demonstrates the effectiveness of the proposed methods NGLwith some empirical results.

Strengths: As far as I know, the idea of the paper is novel and interesting. Graphical lasso is widely used in graph learning to get the sparser result when a larger regularization parameter is used, but this paper presents a completely opposite view. They provide sufficient theoretical proofs to support the their claims, and propose a brand new algorithm for learning a sparse graph which is effective and powerful.

Weaknesses: 1. The descriptions of the first two sections are too casual. The introduction part declare a view that is contrary to the conventional wisdom, but does not make clear the difference between the two cases. Only five papers are mentioned in the related work, which is not enough to show the process in this field. 2. One of the main contributions of this paper is the proof of Theorem 3.1. I notice that the regularization parameter must satisfy a constraint to obtain Theorem 3.1. The left boundary of the constraint set is important to support the view “a large regularization parameter of the L1-norm will make every graphweight strictly positive and thus the estimated graph is fully connected”, whether the left boundary of the constraint set is a large regularization parameter or just a normal size? There should have some discussion about this to make the Theorem 3.1 more credible. 3. The current empirical results are not very convincingand at times confusing. The experiments on the 2019-nCoV data set, while interesting, are a bit too toy. You should try more hyper-parameters for each algorithm and show the whole experiment results. 4. The author should check the whole paper because there are some errors in this paper: --The formula below line 150 should have a comma. --Line 230, the presentation is confusing, “1.01 in h1 and 2.01 in h2’’ is a better statement. --Line 238, right half, the results of. --Line 241-242, the sentence was formed in a confusing way, “results of graphs learned by’’seems better than“results of learning graphs by’’.

Correctness: Yes, this paper provides theoretical proofs to support their claims.

Clarity: I think the paper could be explained better. As I had elaborate in the weakness section, more details should be included in the introduction part and related work part.

Relation to Prior Work: The paper raises a completely opposite point from the popular opinions. Difference between this work and prior work is clearly. But some more straightforward and potentially important comparisons are missing.

Reproducibility: Yes

Additional Feedback: There should be more discussion about the problems that the algorithm may encounter in practice.


Review 4

Summary and Contributions: Authors propose a nonconvex penalized maximum likelihood estimation method, and prove that the order of statistical error matches that of the minimax lower bound.

Strengths: Authors studied l_1-norm regularized Laplacian-structured Gaussian graphical model and observed that a large regularization parameter leads to fully connected graph. Nonconvex regularizations are used such as MCP and SCAD to improve the sparsity of the Laplacian-structured Gaussian graphical model. Theoretical results are presented. Experiments are conducted on synthetic and real-world data.

Weaknesses: Theorem 3.1 and its empirical results in Figure 1 bring the following concern: The fully connected graph obtained by lambda larger than certain value (e.g., lambda*), which is related to the input data covariance S. So, the scaling of X impacts lambda*. How does the lambda from 0 to lambda* theoretically impact the sparsity of the optimal graph? The proposed algorithm for solving nonconvex optimization is the combination of linearization of nonconvex regularization and projected gradient descent method. The convergence results are shown in Theorem 3.8. This can guarantee the convergence, but how many iterations are required for both inner and outer iterations in Algorithm 1 and what is the computational complexity? Due to the high complexity of the algorithm for computing logdet and inverse of pxp matrices, it is interesting to see if it can be scaled for mediate or high-dimensional data sets. The real-word data 2019-nCoV is used in this paper, but it is doubt that this application is suitable for the proposed model. There are four main concerns: 1) the categorical and integer features are encoded to zero/one features via one-hat-encoding. This will break the assumption of Gaussian graphical model since the feature is no longer following Gaussian distribution. 2) the results are not verified. In other words, which results are more correct is unknown. 3)As to sparsity, authors only use lambda_{ADMM}=0, it is not convincing if GLE-ADMM can also obtain the similar sparser graph as the proposed method by fine-tuning lambda. 4) the data sets in experiments are too small. It is interesting to see results on moderate or high-dimensional data.

Correctness: The models are reasonable, but the empirical results might be questionable.

Clarity: It is well written.

Relation to Prior Work: A number of relevant works are discussed.

Reproducibility: Yes

Additional Feedback: I have read the authors' rebuttal, and most of my concerns are addressed. I recommend the paper for accept.

[Author Response · NeurIPS 2020]

**To Reviewer #1: Thank you for your supportive comments.**

**A1: Special insights in proofs.** The proof strategy of Theorem 3.1 is based on the construction of an inverse $M$-matrix. To the best of our knowledge, this is the first work to discuss and prove this special behavior of the $\ell_1$ norm. In the proof of Theorem 3.8, we introduce a linear operator to handle the Laplacian constraints and derive a concentration bound for Laplacian GGM, which are new and necessary for the proof. The previous approaches do not solve these problems.

**Other comments:** The initialization of Algorithm 1 is generated by converting the lower triangle matrix of $(\boldsymbol{S} + \epsilon\boldsymbol{I})^{-1}$ into a vector and removing the negative elements. For the $k$-th iteration, the initialization of Step 3 is set as $\hat{\boldsymbol{w}}^{k-1}$. The $c_0$ only depends on the true graph weights. The papers Wang-Ren-Gu AISTATS 2016 and Xu-Ma-Gu NeurIPS 2017 will be discussed. The performance of GGM methods is usually not satisfactory due to lack of Laplacian constraints.

**To Reviewer #2: Thank you for your supportive comments.**

**B1: COVID-19 data with Laplacian GGM.** In the case of the data following Laplacian GGM, our formulation in Eq. (3) can be viewed as a regularized maximum likelihood estimation of precision matrix. In a more general setting with non-Gaussian distribution such as COVID-19 data, Eq. (3) can be related to the log-determinant Bregman divergence regularized optimization, and the learned graph weights can quantify the similarity between nodes. This is because the trace term in Eq. (3) can be written as Laplacian quadratic [10, 20], which tends to assign a large weight between nodes if their signal values are similar. The COVID-19 data consists of two groups, red and green nodes. It is natural to assume that the nodes belonging to different groups are dissimilar from each other, while the nodes in the same group are similar. In this sense, the performance of our learned graph in Figure 4 is significant, because most connections are between nodes within the same group, and only a few connections (gray edges) are between nodes from distinct groups.

**Other comments:** The insight behind the unexpected behavior of $\ell_1$-norm is related to the Laplacian constraints, and related discussions will be added. We will further clarify the steps in Section 3.2, and update the broader impact section.

**To Reviewer #3: Thank you for your helpful comments.**

**C1: Introduction and Related Work.** We will include additional discussions on GGM and clarify the difference between GGM and Laplacian GGM. Some related works on GGM methods are discussed in the Introduction, and thus not repeated in the Related Work due to limited pages. We will reorganize the first two sections in the final version.

**C2: Left boundary in Theorem 3.1.** We apologize that we did not make it clear our statement that a large regularization parameter of the $\ell_1$-norm method will lead to learn a fully connected graph. We rephrase it here: As long as the regularization parameter is large enough (larger than the left boundary in Theorem 3.1), the learned graph by the $\ell_1$-norm method must be fully connected. In practice, the bound leading to a fully connected graph may be smaller than the left boundary in the theorem.

**C3: Experiments on COVID-19 data.** We indeed tested many values of $\lambda$ for GLE-ADMM. The presented result with $\lambda = 0$ in the paper is the sparsest one, and the other estimated graphs are denser. Due to the limited space, here we only show a result with $\lambda = 0.1$ in Figure 1 that is already a fully connected graph. We will add additional results in Appendix of the final version. We agree that the COVID-19 data set is small. This data set was collected while the pandemic was in its infancy, and thus the data availability was limited. We will update our results with a bigger data set in the final version.

Figure 1: GLE-ADMM with $\lambda = 0.1$.

**Other comments:** Thank you for pointing out these typos and we will correct them.

**To Reviewer #4: Thank you for your supportive comments.**

**D1: Theorem 3.1 and its empirical results.** 1. We agree that the scaling of the sample data impacts $\lambda^*$. The theoretical characterization of the function of sparsity with respect to $\lambda$ for $\lambda \in [0, \lambda^*]$ is challenging, because the sparsity involves counting the number of nonzero elements which is always highly non-convex and non-continuous. 2. It is precisely the point of our paper to show under solid mathematical grounds and illustrate via extensive numerical results that the $\ell_1$-norm method is an incorrect approach and many papers are following it improperly.

**D2: Scalability of the algorithm.** The computational complexity of our algorithm is dominated by Cholesky decomposition, which can save computation for logdet and matrix inverse. Table 1 shows the running time for different numbers of nodes $p$, where the graph weight $\boldsymbol{w}$ has the dimension $p(p-1)/2$. The numbers of outer iterations for different $p$ are 28, 28, 29 and 26, respectively, and the average numbers of inner iterations are usually less than 5. It is of great interest to develop more efficient algorithms and implement algorithms in faster programming languages such as C++ in order to accommodate larger data sets in future work.

**D3: COVID-19 data.** 1. Regarding the concern that the data does not follow Laplacian GGM, please refer to **B1**; 2. We agree that the quality of the results cannot be precisely verified, because there is no underlying true graph in practice.

Table 1: Comparison of computational time (seconds)

| $p$ | 50 | 100 | 500 | 1000 |
|---|---|---|---|---|
| GLE-ADMM | 0.117 | 0.676 | 57.032 | 485.465 |
| NGL-SCAD | 0.023 | 0.097 | 9.000 | 67.018 |

Nonetheless, the synthetic experiments has shown the ability of our method to learn the ground truth. The significance of our result on COVID-19 data can be verified in some sense (Refer to **B1**). 3. Regarding GLE-ADMM with other parameter values and the size of data set, please refer to **C3**.

[Meta-Review · NeurIPS 2020]

All reviewers but one agree that the contribution of the paper is novel, interesting and that the experiments are compelling. Bit all reviewers also agree that the paper lacks either an intuitive or even formal explanation of the surprizing properties of the L1 norm regularization in this setting. The proofs provided show that the stated facts are correct, but this is not explained in the text. The authors are strongly encouraged to include an explanation in the final version of the manuscript.